# ANIMATE YOUR THOUGHTS: RECONSTRUCTION OF DYNAMIC NATURAL VISION FROM HUMAN BRAIN ACTIVITY

Yizhuo Lu[1,2,†], Changde Du[1,†], Chong Wang[4], Xuanliu Zhu[5], Liuyun Jiang[1,2], Xujin Li[1,2], and Huiguang He[1,2,3*]

[1]Key Laboratory of Brain Cognition and Brain-inspired Intelligence Technology, Institute of Automation, Chinese Academy of Sciences
[2]School of Future Technology, University of Chinese Academy of Sciences
[3]School of Artificial Intelligence, University of Chinese Academy of Sciences
[4]School of Computer and Artificial Intelligence, Zhengzhou University
[5]Beijing University of Posts and Telecommunications
[†]Equal Contribution
[*]Corresponding Author
{luyizhuo2023, changde.du, huiguang.he}@ia.ac.cn

## ABSTRACT

Reconstructing human dynamic vision from brain activity is a challenging task with great scientific significance. Although prior video reconstruction methods have made substantial progress, they still suffer from several limitations, including: (1) difficulty in simultaneously reconciling semantic (e.g. categorical descriptions), structure (e.g. size and color), and consistent motion information (e.g. order of frames); (2) low temporal resolution of fMRI, which poses a challenge in decoding multiple frames of video dynamics from a single fMRI frame; (3) reliance on video generation models, which introduces ambiguity regarding whether the dynamics observed in the reconstructed videos are genuinely derived from fMRI data or are hallucinations from generative model. To overcome these limitations, we propose a two-stage model named Mind-Animator. During the fMRI-to-feature stage, we decouple semantic, structure, and motion features from fMRI. Specifically, we employ fMRI-vision-language tri-modal contrastive learning to decode semantic feature from fMRI and design a sparse causal attention mechanism for decoding multi-frame video motion features through a next-frame-prediction task. In the feature-to-video stage, these features are integrated into videos using an inflated Stable Diffusion, effectively eliminating external video data interference. Extensive experiments on multiple video-fMRI datasets demonstrate that our model achieves state-of-the-art performance. Comprehensive visualization analyses further elucidate the interpretability of our model from a neurobiological perspective. Project page: https://mind-animator-design.github.io/.

## 1 INTRODUCTION

Advances in sensory neuroscience offer new perspectives on brain function and could enhance artificial intelligence development (Palazzo et al. (2021); Yargholi & Hossein-Zadeh (2016)). One of the critical aspects to the research is neural decoding, which links visual stimuli to corresponding functional magnetic resonance imaging (fMRI) brain recordings. Neural decoding methods include classification, identification, and reconstruction, with this study focusing on the most challenging aspect: reconstruction.

Prior methods have made significant strides in the classification (Yargholi & Hossein-Zadeh (2016); Horikawa & Kamitani (2017); Fujiwara et al. (2013)) and identification (Kay et al. (2008); Wildgruber et al. (2005)) of **static** stimulus images. Remarkably, some researchers have advanced to the point

where they can reconstruct (Naselaris et al. (2009); Van Gerven et al. (2010); Chen et al. (2023); Takagi & Nishimoto (2023); Ozcelik et al. (2022); Beliy et al. (2019)) images from brain signals that closely resemble the original stimulus images. In reality, the majority of visual stimuli we encounter in daily life are **continuous** and **dynamic**, hence there is a growing interest in reconstructing video from brain signals. Building on previous work that decoupled semantic and structural information from fMRI to reconstruct images (Scotti et al. (2024); Lu et al. (2023); Fang et al. (2020)), we argue that when the visual stimulus shifts from static images to dynamic videos, as shown in Figure 1, it is crucial to account for three dimensions: semantic, structural, and motion, considering the brain's processing of dynamic visual information.

Due to the inherent nature of fMRI, which relies on the slow blood oxygenation level dependent (BOLD) signal (Logothetis (2002); Kim & Ogawa (2012)), neural activity is integrated over a period exceeding 10 seconds ( 300 video frames). This integration delay poses a fundamental challenge in capturing rapid motion dynamics. Consequently, the task of **reconstructing videos from fMRI signals** becomes exceedingly challenging.

To address this challenge, Nishimoto et al. (2011) transforms the video reconstruction task into an identification task, employing the Motion-Energy model (Adelson & Bergen (1985)) and Bayesian inference to retrieve videos from a predefined video library. Subsequently, Han et al. (2019), Wen et al. (2018) and Wang et al. (2022) map brain responses to the feature spaces of deep neural network (DNN) to reconstruct video stimuli. To mitigate the scarcity of video-fMRI data, Kupershmidt et al. (2022) utilizes self-supervised learning (Kingma & Welling (2014)) to incorporate a large

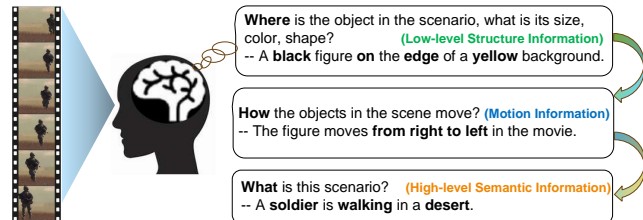

Figure 1: **The human brain's comprehension of dynamic visual scenes.** When receiving dynamic visual information, human brain gradually comprehends *low-level structural* details such as position, shape and color in the primary visual cortex, discerns *motion* information, and ultimately constructs *high-level semantic* information in the higher visual cortex, such as an overall description of the scene.

amount of unpaired video data. While these efforts have confirmed the feasibility of video reconstruction from fMRI, the results are notably deficient in explicit **semantic** information. Chen et al. (2024) utilizes contrastive learning to map fMRI to the Contrastive Language-Image Pre-training (CLIP) (Radford et al. (2021)) representation space and co-training with a video generation model, successfully reconstructing coherent videos with clear semantic information for the first time. However, this work does not consider **structure** information such as color and position, and it is uncertain whether the **motion** information in the reconstructed videos originate from the fMRI or the external video data used in training the video generation model.

In summary, current video reconstruction models face two challenges:

(1) They fail to simultaneously capture semantic, structure, and motion information within the reconstructed videos.

(2) The reliance on external video datasets and video generation models introduces ambiguity regarding whether the dynamics observed in the reconstructed videos are genuinely derived from fMRI data or are **hallucinations from video generative model**.

To address the issues, we introduce Mind-Animator, a video reconstruction model that decouples semantic, structure, and motion information from fMRI, as illustrated in Figure 3. Specifically, we map fMRI to the CLIP representation space and the Vector Quantized-Variational Autoencoder (VQ-VAE) (Van Den Oord et al. (2017)) latent space to capture semantic and structure information. We design a Transformer-based (Vaswani et al. (2017)) motion decoder to extract motion information frame by frame from fMRI through a next-frame-prediction task. Finally, the decoded semantic, structure, and motion information is fed into an inflated Stable Diffusion (Rombach et al. (2022); Wu et al. (2023)) **without any fine-tuning with video data** to generate each frame of the video.

Our contributions are summarized as follows:

**(1) Method:** We propose Mind-Animator, which enables video reconstruction by decoupling semantic, structural, and motion information from fMRI data for the first time.

To address the mismatch in timescales between fMRI and video data, we propose a Consistency Motion Generator with Sparse Causal Attention. This model decodes subtle yet significant motion patterns through a next-frame token prediction task despite the limitations imposed by the slow BOLD signal integration in fMRI.

**(2) Interpretability:** We use voxel-wise and ROI-wise visualization techniques to elucidate the interpretability of our proposed model from a neurobiological perspective.

**(3) Comprehensive evaluation:** We introduce eight evaluation metrics that comprehensively assess the reconstruction results of our model and all previous models across three dimensions—semantic, structure, and spatiotemporal consistency—on three publicly available video-fMRI datasets. This establishes our work as the first unified benchmark for subsequent researchers. We will release all data and code to facilitate future research.

## 2 RELATED WORK

### 2.1 RECONSTRUCTING VIDEOS FROM BRAIN ACTIVITIES

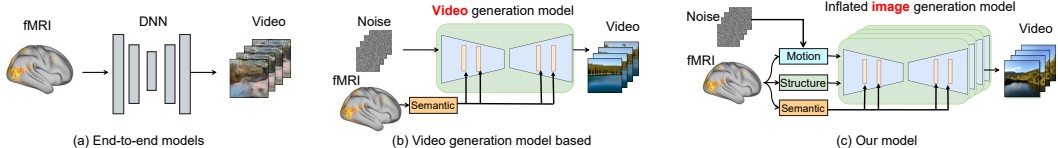

Figure 2: Overview of the video reconstruction paradigms.

The video reconstruction task involves recreating the video frames a subject was viewing based on their brain responses (e.g., fMRI). The challenge in video reconstruction lies in the significant discrepancy between the temporal resolution of fMRI (0.5 Hz) and the frame rate of the stimulus video (30 Hz), making it difficult to model the mapping between fMRI signals and video content.

To tackle this challenge, Nishimoto et al. (2011) reframes video reconstruction as an identification task, using the Motion-Energy model (Adelson & Bergen (1985)) and Bayesian inference to retrieve videos from a predefined library. With the advancement of deep learning, early works by Han et al. (2019), Wen et al. (2018) and Wang et al. (2022), as shown in Figure 2 (a), mapped brain responses to the feature spaces of deep neural networks (DNNs) for end-to-end video reconstruction. To address the scarcity of paired video-fMRI data, Kupershmidt et al. (2022) further advanced this approach by leveraging self-supervised learning to incorporate a large volume of unpaired video data. Although these studies demonstrated the feasibility of reconstructing videos from fMRI signals, the results notably **lacked explicit semantic information**. As shown in Figure 2 (b), with advancements in multimodal and generative models, Chen et al. (2024), Sun et al. (2024) used contrastive learning to map fMRI signals to the CLIP latent space for semantic decoding, followed by input into a video generation model for reconstruction. This approach produces semantically coherent and smooth videos, but it remains unclear whether the motion information in the reconstructions originates from the fMRI or from the external video data used to train the video generation model.

To address the above issues, we propose Mind-Animator. By independently decoding semantic, structural, and motion information from fMRI signals and inputting them into an inflated image generation model, we ensure that the motion in the reconstructed videos originates solely from the fMRI data.

### 2.2 DIFFUSION MODELS FOR VIDEO GENERATION

After significant progress in text-to-image (T2I) generation, diffusion models have drawn interest for text-to-video (T2V) tasks. Ho et al. (2022b) made a breakthrough by introducing 3D diffusion U-Net for video generation, followed by advancements like cascaded sampling frameworks and super-resolution methods (Ho et al. (2022a)), and the integration of temporal attention mechanisms (Singer et al. (2022); Zhou et al. (2022); He et al. (2022b)). However, due to limited paired text-video datasets and high memory demands of 3D U-Nets, alternative approaches have emerged, refining pre-trained T2I models for T2V tasks. Khachatryan et al. (2023) and Wu et al. (2023) introduced techniques like cross-frame attention and inter-frame correlation consideration to adapt T2I models for video generation.

In our work on video reconstruction from fMRI, we avoided pre-trained T2V models to prevent external video data from interfering with motion information decoding from fMRI. As shown in Figure 2 (c), we adapted an inflated T2I model to generate each frame. This ensured that the motion information in the reconstructed videos was derived solely from fMRI decoding, as the generative model had never been exposed to video data.

# 3 METHODOLOGY

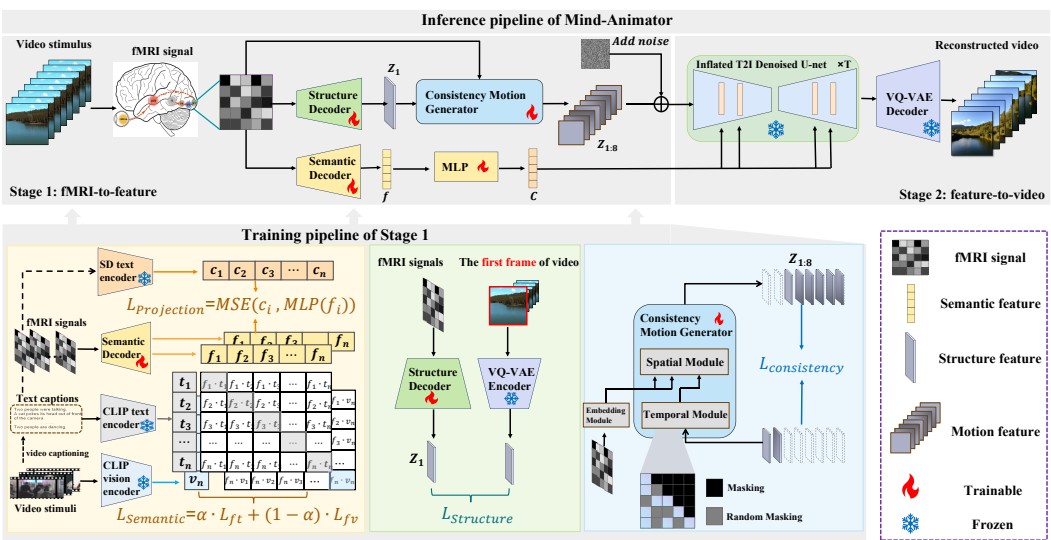

Figure 3: The overall architecture of Mind-Animator, a two-stage video reconstruction model based on fMRI. Three decoders are trained during the **fMRI-to-feature** stage to disentangle semantic, structural, and motion feature from fMRI, respectively. In the **feature-to-video** stage, the decoded information is input into an inflated Text-to-Image (T2I) model for video reconstruction.

## 3.1 PROBLEM STATEMENT

We aim to decode videos from brain activity recorded with fMRI when healthy participants watch a sequence of natural videos. Let $X$ and $Y$ denote the voxel space and pixel space, respectively. Let $\mathbf{x}_i \in \mathbb{R}^{1 \times n}$ be the fMRI signal when a video $\mathbf{v}_{i,j} \in \mathbb{R}^{1 \times 3 \times 512 \times 512}$ is presented to the participant, where $n$ is the number of fMRI voxels, $j$ is the frame ID of video $i$ and $i \in [1, N]$, $j \in [1, 8]$, with $N$ the total number of videos. Let $Z(k)$ denotes the feature space, $k \in \{semantic, structure, motion\}$. The goal of fMRI-to-feature stage is to train decoders $D(k) : X \to Z(k)$, and the goal of feature-to-video stage is to construct a generation model $G : Z(semantic) \times Z(structure) \times Z(motion) \to Y$, without introducing motion information from external video data.

## 3.2 FMRI-TO-FEATURE STAGE

**Semantic Decoder.** Due to the low signal-to-noise ratio of the fMRI signal $\mathbf{x}_i$ and the substantial dimension discrepancy with the text condition $\mathbf{c}_i \in \mathbb{R}^{1 \times 20 \times 768}$ of Stable Diffusion (SD), learning a mapping between them directly is prone to overfitting. Considering both the lower dimensionality ($\mathbf{t}_i \ or \ \mathbf{v}_i \in \mathbb{R}^{1 \times 512}$) and the robust semantic information embedded in the latent space of CLIP (Gao et al. (2020)), and given that CLIP has been shown to outperform various single-modal DNNs in explaining cortical activity (Wang et al.; Zhou et al. (2024)), we employ bidirectional InfoNCE loss to align the fMRI embedding $\mathbf{f}_i$ with the latent space of CLIP (Vit-B/32)$\subseteq \mathbb{R}^{512}$, followed by a two-layer MLP to map it to text condition $\mathbf{c}_i$. In this context, $\mathbf{f}_i = D_{Semantic}(\mathbf{x}_i)$, where $D_{Semantic}$ is a three-layer MLP,

$$L_{BiInfoNCE} = -\frac{1}{B} \sum_{i=1}^{B} \Big( log \frac{exp(s(\hat{\mathbf{z}}_i, \mathbf{z}_i)/\tau)}{\sum_{j=1}^{B} exp(s(\hat{\mathbf{z}}_i, \mathbf{z}_j)/\tau)} + log \frac{exp(s(\hat{\mathbf{z}}_i, \mathbf{z}_i)/\tau)}{\sum_{k=1}^{B} exp(s(\hat{\mathbf{z}}_k, \mathbf{z}_i)/\tau)} \Big). \quad (1)$$

where $s(\cdot, \cdot)$ is the cosine similarity, $\mathbf{z}$ and $\hat{\mathbf{z}}$ are the latent representation from two modalities, $B$ is the batch size, and $\tau$ is a learned temperature parameter. Then, given $\mathbf{f} \in \mathbb{R}^{B \times 512}$, $\mathbf{v} \in \mathbb{R}^{B \times 512}$,

and $\mathbf{t} \in \mathbb{R}^{B \times 512}$ as the respective embeddings of fMRI, video, and text, the fMRI-vision-language tri-modal loss is:

$$L_{Semantic} = \alpha \cdot L_{BiInfoNCE}(\mathbf{f}, \mathbf{t}) + (1 - \alpha) \cdot L_{BiInfoNCE}(\mathbf{f}, \mathbf{v}). \tag{2}$$

For stable training, we have also designed the following loss function to bring the fMRI and text embeddings closer together:

$$L_{Projection1} = \frac{1}{B} \sum_{i=1}^{B} \|\mathbf{f}_i - \mathbf{t}_i\|_2^2. \tag{3}$$

Subsequently, to map the fMRI embedding $\mathbf{f}_i$ to the text condition $\mathbf{c}_i$ for the purpose of conditioning generative image models, a projection loss is utilized,

$$L_{Projection2} = \frac{1}{B} \sum_{i=1}^{B} \|MLP(\mathbf{f}_i) - \mathbf{c}_i\|_2^2. \tag{4}$$

Finally, we combine the semantic and projection losses using tuned hyperparameters $\lambda_1, \lambda_2$,

$$L_{Combined} = L_{Projection1} + \lambda_1 \cdot L_{Semantic} + \lambda_2 \cdot L_{Projection2}. \tag{5}$$

**Structure Decoder.** For a short video clip, it can be assumed that the low-level feature (e.g. size, shape, and color) contained in each frame remains largely consistent with that of the first frame. Consequently, we utilize the token extracted from the first frame by VQ-VAE as structural feature and train the structural decoder (a two-layer MLP) using the standard mean squared error (MSE) loss function. Let $\Phi$ denote the encoder of VQVAE, the structure loss is defined as:

$$L_{Structure} = \frac{1}{B} \sum_{i=1}^{B} \|D_{Structure}(\mathbf{f}_i) - \Phi(\mathbf{v}_{i,1})\|_2^2. \tag{6}$$

**Consistency Motion Generator.** Drawing inspiration from natural language processing, we treat each video frame token as a word embedding and develop an L-layer Transformer-based Consistency Motion Generator (CMG) to implicitly decode dynamic information between consecutive frames.

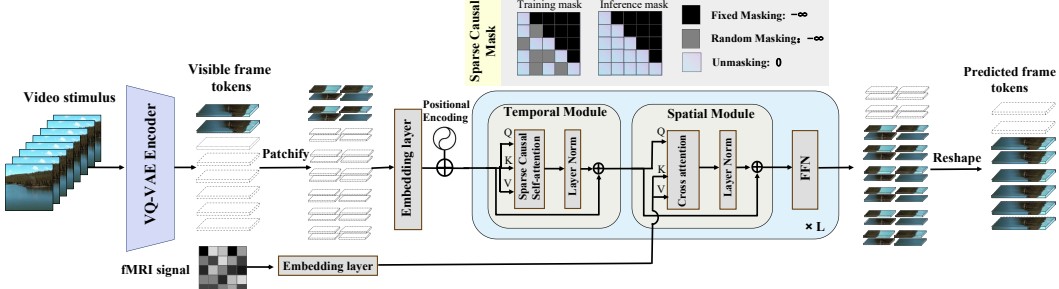

Figure 4: The architecture of CMG with Temporal Module and fMRI guided Spatial Module.

Handling raw pixels for a video clip $\mathbf{v}_i \in \mathbb{R}^{f \times 3 \times H \times W}$ in the CMG can be computationally intensive. To overcome this, we follow Stable Diffusion's approach (Rombach et al. (2022)) by projecting the video into a latent space using a pre-trained VAE tokenizer. After compressing the video clip to $\Phi(\mathbf{v}_i) \in \mathbb{R}^{f \times 3 \times \frac{H}{8} \times \frac{W}{8}}$, it is divided into patches, which are then converted into frame tokens $\Phi_{tok}(\mathbf{v}_i) \in \mathbb{R}^{P \times d_{token}}$ through an embedding layer.

In the Temporal Module, visible frame tokens $\Phi_{tok}(\mathbf{v}_i) \in \mathbb{R}^{m \times d_{token}}$ and positional encoding $\mathbf{E_{pos}} \in \mathbb{R}^{m \times d_{token}}$ are jointly input into a Sparse Causal Self-Attention layer to learn inter-frame temporal information. This attention layer incorporates a specially designed Sparse Causal Mask to ensure sparsity between frames. As illustrated in Figure 4, the mask is divided into **fixed** and **random** components. The fixed mask ensures that each frame cannot access information from subsequent frames, while the random mask maintains sparsity among visible frames, preventing the model from

taking shortcuts (Tong et al. (2022)). During inference, we eliminate the random mask. For other variants of the spatial-temporal module, please refer to Appendix D.2.

In the Spatial Module, to extract spatial information for subsequent frames from fMRI, the embedding of the visible frames $\mathbf{z}_l$ serves as the Query, while the fMRI signal $\mathbf{f}$, after passing through an embedding layer, serves as the Key and Value in the cross-attention block, as shown in Eq. (7). Following residual connections and layer normalization, $\mathbf{z}_l$ is input into the Feed Forward Network (FFN) to predict the subsequent unseen frame tokens $\Phi(\hat{\mathbf{v}}_{i,j}), j \in [m+1, n]$:

$$\mathbf{z}_l = CrossAttention(\mathbf{Q}, \mathbf{K}, \mathbf{V}), \quad l = 1, 2, \ldots, L \tag{7}$$

$$\mathbf{Q} = \mathbf{W_Q}^l \cdot \mathbf{z}_l, \quad \mathbf{K} = \mathbf{W_K}^l \cdot Emb(\mathbf{f}), \quad \mathbf{V} = \mathbf{W_V}^l \cdot Emb(\mathbf{f}),$$

$$\mathbf{z}_l = FFN(LN(\mathbf{z}_l) + \mathbf{z}_{l-1}). \quad l = 1, 2, \ldots, L \tag{8}$$

Then, the final motion consistency loss is defined as:

$$L_{Consistency} = \frac{1}{B} \sum_{i=1}^{B} \sum_{j=m+1}^{n} \|\Phi_{tok}(\hat{\mathbf{v}}_{i,j}) - \Phi_{tok}(\mathbf{v}_{i,j})\|_2^2. \tag{9}$$

### 3.3 FEATURE-TO-VIDEO STAGE

**Inflated Stable Diffusion for Video Reconstruction.** Despite the rapid development of video generation models capable of producing vivid videos from text conditions, it is crucial to emphasize that the objective of our work is to disentangle semantic, structural, and motion information from fMRI to reconstruct the stimulus video. Utilizing pre-trained video generation models could obscure **whether the motion information in the reconstructed video originates from the fMRI or external video data**.

To address this issue, we employ the network inflation (Carreira & Zisserman (2017); Khachatryan et al. (2023); Wu et al. (2023)) technique to implement an inflated Stable Diffusion, which is used to reconstruct each frame of the video without introducing additional motion information. Specifically, after the motion features $\Phi(\mathbf{v}_i) \in \mathbb{R}^{B \times f \times 3 \times \frac{H}{8} \times \frac{W}{8}}$ are decoded, they are reshaped $((B, f, 3, \frac{H}{8}, \frac{W}{8}) \rightarrow (B \cdot f, 3, \frac{H}{8}, \frac{W}{8}))$ and input into the U-Net of Stable Diffusion for reverse denoising. The result is then mapped back to pixel space through the VQ-VAE decoder and reshaped $((B \cdot f, 3, H, W) \rightarrow (B, f, 3, H, W))$ to yield the final video $\mathbf{v}_i \in \mathbb{R}^{B \times f \times 3 \times H \times W}$. In this context, $B$ denotes the batch dimension, with $B = 1$ during inference.

## 4 EXPERIMENT

### 4.1 DATASETS

In this study, we utilize three publicly available video-fMRI datasets, which encompass paired stimulus videos and their corresponding fMRI responses. As depicted in Table 1, these datasets collectively comprise brain signals recorded from multiple healthy subjects while they are viewing the videos. The video stimuli are diverse, covering animals, humans, and natural scenery. For detailed information on the datasets and preprocessing steps, please refer to Appendix B.

Table 1: Characteristics of the video-fMRI datasets used in our experiments.

| Dataset | Adopted subjects | TR | Train samples | Test samples |
|---|---|---|---|---|
| CC2017 (Wen et al. (2018)) | 3 | 2s | 4320 | 1200 |
| HCP (Marcus et al. (2011)) | 3 | 1s | 2736 | 304 |
| Algonauts2021 (Cichy et al. (2021)) | 10 | 1.75s | 900 | 100 |

### 4.2 EVALUATION METRICS

To comprehensively evaluate the performance of our model, we use the following evaluation metrics.

**Semantic-level metrics.** Following prior studies (Chen et al. (2023; 2024)), we use the N-way top-K accuracy classification test and VIFI-score as the semantic-level metrics. For the classification test, we implement two modes: image-based (2-way-I) and video-based (2-way-V). We describe this evaluation method in Algorithm 2. For the VIFI-score, we utilize a CLIP model fine-tuned on the video dataset (VIFICLIP) (Rasheed et al. (2023)) to extract features from both the ground truth and predicted videos, followed by the calculation of cosine similarity.

**Pixel-level metrics.** We employ the structural similarity index measure (SSIM), peak signal-to-noise ratio (PSNR), and hue-based Pearson correlation coefficient (Swain & Ballard (1991)) (Hue-pcc) as pixel-level metrics.

**Spatiotemporal (ST) -level metrics.** We adopt CLIP-pcc, a widely used metric in video editing research (Wu et al. (2023)), to evaluate the smoothness and consistency between consecutive video frames. This metric computes the CLIP image embeddings for each frame in the predicted videos and reports the average cosine similarity between all pairs of adjacent frames. Considering the input to the video reconstruction task contains substantial noise, there are instances where every pixel of each reconstructed video frame is either zero or noise, which would artificially inflate the CLIP score if used directly. Therefore, we calculate the CLIP score only when the VIFI-CLIP value exceeds the average level (0.6); otherwise, we assign a score of 0.

To measure the similarity of motion trajectories, we introduce End-Point Error (EPE) (Barron et al. (1994)), which calculates the Euclidean distance between the predicted and ground truth endpoints for each frame.

## 5 RESULTS

### 5.1 COMPARATIVE EXPERIMENTAL RESULTS

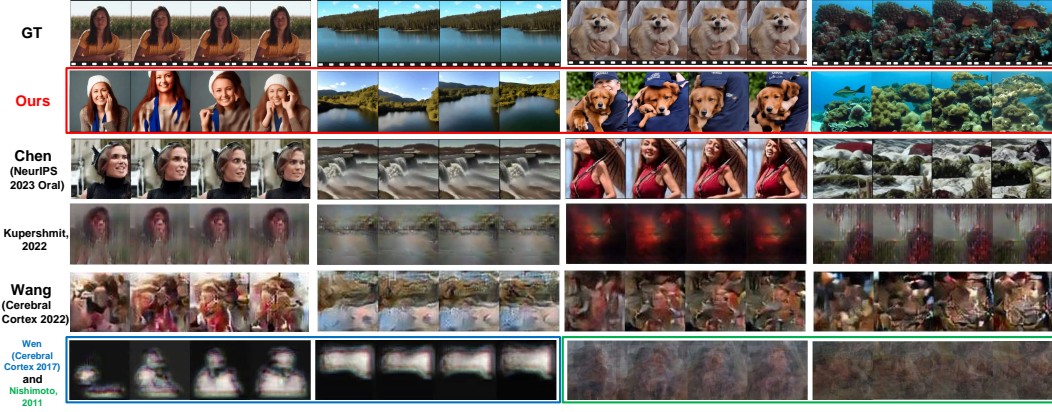

Figure 5: Reconstruction results on CC2017 dataset. Our reconstructed results are highlighted with a red box, while those of Wen and Nishimoto are delineated by blue and green boxes, respectively.

Table 2: Quantitative comparison of reconstruction results on the CC2017 dataset. All metrics are averaged across all samples and three subjects, with the best results highlighted in bold and the second-best results underlined. EV refers to the External Video Dataset. The symbol † refers to using Stable Diffusion fine-tuned on video data. 100 sample sets were constructed using the bootstrap method for hypothesis testing. Colors reflect statistical significance (Wilcoxon test for paired samples) compared to our model. $p < 0.0001$ (purple); $p < 0.01$ (pink); $p < 0.05$ (yellow); $p > 0.05$ (green).

| Models | Training data | Semantic-level ↑ | | | Pixel-level ↑ | | | ST-level | |
| --- | --- | --- | --- | --- | --- | --- | --- | --- | --- |
| | | 2-way-I | 2-way-V | VIFI | SSIM | PSNR | Hue-pcc | CLIP↑ | EPE↓ |
| Nishimoto (Nishimoto et al. (2011)) | CC2017 | 0.742 | —— | —— | 0.119 | 8.383 | 0.737 | —— | —— |
| Wen (Wen et al. (2018)) | CC2017 | 0.771 | —— | —— | 0.130 | 8.031 | 0.637 | —— | —— |
| Kupershmidt (Kupershmidt et al. (2022)) | CC2017+EV | 0.769 | 0.768 | 0.591 | 0.140 | 10.637 | 0.616 | 0.382 | —— |
| f-CVGAN (Wang et al. (2022)) | CC2017 | 0.721 | 0.777 | 0.592 | 0.108 | **11.043** | 0.583 | 0.399 | 6.344 |
| Mind-video † (Chen et al. (2024)) | CC2017+HCP | 0.797 | **0.848** | 0.593 | 0.177 | 8.868 | 0.768 | 0.409 | 6.125 |
| Ours | CC2017 | **0.805** | 0.830 | **0.608** | **0.321** | 9.220 | **0.786** | **0.425** | **5.422** |

We compare our model with all previous video reconstruction models on the aforementioned datasets. In the computation of quantitative metrics, the results of Wen et al. (2018) pertain to the first segment of the test set, whereas the results of other researchers are derived from the whole test set. Visual comparisons on CC2017 dataset are presented in Figure 5, while quantitative comparisons are detailed in Table 2, which indicates that our model achieves SOTA performance in six out of eight metrics. Specifically, our model outperforms the previous SOTA model by 83% and 13% in terms of SSIM and EPE respectively, which underscores the benefits of incorporating structural and motion information. The results in Tables 3 and 4 demonstrate that our model maintains strong performance on other datasets as well. For instance, our model outperforms Mind-video by 196%/ 21%/ 4% on the HCP and 275%/ 27%/ 5% on the Algonauts 2021 dataset across the three pixel-level metrics.

Table 3: Quantitative comparison of reconstruction results on the HCP dataset. The full table can be found in Appendix E.7.

| Models | Semantic-level ↑ | Pixel-level ↑ | | |
|---|---|---|---|---|
| | 2-way-I | SSIM | PSNR | Hue-pcc |
| Nishimoto | 0.658 | 0.321 | 11.316 | 0.645 |
| Wen | 0.702 | 0.058 | 10.197 | 0.727 |
| f-CVGAN | —— | 0.159 | **13.033** | —— |
| Mind-video | 0.779 | 0.116 | 9.275 | 0.793 |
| Ours | **0.786** | **0.344** | 11.233 | **0.829** |

Table 4: Quantitative comparison of reconstruction results on the Algonauts 2021 dataset. The full table can be found in Appendix E.8.

| Models | Semantic-level ↑ | Pixel-level ↑ | | |
|---|---|---|---|---|
| | 2-way-I | SSIM | PSNR | Hue-pcc |
| Nishimoto | 0.687 | 0.443 | 9.578 | 0.666 |
| Wen | 0.625 | 0.172 | 8.822 | 0.627 |
| Mind-video | 0.681 | 0.124 | 8.673 | 0.796 |
| Ours | **0.701** | **0.465** | **10.989** | **0.833** |

## 5.2 ABLATION STUDY

Table 5: **Ablation study** about our proposed decoders on subject 1 of CC2017 dataset. More results on subject 2 and 3 can be found in Appendix 11 and 12. 100 sample sets were constructed using the bootstrap method for hypothesis testing. Colors reflect statistical significance (Wilcoxon test for paired samples) compared to the Full Model. $p < 0.0001$ (purple); $p < 0.01$ (pink); $p < 0.05$ (yellow); $p > 0.05$ (green).

| Models | Semantic-level ↑ | | | Pixel-level ↑ | | | ST-level | |
|---|---|---|---|---|---|---|---|---|
| | 2-way-I | 2-way-V | VIFI-score | SSIM | PSNR | Hue-pcc | CLIP-pcc↑ | EPE↓ |
| w/o Semantic | 0.679 | 0.766 | 0.523 | 0.097 | 8.005 | 0.737 | 0.123 | 8.719 |
| w/o Structure | 0.789 | 0.814 | 0.555 | 0.184 | 8.712 | **0.791** | 0.260 | 7.683 |
| w/o Motion | 0.674 | 0.789 | 0.585 | 0.136 | 8.611 | 0.715 | 0.376 | 6.374 |
| Full Model | **0.812** | **0.839** | **0.604** | **0.319** | **9.116** | 0.778 | **0.413** | **5.572** |

In this subsection, we conduct a detailed ablation study to assess the effectiveness of the three decoders we proposed and evaluate the impact of various hyperparameters on video reconstruction. First, we present the results obtained using the full model. Then, based on the full model, we eliminate the semantic decoder (w/o Semantic) and the structure decoder (w/o Structure) separately, replacing their outputs with random Gaussian noise. For the consistency motion generator, we replace it with 8 simple MLPs to model each frame individually (w/o Motion). Table 5 demonstrates that the removal of any decoder results in a significant decline in the model's performance across nearly all metrics, which shows the efficacy of our proposed decoders. Notably, the Hue-pcc significantly increased after removing the structure decoder. We hypothesize that while fMRI contains structure and motion information, its low signal-to-noise ratio introduces noise that affects the generation quality of Sable Diffusion.

## 6 INTERPRETABILITY ANALYSIS

### 6.1 HAVE WE TRULY DECODED MOTION INFORMATION FROM fMRI?

Following the work of Wang et al. (2022), we conduct shuffle tests on three subjects from the CC2017 dataset to evaluate whether the CMG accurately decodes motion information from fMRI, focusing on reconstructed videos with clear semantic decoding. Specifically, for each 8-frame reconstructed video clip from each subject, we shuffle the frame order 100 times randomly and compute spatiotemporal-level metrics on the original and shuffled frames. Subsequently, we estimate the P-value by the following formula: $P = \sum_{i=1}^{100} \delta_i / 100$, where $\delta_i = 1$ if the $i$-th shuffle outperforms the reconstruction result in the original order based on the metrics; otherwise, $\delta_i = 0$. A lower P-value signifies a closer

alignment between the sequential order of the reconstructed video and the ground truth. We repeat the shuffle test 5 times under conditions with and without the CMG, as illustrated in Figure 6.

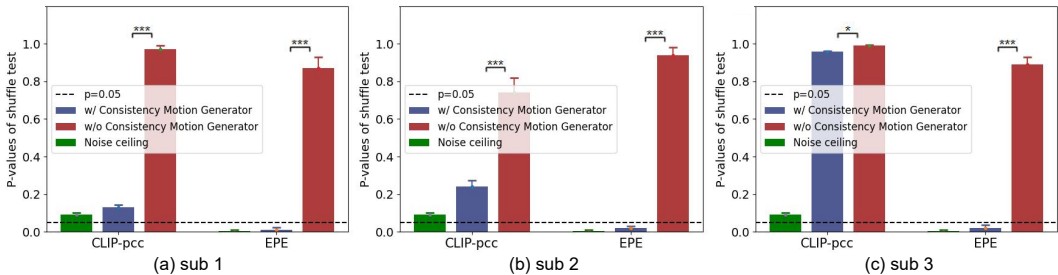

Figure 6: The results of shuffle test on the CC2017 dataset. The experiment is repeated 5 times on 3 subjects, with the mean and std presented in subplots (a), (b), and (c), respectively. Paired t-tests with Bonferroni correction are performed, with significance denoted as $p < 0.001 (***)$, $p < 0.01 (**)$, $p < 0.05 (*)$, and $p > 0.05 (NS)$ for non-significant results.

It can be observed that the P-value of EPE is significantly lower than 0.05 when CMG is applied. However, although the P-value of CLIP-pcc is significantly smaller with CMG compared to without CMG, the P-value remains significantly greater than 0.05. To explain this, we further repeated the shuffle test on the reconstruction results' noise ceiling (videos generated directly using the test set features). The results show that even for the noise ceiling, the P-value of CLIP-pcc remains significantly greater than 0.05. This indicates that: (1) we have indeed decoded motion information from fMRI, and (2) EPE is a more effective metric than CLIP-pcc for evaluating the model's ability to decode motion information.

To further validate whether the decoded motion information originates from fMRI guidance or the CMG's autoregressive training, we removed the fMRI guidance during the CMG module's training (w/o fMRI guidance) by replacing the cross-attention in the Spatial Module with self-attention, while keeping the rest of the architecture and hyperparameters unchanged. As shown in Figure 7, removing the fMRI guidance led to a significant deterioration in EPE, confirming that the proposed CMG effectively decodes motion information from fMRI. Additionally, comparing the removal of the entire CMG module (w/o Motion) with the removal of fMRI guidance (w/o fMRI guidance), we find that the latter accounts for the majority of the impact on EPE

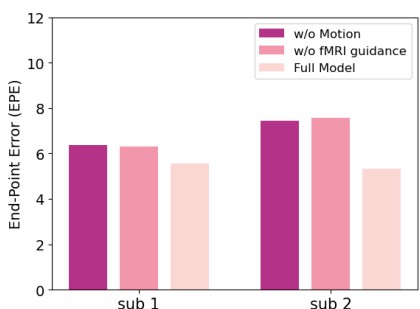

Figure 7: Ablation experiment results of fMRI guidance on the test sets of subject 1 and subject 2 from the CC2017 dataset.

(i.e., 90% of the decrease in EPE can be attributed to the absence of fMRI guidance). This further underscores the critical role of fMRI guidance in accurately decoding motion information from brain signals.

## 6.2 WHICH BRAIN REGIONS ARE RESPONSIBLE FOR DECODING DIFFERENT FEATURES, RESPECTIVELY?

To investigate voxels in which brain regions are responsible for decoding different features (semantic, structure, motion) during the fMRI-to-feature stage, we compute the **voxel-wise** importance maps in the visual cortex. Specifically, for a trained decoder, we multiply the weight matrix of the linear layers, then average the results across the feature dimension, and normalize them to estimate the importance weights for each voxel. A higher weight indicates that the voxel plays a more significant role in feature decoding. We project the importance maps of subject 1's voxels from the CC2017 dataset onto the visual cortex, as depicted in Figure 8. To obtain **ROI-wise** importance maps, we calculate the average of the importance weights of voxels contained within each Region of Interest (ROI), with the results presented in Figure 9. The results from other subjects are presented in Appendix F.

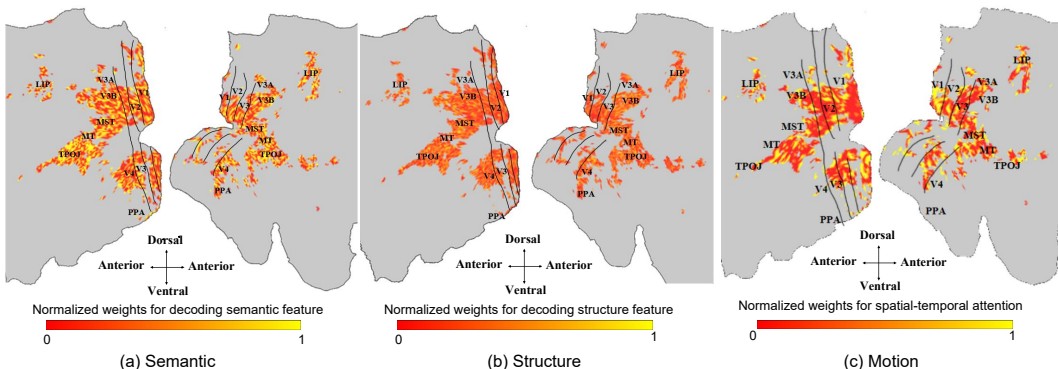

Figure 8: **Voxel-wise** importance maps projected onto the visual cortex of subject 1. The lighter the color, the greater the weight of the voxel in the interpretation of feature.

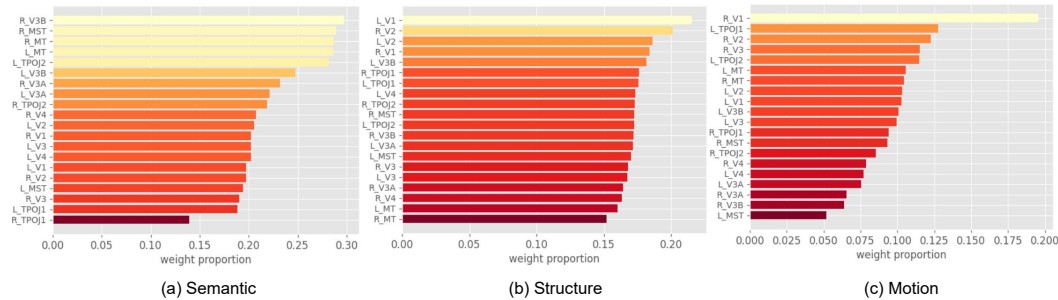

Figure 9: **ROI-wise** importance maps in the visual cortex of subject 1.

Figure 8 (a) indicates that high-level visual cortical areas (HVC, such as MT) contribute more significantly to the decoding of semantic feature, accounting for 60.5% of the total, as shown in Figure 9 (a). Figure 8 (c) and 9 (c) indicates that both LVC and HVC contribute to the decoding of motion information, with significant weight attributed to MT and TPOJ. This observation is consistent with previous work (Born & Bradley (2005)), which validates the function of MT and TPOJ in visual motion perception and processing.

We also identify the following findings in Figure 9: (1) MT shows significant activation for semantic decoding. This observation aligns with the functional segregation and interaction between the dorsal and ventral pathways during dynamic visual input processing (Ingle et al. (1982)). Specifically, the dorsal-dorsal pathway is associated with action control, whereas the ventral-dorsal pathway is involved in action understanding and recognition (Rizzolatti & Matelli (2003)). This finding aligns with the latter. (2) V1 is predominantly activated when decoding motion features, reflecting the visual system's parallel processing capability. Motion information in the dorsal pathway does not strictly follow hierarchical processing (Zeki & Shipp (1988)). As noted by Nassi et al. (Nassi & Callaway (2009)), V1 directly projects motion-related information, such as direction and speed, to MT for further processing. For detailed neurobiological explanations, please refer to the Appendix G.

## 7    CONCLUSION

We introduce a video reconstruction model (Mind-Animator) that decouples semantic, structural, and motion information from fMRI, achieving state-of-the-art performance across 3 public datasets. We mitigate the interference of external video data on motion information decoding through a rational experimental design. The results of the shuffle test demonstrate that the motion information we decoded indeed originates from fMRI, rather than being a spontaneity from generative model. Additionally, the visualization of voxel-wise and ROI-wise importance maps substantiate the neurobiological interpretability of our model.

## 8    ACKNOWLEDGEMENT

This work was supported in part by the Strategic Priority Research Program of the Chinese Academy of Sciences (XDB0930000); in part by Beijing Natural Science Foundation under Grant L243016; and in part by the National Natural Science Foundation of China under Grant 62206284 and 62020106015. Thanks to Prof. Wei Wang for the discussions and suggestions provided for our revisions in Section 6.2. We are grateful to Prof.Juan Helen Zhou and Dr.Zijiao Chen for their patient answers to our questions and for making all the results of the Mind-video test set public. We also extend our thanks to Prof.Michal Irani, Dr.Ganit Kupershmidt, and Dr.Roman Beliy for providing us with all the reconstruction results of their models on the test set. We would like to express our appreciation to Prof.Zhongming Liu and Dr.Haiguang Wen for their open-sourced high-quality video-fMRI dataset and the preprocessing procedures.

## 9    ETHICS STATEMENT

The pursuit of unraveling and emulating the brain's intricate visual processing systems has been a cornerstone endeavor for researchers in computational neuroscience and artificial intelligence. Recent advancements in neural decoding and the reconstruction of visual stimuli from brain activity have opened up numerous possibilities, fueling concerns about the potential harmful use cases of mind reading.

We argue that these concerns can be alleviated for two main reasons: (1) Mind reading requires brain activity recording devices with very high spatial resolution, and data acquisition systems like fMRI, which possess high spatial resolution, are not easily portable; (2) Although there are now several portable brain activity recording devices, achieving mind reading would require the subject to maintain intense focus and cooperate with the data collection process.

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

# Animate Your Thoughts: Decoupled Reconstruction of Dynamic Natural Vision from Slow Brain Activity

# ————Appendix————

## CONTENTS

# A RELATED WORK

## A.1 RECONSTRUCTING HUMAN VISION FROM BRAIN ACTIVITIES

### RECONSTRUCTING IMAGES FROM BRAIN ACTIVITIES

Building on Haxby et al. (2001)'s seminal work, the field of neural decoding has seen a proliferation of tasks with significant implications for guiding research. These tasks can be broadly classified into three categories: stimulus classification, identification, and reconstruction, with the latter being the most challenging and the focus of our study.

Traditional image reconstruction techniques rely on linear regression models to correlate fMRI with manually defined image features (Kay (2008); Naselaris et al. (2009); Fujiwara et al. (2013)), yielding blurry results and a heavy reliance on manual feature selection. However, the advent of deep learning has revolutionized this domain. Deep neural networks (DNNs) have become increasingly prevalent for their ability to address the scarcity of stimulus-fMRI pairs through semi-supervised learning (Chapelle et al. (2009)), as demonstrated by Beliy et al. (2019) and Gaziv et al. (2022). Yet, these models often fail to capture discernible semantic information. Chen et al. (2023) employed a pre-training and fine-tuning approach on fMRI data, leveraging methods akin to Masked Autoencoder(MAE) (He et al. (2022a)) and Latent Diffusion Models(LDM) (Rombach et al. (2022)) to improve reconstruction quality. Ozcelik et al. (2022) and Gu et al. (2022) utilized self-supervised models for feature extraction, followed by iterative optimization to refine the reconstruction process. The integration of semantic information from text, facilitated by Contrastive Language-Image Pre-Training (CLIP) (Radford et al. (2021)), has been instrumental in reconstructing complex natural images. Lin et al. (2022) and Takagi & Nishimoto (2023) demonstrated the potential of aligning fMRI with CLIP representations and mapping fMRI to text and image features for high-fidelity reconstruction.

While rapid advancements have been made in stimulus reconstruction, with some researchers achieving reconstructions from brain signals that closely approximate the original stimuli, the majority of prior work has focused on static image reconstruction. This study, however, shifts the focus to the more challenging task of video reconstruction.

### RECONSTRUCTING VIDEOS FROM BRAIN ACTIVITIES

Compared to image reconstruction, the challenge in video reconstruction lies in the significant discrepancy between the temporal resolution of fMRI (0.5Hz) and the frame rate of the stimulus video (30Hz), which presents a substantial challenge in modeling the mapping between fMRI signals and video content. To overcome the challenge, Nishimoto et al. (2011) transformed the video reconstruction task into a identification problem, employing the Motion-Energy model (Adelson & Bergen (1985)) and Bayesian inference to reconstruct videos from a predefined video library. Subsequently, Han et al. (2019) and Wen et al. (2018) mapped brain responses to the feature spaces of DNN to reconstruct down-sampled (with the frame rate reduced to 1Hz) video stimuli. Specifically, Han et al. mapped fMRI data to a VAEKingma & Welling (2014) pretrained on the ImageNet ILSVRC2012 Russakovsky et al. (2015) dataset to reconstruct a single frame, while Wen et al. mapped fMRI data to the feature space of AlexNetKrizhevsky et al. (2012) and used a deconvolutional neural networkZeiler et al. (2010) to reconstruct a single frame. The aforementioned studies have preliminarily validated the feasibility of reconstructing video frames from fMRI. Wang et al. (2022) developed an f-CVGAN that learns temporal and spatial information in fMRI through separate discriminators (Goodfellow et al. (2020)). To mitigate the scarcity of fMRI-video data, Kupershmidt et al. (2022) utilized self-supervised learning (Kingma & Welling (2014)) to incorporate a large amount of unpaired video data. These efforts have validated the feasibility of video reconstruction from fMRI, albeit with a lack of explicit semantic information in the results. Chen et al. (2024) utilized contrastive learning to map fMRI to the CLIP representation space and fine-tuned inflated Stable Diffusion (Rombach et al. (2022); Wu et al. (2023)) on a video-text dataset as a video generation model, successfully reconstructing coherent videos with clear semantic information for the first time. However, Chen did not consider structure information such as color and position, and it was uncertain whether the motion information in the reconstructed videos originated from the fMRI or the video generation model.

## A.2 DIFFUSION MODELS

Diffusion models (Wijmans & Baker (1995); Ho et al. (2020)), a class of probabilistic generative models, have increasingly rivaled or surpassed the performance of Generative Adversarial Networks (GAN) (Goodfellow et al. (2020)) in specific tasks within the field of computer vision. Diffusion models encompass a forward diffusion process and a reverse denoising process, each exhibiting

Markovian behavior. The forward process incrementally introduces Gaussian noise into the original image, culminating in a transition to standard Gaussian noise. The forward diffusion process can be represented as $q(x_t|x_{t-1}) = N(x_t; \sqrt{\alpha_t}x_{t-1}, (1-\alpha_t)I)$, where $t$ denotes the time step of each noise addition. The reverse denoising process employs the U-Net (Ronneberger et al. (2015)) architecture to accurately model the noise distribution at each timestep $t$. The image synthesis is achieved through a sequential denoising and sampling procedure, initiated from standard Gaussian noise.

In the context of image generation tasks, the conventional diffusion model executes two Markov processes in a large pixel space, resulting in substantial computational resource utilization. To address this issue, Latent Diffusion Models (LDM) (Rombach et al. (2022)) employs a VQ-VAE (Van Den Oord et al. (2017)) encoder to transform the pixel space into a low-dimensional latent space. Subsequently, the diffusion model's training and generation are performed in the latent space, with the final generated image obtained by utilizing the VQ-VAE decoder. This approach significantly reduces computational resource requirements and inference time while preserving the quality of generated images.

### A.3 DIFFUSION MODELS FOR VIDEO GENERATION

After achieving significant progress in text-to-image (T2I) generation tasks, diffusion models have piqued the interest of researchers in exploring their potential for text-to-video (T2V) generation. The pioneering work by Ho et al. (2022b) , introducing the 3D diffusion U-Net, marked significant progress in applying Diffusion Models to video generation. This was followed by further advancements by Ho et al. (2022a) , who utilized a cascaded sampling framework and super-resolution method to generate high-resolution videos. Subsequent contributions have expanded upon this work, notably with the incorporation of a temporal attention mechanism over frames by Singer et al. (2022) in Make-A-Video. Zhou et al. (2022) with MagicVideo, and He et al. (2022b) with LVDM, have integrated this mechanism into latent Diffusion Models, significantly enhancing video generation capabilities. However, due to the scarcity of paired text-video datasets and the high memory requirements for training 3D U-Nets, alternative approaches are being explored. These involve refining pre-trained T2I models to directly undertake T2V tasks. Khachatryan et al. (2023) introduced two enhancements to enable zero-shot adaptation of T2I models to T2V tasks: (1) the implementation of cross-frame attention, ensuring that the generation of each current frame in a video considers information from preceding frames; and (2) the consideration of inter-frame correlations during noise sampling, rather than random sampling for each frame independently. Wu et al. (2023) also employed cross-frame attention and achieved one-shot video editing by fine-tuning partial model parameters on individual videos.

In this work, tasked with video reconstruction from fMRI, we eschewed the use of pre-trained T2V models to mitigate the interference of external video data with the decoding of motion information from fMRI. Inspired by the cross-frame attention mechanism, we adapted a T2I model through network inflation techniques, enabling it to generate multi-frame videos. Consequently, the generative model employed in our study has never been exposed to video data, ensuring that the motion information in the reconstructed videos is solely derived from the fMRI decoding process.

## B DATA PREPROCESSING

For the stimulus videos of the three datasets described below, we segmented them into 2-second clips, down-sampled the frame rate to 4Hz (i.e. evenly extracting 8 frames), then centrally cropped each frame, and resized each to a shape of $512\times512$. Following the approach of Chen et al. (2024), we employed BLIP2 (Li et al. (2023)) to obtain textual descriptions for each video clip, with lengths not exceeding 20 words.

### B.1 VIDEO CAPTIONING WITH BLIP2

Due to the absence of open-source video captioning models at the time of experimentation, we utilize the image captioning model BLIP2 to obtain text descriptions corresponding to each video clip. Two considerations are paramount in the design of the video captioning process: (1) the length of the text descriptions should not be excessively long, and (2) the text descriptions must reflect the scene transitions within the video segments. To achieve the first objective, we employ the following prompt: 'Question: What does this image describe? Answer in 20 words or less. Answer:'. Regarding the second point, we extract 1st and 6th frames from every set of 8 frames, inputting them into BLIP2 to obtain their text descriptions. Subsequently, we calculate the CLIP similarity between each of the two text descriptions and 3rd frame. If the difference is no more than 0.05, it indicates minimal

scene change, leading to the random selection of one text description from either 1st or 6th frame to represent the video clip. Otherwise, indicating a scene transition, we concatenate the two text descriptions with ', then' to provide a comprehensive textual description of the video clip. PyTorch code for the video captioning process is depicted in Algorithm 1.

---

**Algorithm 1** PyTorch code for the video captioning process

---

```python
import torch
import clip
import random
import numpy as np
from PIL import Image
from lavis.models import load_model_and_preprocess

device = torch.device('cuda:5')
clip_model, clip_preprocess = clip.load("ViT-B/32", device=device)
model, vis_processors, _ = load_model_and_preprocess(name = "blip2_t5", model_type = "
    pretrain_flant5xxl", is_eval = True, device = device)
prompt = "Question: What does this image describe? Answer in 20 words or less. Answer:"

def Video_Captioning(Train_video_path_root):
    Train_Captions = []
    for i in tqdm(range(18)):
        for j in tqdm(range(240)):
            frames_root = Train_video_path_root + 'seg{}_{}/'.format(i + 1, j + 1)
            frame1 = Image.open(frames_root + '0000000.jpg').convert("RGB")
            frame1 = vis_processors["eval"](frame1).unsqueeze(0).to(device)
            frame2 = Image.open(frames_root + '0000056.jpg').convert("RGB")
            frame2 = vis_processors["eval"](frame2).unsqueeze(0).to(device)
            frame_mid = Image.open(frames_root + '0000024.jpg').convert("RGB")
            clip_image = clip_preprocess(frame_mid).unsqueeze(0).to(device)

            caption1 = model.generate({"image": frame1, "prompt": prompt})
            caption2 = model.generate({"image": frame2, "prompt": prompt})

            text1 = clip.tokenize(caption1).to(device)
            text2 = clip.tokenize(caption2).to(device)

            with torch.no_grad():
                image_features3 = clip_model.encode_image(clip_image)
                text_features1 = clip_model.encode_text(text1)
                text_features2 = clip_model.encode_text(text2)

            cos_sim1 = torch.cosine_similarity(image_features3, text_features1)
            cos_sim2 = torch.cosine_similarity(image_features3, text_features2)

            if abs(cos_sim1 - cos_sim2) <= 0.05:
                number = random.random()
                if number >= 0.5:
                    caption = caption1[0]
                else:
                    caption = caption2[0]
            else:
                caption = caption1[0] + ', and then ' + caption2[0]

            Train_Captions.append(caption)

    Train_Captions = np.array(Train_Captions)
    return Train_Captions
```

---

### B.2 CC2017

CC2017 (Wen et al. (2018)) dataset was first used in the work of Wen et al. (2018) This dataset include fMRI data from 3 subjects who view a variety of movie clips (23°×23°) with a central fixation cross (0.8°×0.8°). Clips are divided into 18 training movies and 5 testing movies, each eight minutes long, presented 2 and 10 times to each subject, respectively. MRI (T1 and T2-weighted) and fMRI data (2-second temporal resolution) are acquired using a 3-T system. The fMRI volumes are processed for artifact removal, motion correction (6 DOF), registered to MNI space, and projected onto cortical surfaces coregistered to a template.

To extract voxels in the activated visual areas, we calculate the correlation of the time series for each voxel's activation across 2 trials within the training set. Subsequently, we apply Fisher's z-transform to the computed correlations, average the results across 18 sessions, and identify the most significant 4500 voxels using a paired t-test to form a mask. This mask is computed separately for each of the 3 subjects on their respective training sets and then applied to both training and test set data, with the selected voxels averaged across trials. Following the work of Nishimoto et al. (2011) and Han

et al. (2019) , we utilize BOLD signals with a 4-second lag to represent the movie stimulus responses, thereby accounting for the hemodynamic response delay.

### B.3 HCP

This dataset is part of the Human Connectome Project (HCP) (Marcus et al. (2011)), encompassing BOLD (blood-oxygen-level dependent) responses from 158 subjects. For the subsequent experiments, three subjects (100610, 102816, and 104416) are randomly selected from this dataset. Data acquisition is performed using a 7T MRI scanner with a spatial resolution of 1.6 millimeters and a repetition time (TR) of 1 second. The utilized BOLD signals undergo standard HCP preprocessing procedures, which include correction for head motion and distortion, high-pass filtering, and removal of temporal artifacts via independent component analysis (ICA). The preprocessed BOLD responses are then registered to the MNI standard space.

Due to the difficulty in directly acquiring fMRI data from multiple trials, we directly utilize the parcellation of the human cerebral cortex proposed by Glasser et al. (2016) to extract voxels within the activated visual cortex. The resulting ROIs we extract include: V1, V2, V3, hV4, PPA, FFA, LO, PHC, MT, MST, and TPOJ, totaling 5820 voxels. Following the work of Nishimoto et al. (2011) and Han et al. (2019) , we utilize BOLD signals with a 4-second lag to represent the movie stimulus responses, thereby accounting for the hemodynamic response delay. Given that no prior work has conducted video reconstruction experiments on these three subjects from the HCP dataset, except for Wang et al. (2022), we randomly shuffle all video segments and allocate 90% for the training set, with the remaining 10% reserved for the test set.

### B.4 ALGONAUTS2021

This dataset is publicly released for the 2021 Algonauts Challenge (Cichy et al. (2021)). During data acquisition, 10 participants passively view 1100 silent videos of everyday events, each approximately 3 seconds in duration, presented three times. The participants' fMRI is recorded using a 3T Trio Siemens scanner with a spatial resolution of 2.5 millimeters and a repetition time (TR) of 1.75 seconds. The fMRI preprocessing involves steps such as slice-timing correction, realignment, coregistration, and normalization to the MNI space. Additionally, the fMRI data are interpolated to a TR of 2 seconds.

The dataset has been officially preprocessed, allowing us to extract brain responses from nine regions of interest (ROIs) within the visual cortex, including four primary and intermediate visual cortical areas V1, V2, V3, and V4, as well as five higher visual cortical areas: the Extrastriate Body Area (EBA), Fusiform Face Area (FFA), Superior Temporal Sulcus (STS), Lateral Occipital Cortex (LOC), and Parahippocampal Place Area (PPA). These areas selectively respond to body, face, biological motion and facial information, objects, and scene information, respectively. In the experiment, the average neural response across three stimulus repetitions is taken for brain activity. As the test set data are not yet public, we utilize the first 900 sets of data for training and the 900-1000 sets for testing.

### B.5 DATA ACQUISITION

The open-source datasets used in this paper can be accessed via the following links:

(1) CC2017: https://purr.purdue.edu/publications/2809/1

(2) HCP: https://www.humanconnectome.org/

(3) Algonauts2021: http://algonauts.csail.mit.edu/2021/index.html

## C IMPLEMENTATION DETAILS

### C.1 HYPERPARAMETER SETTINGS

For all three datasets employed in the experiments, during the training of the Semantic Decoder, we set $\alpha$ to 0.5, $\lambda_1$ to 0.01, and $\lambda_2$ to 0.5. The batch size is set to 64, and the learning rate is set to 2e-4, with training conducted 100 epochs. Given the critical role of data volume and augmentation methods in the training of contrastive learning models, and the scarcity of video-fMRI paired data, we implement specific data augmentation techniques to prevent overfitting. For fMRI data, we randomly select 20% of the voxels during each iteration, zero out 50% of their values. For image data, we randomly crop one frame from eight video frames to a size of 400x400 pixels, and then resize it to 224x224. When extracting the CLIP image features $\mathbf{v}$ for each video, we input each frame into the CLIP visual encoder and then compute the average across all frames. For text data, due to the

presence of similar video clips in the training set (derived from a complete video segment), BLIP2 often provide identical textual descriptions, which leads to overfitting. To mitigate this, we apply more aggressive augmentation techniques. For an input sentence, we perform Synonym Replacement with a 50% probability and Random Insertion, Random Swap, and Random Deletion with a 20% probability each.

During the training of the Structural Decoder, we set the batch size to 64 and the learning rate to 1e-6. To stabilize the learning process, we conduct the training process for 100 epochs with a 50-step warmup.

When training the Consistency Motion Generator, we set the patch size to 64 and the mask ratio of the Sparse Causal mask to 0.6 during the training phase, with a batch size of 64 and a learning rate of 4e-5. Similarly, for stability, we implement a 50-step warmup followed by 300 epochs of training.

Taking three subjects from the CC2017 dataset as an example, we utilize the first 4000 data points as the training set and the subsequent 320 data points as the validation set. Following this, we retrain the model on the entire training set, which comprises 4320 data points. The loss curve is depicted in Figure 10 .

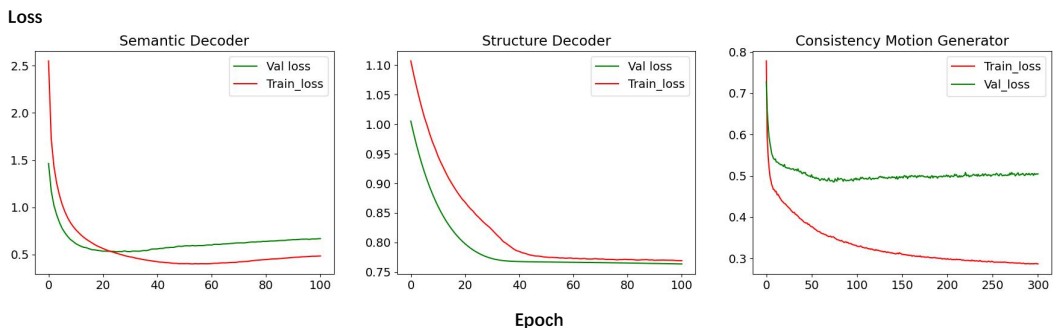

Figure 10: The training and validation loss curves for subject 1 in the CC2017 dataset.

According to Figure 10, in the inference phase, we utilize model parameters saved at the 30th, 75th, and 70th epochs for the Semantic Decoder, Structural Decoder, and Consistency Motion Generator, respectively. For the generative model, we employ inflated Stable Diffusion V1-5. Given that Stable Diffusion operates on Gaussian-distributed latent space inputs (i.e., $Z_T$) for Text-to-Image task, and the distribution of decoded structural information does not align with this, we apply 250 steps of Gaussian smoothing to it, which includes 50 steps of ddim inversion. All experiments are conducted on an A100 80G GPU, with the training phase taking 8 hours and the inference phase taking 12 hours for each dataset.

## C.2 EVALUATION METRIC IMPLEMENTATION

### SEMANTIC-LEVEL

This algorithm performs the N-trial n-way top-1 classification test. We describe our evaluation method in Algorithm 2. For the image-based scenario, we utilize a Vision Transformer (ViT) (Dosovitskiy et al. (2021)) pre-trained on ImageNet as the classifier. For video-based scenario, a pre-trained VideoMAE (Tong et al. (2022)) is employed as the classifier.

### SPATIOTEMPORAL-LEVEL

Considering the input to the video reconstruction task contains substantial noise, there are instances where every pixel of each reconstructed video frame is either zero or noise, which would artificially inflate the CLIP score if used directly. Therefore, we calculate the CLIP score only when the VIFI-CLIP value exceeds the average level; otherwise, we assign a score of 0.

### SHUFFLE TEST

Since not every test sample is accurately reconstructed, some fail to be reconstructed in terms of semantics, structure, or motion information, and some reconstruction results are entirely noise (failed samples are detailed in Appendix E.6.3). Therefore, it is meaningless to perform the shuffle test on the entire test set; we only conduct the shuffle test on the samples that are successfully reconstructed. For details, please refer to the open-source code.

---

**Algorithm 2** N-trial n-way top-1 classification test

---

1: **Input** pre-trained classifiers $\mathcal{C}_{image}(\cdot)$, $\mathcal{C}_{video}(\cdot)$, video pair (Generated Video $x$, Corresponding GT Video $\hat{x}$), mode(video-based or image-based)
2: **Output** success rate $r \in [0, 1]$
3: **if** mode='video-based' **then**
4:     **for** $N$ trials **do**
5:         $\hat{y} \leftarrow \mathcal{C}_{video}(\hat{x})$ get the ground-truth class
6:         $\{p_0, ..., p_{399}\} \leftarrow \mathcal{C}_{video}(x)$ get the output probabilities
7:         $\{p_{\hat{y}}, p_{y_1}, ..., p_{y_{n-1}}\} \leftarrow$ pick $n$-1 random classes
8:         success if $\arg\max_{y}\{p_{\hat{y}}, p_{y_1}, ..., p_{y_{n-1}}\} = \hat{y}$
9:     **end for**
10:     $r =$ number of success / $N$
11: **else**
12:     **for** 8 frames **do**
13:         **for** $N$ trials **do**
14:             $\hat{y}_i \leftarrow \mathcal{C}_{image}(\hat{x}_i)$ get the ground-truth class
15:             $\{p_0, ..., p_{999}\} \leftarrow \mathcal{C}_{image}(x_i)$ get the output probabilities
16:             $\{p_{\hat{y}_i}, p_{y_{i,1}}, ..., p_{y_{i,n-1}}\} \leftarrow$ pick $n$-1 random classes
17:             success if $\arg\max_{y_i}\{p_{\hat{y}_i}, p_{y_{i,1}}, ..., p_{y_{i,n-1}}\} = \hat{y}_i$
18:         **end for**
19:         $r_i =$ number of success / $N$
20:     **end for**
21:     $r = \sum_{i=1}^{8} r_i/8$
22: **end if**

---

## D    MODEL ARCHITECTURE

### D.1    DEFINITION OF FREQUENTLY USED SYMBOLS

The symbols frequently used in this work are defined in Table 6.

### D.2    CONSISTENCY MOTION GENERATOR

Figure 4 illustrates the Consistency Motion Generator, which is primarily composed of two modules: the Temporal Module and the Spatial Module.

The Temporal Module is tasked with learning the temporal dynamics from the visible frames. Given the severe information redundancy between video frame tokens, we specifically design a Sparse Causal mask. As shown in Figure 4 on the top, during training, the mask is divided into **fixed** and **random** components. The fixed mask ensures that each frame cannot access information from subsequent frames, while the random mask maintains sparsity among visible frames, preventing the model from taking shortcuts (Tong et al. (2022)) and accelerating training. During inference, we eliminate the random mask to allow full utilization of information from all preceding frames for predicting future frames.

Since a single fMRI frame captures information from several video frames, we design a cross attention mechanism within the Spatial Module to extract the necessary temporal and spatial information for predicting the next frame token from the fMRI data.

### D.3    TEXT-TO-IMAGE NETWORK INFLATION

To leverage pre-trained weights from large-scale image datasets, such as ImageNet, for the pre-training of massive video understanding models, Carreira & Zisserman (2017) pioneered the expansion of filters and pooling kernels in 2D ConvNets into the third dimension to create 3D filters and pooling kernels. This process transforms N×N filters used for images into N×N×N 3D filters, providing a beneficial starting point for 3D video understanding models by utilizing spatial features learned from large-scale image datasets.

In the field of generative model, several attempts have been made to extend generative image models to video models. A key technique employed in this work involves augmenting the Query, Key, and Value of the attention module, as illustrated below:

$$Q = W^Q \cdot z_{v_i}, \quad K = W^K \cdot [z_{v_0}, z_{v_{i-1}}], \quad V = W^V \cdot [z_{v_0}, z_{v_{i-1}}], \tag{10}$$

Table 6: Definition of frequently used symbols

| Symbol | Definition |
|---|---|
| $X$ | Voxel space |
| $Y$ | Pixel space |
| $Z(k)$ | Feature space, $k \in \{semantic, structure, motion\}$ |
| $D(k)$ | Feature decoder, $k \in \{semantic, structure, motion\}$ |
| $\Phi(\cdot)$ | Encoder of pretrained VQ-VAE |
| $MLP(\cdot)$ | Trainable multilayer perceptron |
| $Emb(\cdot)$ | Linear embedding layer |
| n | Number of voxels |
| $\mathbf{x}_i$ | i-th fMRI activity pattern, $\mathbf{x}_i \in \mathbb{R}^{1 \times n}$ |
| $\mathbf{v}_{i,j}$ | Frames of i-th video , $\mathbf{v}_{i,j} \in \mathbb{R}^{1 \times 3 \times 512 \times 512}$, $j \in [1, 8]$ |
| $\mathbf{c}_i$ | i-th text condition of Stable Diffusion, $\mathbf{c}_i \in \mathbb{R}^{1 \times 20 \times 768}$ |
| $\mathbf{f}_i$ | i-th fMRI embedding, $\mathbf{f}_i \in \mathbb{R}^{1 \times 512}$ |
| $\mathbf{v}_i$ | i-th video embedding in CLIP space, $\mathbf{v}_i \in \mathbb{R}^{1 \times 512}$ |
| $\mathbf{t}_i$ | i-th text embedding in CLIP space, $\mathbf{t}_i \in \mathbb{R}^{1 \times 512}$ |
| $\mathbf{W}_Q, \mathbf{W}_K, \mathbf{W}_V$ | Projection matrices of attention mechanisms |
| $\tau$ | Learned temperature parameter |
| $\alpha, \lambda_1, \lambda_2$ | Hyperparameters of semantic loss function |
| $L$ | Number of CMG blocks |
| $d_{token}$ | Dimension of frame tokens |
| $B$ | Batch size |
| $\| \cdot \|_2$ | $\mathcal{L}_2$-norm operator |
| $s(\cdot, \cdot)$ | Cosine similarity |

where $z_{v_i}$ denotes the latent of the $i$-th frame during the generation process.

## E  ADDITIONAL EXPERIMENTAL RESULTS

### E.1  THE IMPACT OF DIFFERENT SPATIAL-TEMPORAL ARCHITECTURE DESIGNS.

Due to computational resource constraints, we adopted a design in the CMG architecture that separates temporal and spatial attention modules. To capture motion information between video frames from fMRI, we designed distinct interactions between the temporal and spatial information of fMRI and video data.

For feature interaction, we primarily adopt two strategies:

**(1) Cross-attention:**  As shown in Eq. (11), we treat video representations as Queries and fMRI representations as Keys and Values, using cross-attention to extract useful spatial and temporal information from fMRI.

$$\mathbf{z}_l = CrossAttention(\mathbf{Q}, \mathbf{K}, \mathbf{V}), \quad l = 1, 2, \ldots, L$$
$$\mathbf{Q} = \mathbf{W_Q}^l \cdot \mathbf{z}_l, \quad \mathbf{K} = \mathbf{W_K}^l \cdot Emb(\mathbf{f}), \quad \mathbf{V} = \mathbf{W_V}^l \cdot Emb(\mathbf{f}). \tag{11}$$

**(2) Adaptive layer normalization:**  Inspired by the success of space-time self-attention in video modeling, we modulate the spatial and temporal information of video representations using fMRI representations, as shown in Eq. (12).

$$adaLN(\mathbf{z}_l, \mathbf{f}) = Emb(\mathbf{f})_{scale} LayerNorm(z_l) + Emb(\mathbf{f})_{shift}. \tag{12}$$

As shown in Figure 11, we designed four different architectures for the Spatial-Temporal Fusion Layer (STFL). Under the same hyperparameters used in Section C, the training/validation loss curves for these architectures are presented in Figure 12.

As shown in Figure 12, regardless of whether cross-attention or adaptive layer normalization is used, any interaction between fMRI representations and temporal information in the video leads to extreme instability in training and difficulty in convergence. We hypothesize that this is due to the significantly lower temporal resolution of fMRI compared to video stimuli, making it difficult

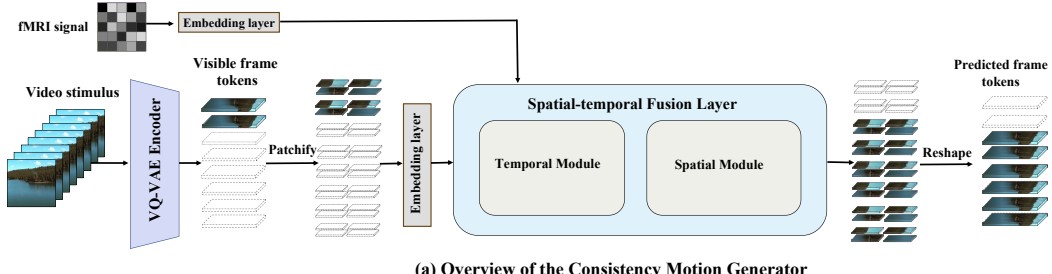

(a) Overview of the Consistency Motion Generator

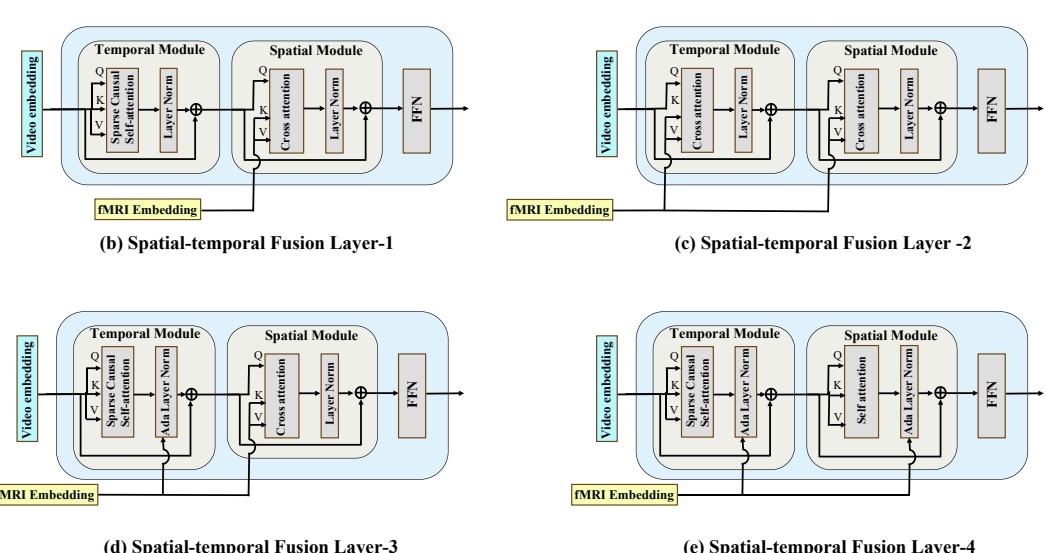

(b) Spatial-temporal Fusion Layer-1

(c) Spatial-temporal Fusion Layer -2

(d) Spatial-temporal Fusion Layer-3

(e) Spatial-temporal Fusion Layer-4

Figure 11: Variants of the Spatial-Temporal Fusion Layer.

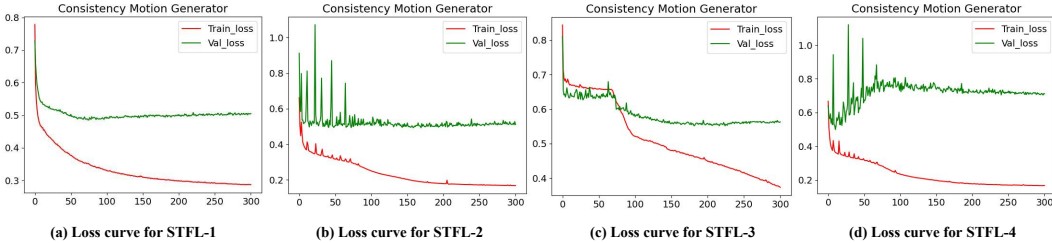

(a) Loss curve for STFL-1

(b) Loss curve for STFL-2

(c) Loss curve for STFL-3

(d) Loss curve for STFL-4

Figure 12: Loss curves for variants of the Spatial-Temporal Fusion Layer.

to **explicitly** extract useful temporal information from fMRI. Therefore, in the design of the CMG, we only allow interaction between fMRI representations and spatial information in the video. For temporal information, we use a sparse causal random mask to **implicitly** learn motion information between frames.

### E.2 A DETAILED PARAMETER SENSITIVITY ANALYSIS ON PATCH SIZE.

Table 7: Parameter sensitivity analysis on Patch size.

| Patch size | Semantic-level ↑ | | | Pixel-level ↑ | | | ST-level | |
|---|---|---|---|---|---|---|---|---|
| | 2-way-I | 2-way-V | VIFI-score | SSIM | PSNR | Hue-pcc | CLIP-pcc↑ | EPE↓ |
| 4 | 0.704 | 0.792 | 0.495 | 0.214 | **10.257** | 0.756 | 0.072 | 8.473 |
| 8 | 0.698 | 0.794 | 0.477 | 0.148 | 9.744 | 0.720 | 0.043 | 9.935 |
| 16 | 0.675 | 0.778 | 0.481 | 0.139 | 9.773 | 0.761 | 0.049 | 7.625 |
| 32 | 0.718 | 0.798 | 0.503 | 0.176 | 9.330 | 0.751 | 0.094 | 7.671 |
| 64 | **0.812** | **0.841** | **0.602** | **0.321** | 9.124 | **0.774** | **0.425** | **5.580** |

We conducted detailed experiments to investigate the sensitivity of video reconstruction to patch size. As shown in Table 7, setting a smaller patch size hinders the model's learning. This is because a smaller patch size results in a larger number of patches, forcing fine-grained patch-level interactions between fMRI and video representations. However, the low signal-to-noise ratio of fMRI does not support such fine-grained interactions, and the excessive noise may even degrade the quality of the video representations.

### E.3 A DETAILED PARAMETER SENSITIVITY ANALYSIS ON $\lambda_1$ AND $\lambda_2$.

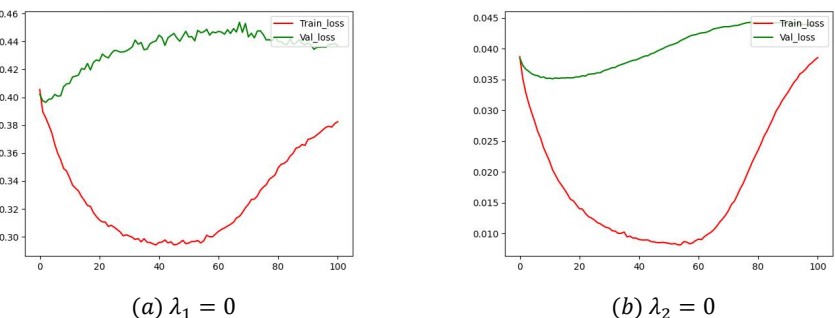

(a) $\lambda_1 = 0$      (b) $\lambda_2 = 0$

Figure 13: Loss curves for variants of $\lambda_1$ and $\lambda_2$.

Table 8: **Parameter sensitivity analysis** for variants of $\lambda_1$ and $\lambda_2$ on subject 1 of CC2017 dataset. 100 sample sets were constructed using the bootstrap method for hypothesis testing. Colors reflect statistical significance (Wilcoxon test for paired samples) compared to the Full Model. $p < 0.0001$ (purple); $p < 0.01$ (pink); $p < 0.05$ (yellow); $p > 0.05$ (green).

| Model | Semantic-level↑ | | | Pixel-level↑ | | | ST-level | |
|---|---|---|---|---|---|---|---|---|
| | 2-way-I | 2-way-V | VIFI-score | SSIM | PSNR | Hue-pcc | CLIP-pcc↑ | EPE↓ |
| $\lambda_1 = 0.01, \lambda_2 = 0.25$ | 0.765 | 0.825 | 0.581 | 0.318 | 10.199 | 0.780 | 0.396 | 6.025 |
| $\lambda_1 = 0.005, \lambda_2 = 0.5$ | 0.786 | 0.824 | 0.591 | 0.320 | 9.109 | 0.776 | 0.407 | 5.898 |
| $\lambda_1 = 0.01, \lambda_2 = 0.5$ (Ours) | **0.812** | **0.839** | **0.604** | **0.319** | 9.116 | 0.778 | **0.413** | **5.572** |

We conducted a sensitivity analysis on the selection of $\lambda_1$ and $\lambda_2$ using sub1 from the CC2017 dataset. As shown in Figure 13, when either $\lambda_1$ or $\lambda_2$ is set to 0, the semantic decider fails to converge, indicating that both the contrastive loss and projection loss play crucial roles in decoding semantic information. In Table 8, we set $\lambda_1 = 0.01$ and $\lambda_2 = 0.5$ to ensure that both loss terms are balanced during optimization. When either $\lambda_1$ or $\lambda_2$ is adjusted, breaking this balance does not affect the

structural level metrics of the reconstruction results, but it does impact the semantic and ST level metrics.

### E.4 A DETAILED PARAMETER SENSITIVITY ANALYSIS ON $\alpha$ AND MASK RATIO.

Table 9: **Parameter sensitivity analysis** on contrastive learning on subject 1 of CC2017 dataset. Colors reflect statistical significance (Wilcoxon test for paired samples) compared to Our Model. $p < 0.0001$ (purple); $p < 0.01$ (pink); $p < 0.05$ (yellow); $p > 0.05$ (green).

|  | Semantic-level↑ | | |
|---|---|---|---|
|  | 2-way-I | 2-way-V | VIFI-score |
| $\alpha$=0 (w/o fMRI-T) | 0.794 | 0.833 | 0.594 |
| $\alpha$=0.25 | 0.792 | 0.823 | 0.593 |
| $\alpha$=**0.5** (Our Model) | **0.812** | **0.839** | **0.604** |
| $\alpha$=0.75 | 0.791 | 0.832 | 0.594 |
| $\alpha$=1.0 (w/o fMRI-V) | 0.787 | 0.812 | 0.584 |

Table 10: **Parameter sensitivity analysis** on sparse causal mask ratio on subject 1 of CC2017 dataset. Colors reflect statistical significance (Wilcoxon test for paired samples) compared to Our Model. $p < 0.0001$ (purple); $p < 0.01$ (pink); $p < 0.05$ (yellow); $p > 0.05$ (green).

| Models | Pixel-level ↑ | | | ST-level | |
|---|---|---|---|---|---|
|  | SSIM | PSNR | Hue-pcc | CLIP-pcc↑ | EPE↓ |
| Ratio=0 | 0.297 | 9.037 | 0.758 | 0.397 | 5.896 |
| Ratio=0.2 | 0.276 | 8.847 | 0.767 | 0.382 | **5.311** |
| Ratio=0.4 | 0.285 | 9.045 | 0.768 | 0.404 | 5.375 |
| Ratio=0.6 (Our Model) | **0.319** | **9.116** | **0.778** | **0.413** | 5.572 |
| Ratio=0.8 | 0.296 | 9.057 | 0.767 | 0.409 | 5.733 |

During the training of Semantic Decoder, we control the weighting of the contrastive learning loss between fMRI-text ($L_{BiInfoNCE}(f, t)$) and fMRI-video ($L_{BiInfoNCE}(f, v)$) through the hyperparameter $\alpha$. We set different values for $\alpha$ (0, 0.25, 0.5, 0.75, 1.0), where $\alpha$=0 signifies the exclusion of $L_{BiInfoNCE}(f, t)$, and $\alpha$=1 signifies the exclusion of $L_{BiInfoNCE}(f, v)$. Table 9 indicates that despite achieving optimal results for the three semantic-level metrics when $\alpha$ is set to 0.5, variations in $\alpha$ do not significantly affect the results, except when $\alpha$ is 1, suggesting that the contrastive learning loss of fMRI-video predominates. During the training of CMG, we set multiple values for the mask ratio (0, 0.2, 0.4, 0.6, 0.8) and calculate the results on pixel-level and ST-level metrics, as shown in Table 10. The results indicate that setting a moderate mask ratio (0.6) can prevent the model from taking shortcuts during training and effectively capture the temporal features between frames.

### E.5 SUPPLEMENTARY ABLATION STUDIES ON SUBJECTS 2 AND 3 OF THE CC2017 DATASET

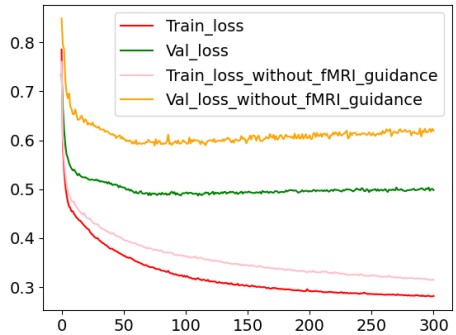

Figure 14: Loss curves for CMG with or without fMRI guidance.

We also analyze the impact of different decoders on video reconstruction performance on subjects 2 and 3 of the CC2017 dataset, as shown in Tables 11 and 12. It is evident from the aforementioned tables that the removal of semantic decoder leads to a significant decline in all metrics, whereas the removal of the other two decoders does not significantly affect the semantic metrics, thereby highlighting the crucial role of the semantic decoder in video reconstruction.

Table 11: **Ablation study** on subject 2 of CC2017 dataset. 100 sample sets were constructed using the bootstrap method for hypothesis testing. Colors reflect statistical significance (Wilcoxon test for paired samples) compared to the Full Model. $p < 0.0001$ (purple); $p < 0.01$ (pink); $p < 0.05$ (yellow); $p > 0.05$ (green).

| Model | Semantic-level↑ | | | Pixel-level↑ | | | ST-level | |
|---|---|---|---|---|---|---|---|---|
| | 2-way-I | 2-way-V | VIFI-score | SSIM | PSNR | Hue-pcc | CLIP-pcc↑ | EPE↓ |
| w/o Semantic | 0.675 | 0.769 | 0.519 | 0.081 | 7.944 | 0.739 | 0.131 | 8.457 |
| w/o Structure | 0.806 | 0.826 | 0.566 | 0.167 | 8.700 | **0.801** | 0.302 | 8.011 |
| w/o Motion | 0.810 | **0.829** | 0.523 | 0.265 | 9.080 | 0.780 | 0.326 | 7.431 |
| **Full Model** | **0.811** | 0.827 | **0.609** | **0.292** | **9.250** | 0.790 | **0.423** | **5.329** |

Table 12: **Ablation study** on subject 3 of CC2017 dataset. 100 sample sets were constructed using the bootstrap method for hypothesis testing. Colors reflect statistical significance (Wilcoxon test for paired samples) compared to the Full Model. $p < 0.0001$ (purple); $p < 0.01$ (pink); $p < 0.05$ (yellow); $p > 0.05$ (green).

| Model | Semantic-level↑ | | | Pixel-level↑ | | | ST-level | |
|---|---|---|---|---|---|---|---|---|
| | 2-way-I | 2-way-V | VIFI-score | SSIM | PSNR | Hue-pcc | CLIP-pcc↑ | EPE↓ |
| w/o Semantic | 0.673 | 0.778 | 0.523 | 0.084 | 8.109 | 0.738 | 0.136 | 7.891 |
| w/o Structure | **0.810** | 0.831 | 0.566 | 0.186 | 8.619 | **0.794** | 0.299 | 7.415 |
| w/o Motion | 0.808 | 0.826 | 0.583 | 0.272 | 8.953 | 0.779 | 0.366 | 6.588 |
| **Full Model** | 0.792 | **0.823** | **0.607** | **0.348** | **9.287** | 0.791 | **0.419** | **5.356** |

Table 13: Retrieval accuracy (%) on CC2017 dataset. For the 'small test set', the chance-level accuracies for top-10 and top-100 accuracy are 0.83% and 8.3%, respectively. For the 'large test set', the chance-level accuracies for top-10 and top-100 accuracy are 0.24% and 2.4%, respectively. Five random seeds were used during the training of the semantic decoder, and the results were tested for statistical significance. ∗ denotes our performance is significantly better than the compared method (Wilcoxon test, p<0.05).

| Dataset | | CC2017 | | | | | | | |
|---|---|---|---|---|---|---|---|---|---|
| | | Subjet1 | | Subjet2 | | Subjet3 | | Average | |
| Model | Test set | top-10 | top-100 | top-10 | top-100 | top-10 | top-100 | top-10 | top-100 |
| Wen (Wen et al. (2018)) | Small | $2.17_*$ | $19.50_*$ | $3.33_*$ | $19.17_*$ | —— | —— | $2.75_*$ | $19.33_*$ |
| Kupershmidt (Kupershmidt et al. (2022)) | Small | $1.09_*$ | $8.57_*$ | $0.92_*$ | $8.24_*$ | $0.84_*$ | $8.24_*$ | $0.95_*$ | $8.35_*$ |
| Mind-video (Chen et al. (2024)) | Small | **$3.22_*$** | $19.08_*$ | $2.75_*$ | $16.83_*$ | $3.58_*$ | $22.08_*$ | $3.18_*$ | $19.33_*$ |
| Ours | Small | 3.08 | **22.58** | **4.75** | **26.90** | **4.50** | **24.67** | **4.11** | **24.72** |
| Wen (Wen et al. (2018)) | Large | $1.41_*$ | $11.58_*$ | $2.08_*$ | $9.58_*$ | —— | —— | $1.75_*$ | $10.58_*$ |
| Kupershmidt (Kupershmidt et al. (2022)) | Large | $0.17_*$ | $2.94_*$ | $0.17_*$ | $2.77_*$ | $0.25_*$ | $2.18_*$ | $0.19_*$ | $2.63_*$ |
| Mind-video (Chen et al. (2024)) | Large | $1.75_*$ | $7.17_*$ | $0.83_*$ | $5.17_*$ | $1.25_*$ | $9.00_*$ | $1.28_*$ | $7.11_*$ |
| Ours | Large | **2.17** | **12.50** | **2.25** | **17.00** | **2.75** | **16.42** | **2.39** | **15.31** |

### E.6 FURTHER RESULTS ON THE CC2017 DATASET

### E.6.1 THE FINE-GRAINED RETRIEVAL EXPERIMENTAL RESULTS.

In addition to the reconstruction task, we evaluate the retrieval task on the CC2017 dataset. We use top-10 accuracy and top-100 accuracy as evaluation metrics. To assess the model's generalization capability, we perform retrieval not only on the CC2017 test set with 1,200 samples ('Small') but also on an extended stimulus set. Specifically, we augment the collection with 3,040 video clips from the HCP dataset, resulting in a total of 4,240 samples ('Large').

As shown in Table 13, our model achieves superior performance across all three subjects in the CC2017 dataset. Compared to Mind-Video, our model exhibits a smaller performance drop when the stimulus set is expanded to the 'Large' scale, demonstrating its generalization capability. Notably, Wen's results surpass those of Kupershmidt largely and are comparable to Mind-Video. This can be attributed to their approach of reconstructing a single frame from each video segment, simplifying the video reconstruction task into an image reconstruction task, thereby achieving superior performance on the retrieval metrics.

### E.6.2 COMPREHENSIVE QUANTITATIVE COMPARISON RESULTS ON THE CC2017 DATASET.

Table 14: Quantitative comparison of reconstruction results across three subjects from the CC2017 dataset. For the 2-way-I and 2-way-V metrics, 100 repetitions were conducted, while other metrics were evaluated using 100 bootstrap trials. All metrics are averaged over the entire test set. The best results are highlighted in bold, and the second-best are underlined. Colors indicate statistical significance (Wilcoxon test for paired samples) compared to our model. $p < 0.0001$ (purple); $p < 0.01$ (pink); $p < 0.05$ (yellow); $p > 0.05$ (green).

| Sub ID | Models | Semantic-level ↑ | | | Pixel-level ↑ | | | ST-level | |
|---|---|---|---|---|---|---|---|---|---|
| | | 2-way-I | 2-way-V | VIFI-score | SSIM | PSNR | Hue-pcc | CLIP-pcc↑ | EPE↓ |
| sub 01 | Nishimoto (Nishimoto et al. (2011)) | 0.727 | —— | —— | 0.116 | 8.012 | 0.753 | —— | —— |
| | Wen (Wen et al. (2018)) | 0.758 | —— | —— | 0.114 | 7.646 | 0.647 | —— | —— |
| | Kupershmidt (Kupershmidt et al. (2022)) | 0.764 | 0.771 | 0.585 | 0.135 | 8.761 | 0.606 | 0.386 | —— |
| | f-CVGAN (Wang et al. (2022)) | 0.713 | 0.773 | 0.596 | 0.118 | **11.432** | 0.589 | 0.402 | 6.348 |
| | Mind-video (Chen et al. (2024)) | 0.792 | **0.853** | 0.587 | 0.171 | 8.662 | 0.760 | 0.408 | 6.119 |
| | Ours | **0.812** | 0.841 | **0.602** | **0.321** | 9.124 | **0.774** | **0.425** | **5.580** |
| sub 02 | Nishimoto (Nishimoto et al. (2011)) | 0.787 | —— | —— | 0.112 | 8.592 | 0.713 | —— | —— |
| | Wen (Wen et al. (2018)) | 0.783 | —— | —— | 0.145 | 8.415 | 0.626 | —— | —— |
| | Kupershmidt (Kupershmidt et al. (2022)) | 0.776 | 0.766 | 0.591 | 0.157 | **11.914** | 0.601 | 0.382 | —— |
| | f-CVGAN (Wang et al. (2022)) | 0.727 | 0.779 | 0.596 | 0.107 | 10.940 | 0.589 | 0.404 | 6.277 |
| | Mind-video (Chen et al. (2024)) | 0.789 | **0.842** | 0.595 | 0.172 | 8.929 | 0.773 | 0.409 | 6.062 |
| | Ours | **0.811** | 0.827 | **0.615** | **0.292** | 9.250 | **0.791** | **0.429** | **5.329** |
| sub 03 | Nishimoto (Nishimoto et al. (2011)) | 0.712 | —— | —— | 0.128 | 8.546 | 0.746 | —— | —— |
| | Wen (Wen et al. (2018)) | —— | —— | —— | —— | —— | —— | —— | —— |
| | Kupershmidt (Kupershmidt et al. (2022)) | 0.767 | 0.766 | 0.597 | 0.128 | **11.237** | 0.641 | 0.377 | —— |
| | f-CVGAN (Wang et al. (2022)) | 0.722 | 0.778 | 0.584 | 0.098 | 10.758 | 0.572 | 0.392 | 6.408 |
| | Mind-video (Chen et al. (2024)) | **0.811** | **0.848** | 0.597 | 0.187 | 9.013 | 0.771 | 0.410 | 6.193 |
| | Ours | 0.792 | 0.823 | **0.607** | **0.349** | 9.287 | **0.794** | **0.421** | **5.356** |

The individual quantitative comparison results on the CC2017 dataset for the three subjects are displayed in Table 14.

### E.6.3 MORE RECONSTRUCTION RESULTS ON MULTIPLE SUBJECTS

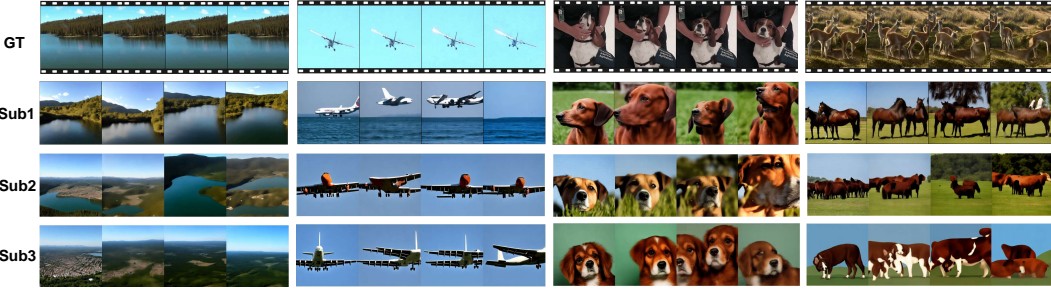

Figure 15: The reconstruction results on three subjects from the CC2017 dataset.

Due to the anatomical and functional connectivity differences among subjects, even when presented with the same stimulus video, different brain signals are elicited. Consequently, we train the Mind-Animator on three subjects from the CC2017 dataset separately, with the reconstruction results shown in Figure 15. It can be observed that, despite training and applying the model directly to the three subjects without additional modifications, the reconstruction outcomes are largely consistent, which substantiates the effectiveness of our model.

However, as it is challenging to obtain a large volume of brain signals from the same subject, the direct extension of a model trained on one subject to others, or the direct training using data from multiple subjects, represents a future research direction that requires further investigation.

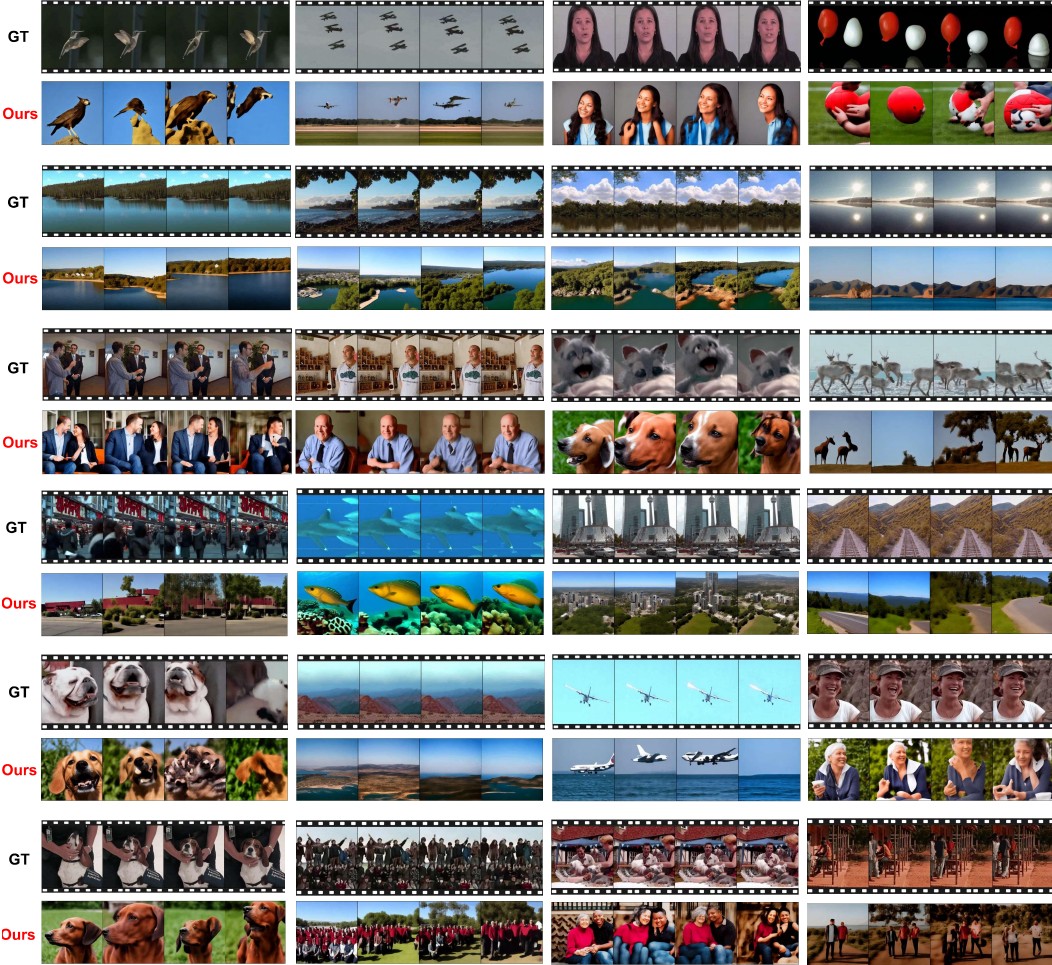

Figure 16: More reconstruction results on the CC2017 dataset.

In Figure 16, we present additional reconstruction results from the CC2017 dataset, demonstrating that our model does not overfit despite the limited data volume. It is capable of decoding a rich array of video clips, such as two people conversing, an airplane in flight, and a dog turning its head, among others. These video clips encompass a wide range of natural scenarios found in everyday life, including human activities, animal behaviors, and natural landscapes.

Additionally, to provide a comprehensive and objective assessment of our model, we also include some reconstruction failure cases in Figure 17. The primary reasons for these failures are twofold. Firstly, the inherent data acquisition paradigm leads to abrupt transitions in content at the junction of video clips, which are evenly segmented from complete videos watched by the subjects during data collection. As illustrated in the first row of Figure 17, the model often struggles to recognize such abrupt changes, resulting in the reconstruction of only the scene prior to the transition. Secondly, errors in feature decoding occur, as shown in the second row of Figure 17. Although the model

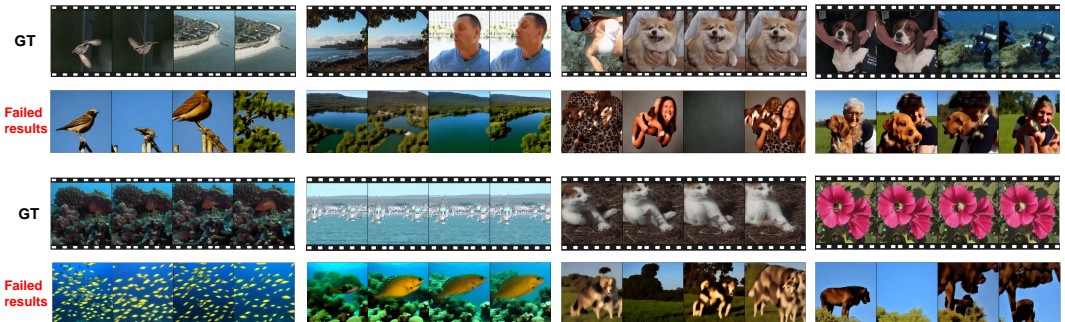

Figure 17: Reconstruction failure cases.

successfully decodes an ocean scene, structural decoding errors led to the appearance of numerous fish in the reconstructed frames.

### E.7 FURTHER RESULTS ON THE HCP DATASET

To validate the broad effectiveness of our proposed model across different datasets, we also conduct experiments on the HCP dataset. The selection of three subjects (100610, 102816, and 104416) aligns with that of Wang et al. (2022), and the parameter settings for Mind-Animator during both training and testing are identical to those used on the CC2017 dataset. We replicate Nishimoto et al. (2011) 's model on HCP, utilizing video clips from the training sets of both CC2017 and HCP as video priors. Wang only reports SSIM and PSNR metrics on the HCP dataset. Visual comparisons are presented in Figure 18, while quantitative comparisons are detailed in Table 15. The results indicate that our model still performs well across multiple subjects in the HCP dataset.

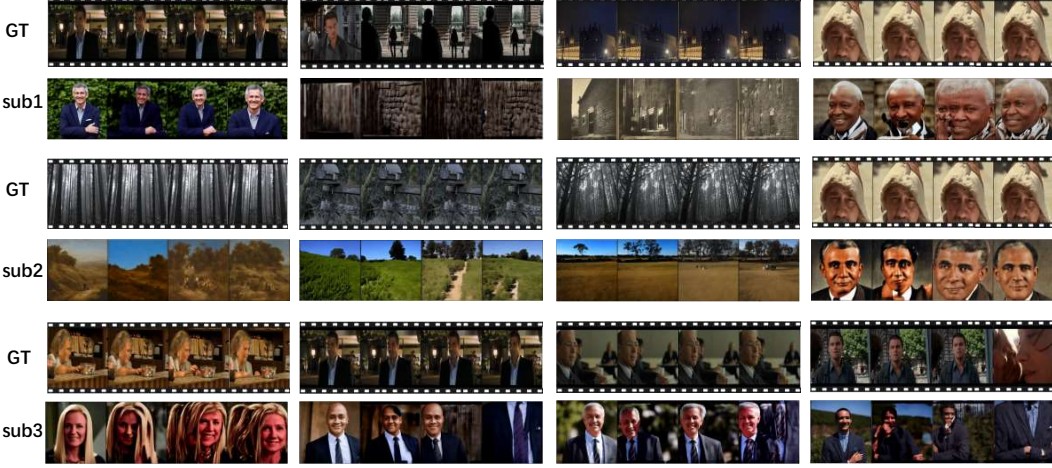

Figure 18: The reconstruction results on three subjects from the HCP dataset.

### E.8 FURTHER RESULTS ON THE ALGONAUTS2021 DATASET

We conduct experiments on the Algonauts2021 competition dataset with 10 subjects, employing the same parameter settings for Mind-Animator during both training and testing as those used on the CC2017 dataset. Since no video reconstruction results has previously been published on this dataset, we replicate Nishimoto et al.'s model as a baseline and utilize video clips from the training sets of both CC2017 and HCP as video priors. Visual comparisons and quantitative assessments across various metrics are depicted in Figure 19 and detailed in Table 16, respectively. Although Table 16 indicates that our model outperforms the earlier baseline in nearly all metrics across the 10 subjects, the reconstructed results presented in Figure 19 are not entirely satisfactory, with some video frames semantically misaligned with the stimulus video. We attribute this to the scarcity of training data,

Table 15: Quantitative comparison of reconstruction results across three subjects from the HCP dataset. For the 2-way-I and 2-way-V metrics, 100 repetitions were conducted, while other metrics were evaluated using 100 bootstrap trials. All metrics are averaged over the entire test set. The best results are highlighted in bold, and the second-best are underlined. Colors indicate statistical significance (Wilcoxon test for paired samples) compared to our model. $p < 0.0001$ (purple); $p < 0.01$ (pink); $p < 0.05$ (yellow); $p > 0.05$ (green).

| Sub ID | Models | Semantic-level ↑ | | | Pixel-level ↑ | | | ST-level | |
|---|---|---|---|---|---|---|---|---|---|
| | | 2-way-I | 2-way-V | VIFI-score | SSIM | PSNR | Hue-pcc | CLIP-pcc↑ | EPE↓ |
| | Nishimoto (Nishimoto et al. (2011)) | 0.658 | —— | —— | 0.307 | 11.711 | 0.649 | —— | —— |
| | Wen (Wen et al. (2018)) | 0.728 | —— | —— | 0.052 | 10.805 | 0.751 | —— | —— |
| sub 01 | f-CVGAN (Wang et al. (2022)) | —— | —— | —— | 0.154 | **13.200** | —— | —— | —— |
| | Mind-video (Chen et al. (2024)) | 0.798 | 0.752 | 0.605 | 0.123 | 9.302 | 0.774 | **0.486** | 12.746 |
| | Ours | **0.819** | **0.783** | **0.613** | **0.325** | 10.757 | **0.820** | 0.476 | **7.825** |
| | Nishimoto (Nishimoto et al. (2011)) | 0.661 | —— | —— | 0.338 | 11.249 | 0.643 | —— | —— |
| | Wen (Wen et al. (2018)) | 0.688 | —— | —— | 0.055 | 10.475 | 0.720 | —— | —— |
| sub 02 | f-CVGAN (Wang et al. (2022)) | —— | —— | —— | 0.178 | **13.700** | —— | —— | —— |
| | Mind-video (Chen et al. (2024)) | **0.761** | **0.777** | **0.611** | 0.115 | 9.414 | 0.804 | 0.483 | 7.358 |
| | Ours | 0.756 | 0.759 | 0.609 | **0.371** | 11.894 | **0.834** | **0.485** | **6.624** |
| | Nishimoto (Nishimoto et al. (2011)) | 0.657 | —— | —— | 0.319 | 10.988 | 0.645 | —— | —— |
| | Wen (Wen et al. (2018)) | 0.691 | —— | —— | 0.067 | 9.312 | 0.710 | —— | —— |
| sub 03 | f-CVGAN (Wang et al. (2022)) | —— | —— | —— | 0.147 | **12.200** | —— | —— | —— |
| | Mind-video (Chen et al. (2024)) | 0.779 | 0.778 | 0.612 | 0.118 | 9.109 | 0.803 | 0.529 | 7.767 |
| | Ours | **0.781** | **0.793** | **0.634** | **0.336** | 11.018 | **0.834** | **0.573** | **6.792** |

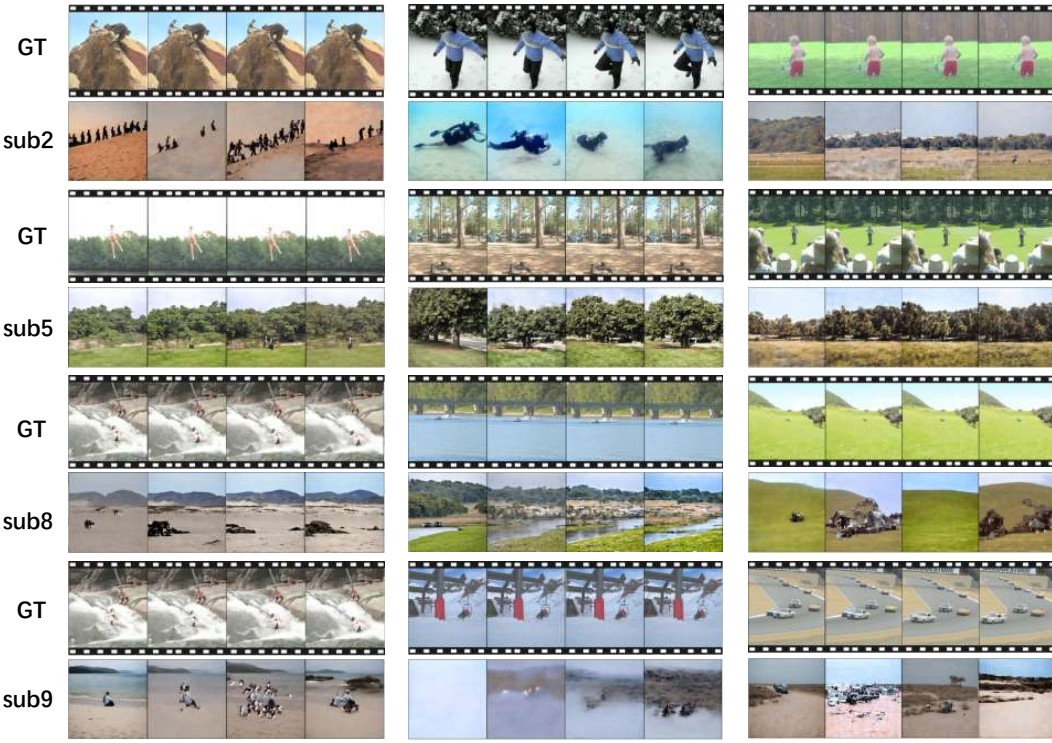

Figure 19: The reconstruction results on four subjects from the Algonauts2021 dataset.

which renders the video reconstruction task on this dataset challenging, given that the data volume per subject is approximately one-fifth of that in the CC2017 dataset.

Based on the experimental results across multiple datasets, we can draw the following conclusions: (1) The volume of training data from a single subject significantly influences the performance of current video reconstruction models, with greater sample size and data diversity leading to better reconstruction performance. (2) There is an urgent need to develop a new model using incremental

Table 16: Quantitative comparison of reconstruction results across ten subjects from the Algonauts2021 dataset. For the 2-way-I and 2-way-V metrics, 100 repetitions were conducted, while other metrics were evaluated using 100 bootstrap trials. All metrics are averaged over the entire test set. The best results are highlighted in bold, and the second-best are underlined. Colors indicate statistical significance (Wilcoxon test for paired samples) compared to our model. $p < 0.0001$ (purple); $p < 0.01$ (pink); $p < 0.05$ (yellow); $p > 0.05$ (green).

| Sub ID | Models | Semantic-level ↑ | | | Pixel-level ↑ | | | ST-level | |
|---|---|---|---|---|---|---|---|---|---|
| | | 2-way-I | 2-way-V | VIFI-score | SSIM | PSNR | Hue-pcc | CLIP-pcc↑ | EPE↓ |
| sub 01 | Nishimoto (Nishimoto et al. (2011)) | 0.688 | —— | —— | **0.446** | 9.626 | 0.672 | —— | —— |
| | Wen (Wen et al. (2018)) | 0.653 | —— | —— | 0.147 | 9.802 | 0.653 | —— | —— |
| | Mind-video (Chen et al. (2024)) | 0.702 | 0.761 | 0.568 | 0.135 | 8.642 | 0.794 | 0.277 | 8.368 |
| | Ours | **0.722** | **0.790** | **0.599** | 0.401 | **10.088** | **0.824** | **0.439** | **4.420** |
| sub 02 | Nishimoto (Nishimoto et al. (2011)) | 0.682 | —— | —— | 0.443 | 9.553 | 0.676 | —— | —— |
| | Wen (Wen et al. (2018)) | 0.626 | —— | —— | 0.231 | 8.456 | 0.677 | —— | —— |
| | Mind-video (Chen et al. (2024)) | 0.698 | **0.769** | 0.573 | 0.132 | 9.004 | 0.773 | 0.265 | 7.458 |
| | Ours | **0.734** | 0.765 | **0.596** | **0.465** | **10.932** | **0.796** | **0.425** | **3.806** |
| sub 03 | Nishimoto (Nishimoto et al. (2011)) | 0.679 | —— | —— | 0.441 | 9.576 | 0.682 | —— | —— |
| | Wen (Wen et al. (2018)) | 0.647 | —— | —— | 0.172 | 8.973 | 0.611 | —— | —— |
| | Mind-video (Chen et al. (2024)) | **0.701** | 0.729 | 0.564 | 0.117 | 8.796 | 0.806 | 0.271 | 7.659 |
| | Ours | 0.679 | **0.794** | **0.591** | **0.466** | **11.089** | **0.863** | **0.397** | **3.406** |
| sub 04 | Nishimoto (Nishimoto et al. (2011)) | **0.702** | —— | —— | 0.446 | 9.537 | 0.665 | —— | —— |
| | Wen (Wen et al. (2018)) | 0.592 | —— | —— | 0.087 | 8.473 | 0.534 | —— | —— |
| | Mind-video (Chen et al. (2024)) | 0.665 | 0.785 | 0.556 | 0.126 | 8.439 | 0.811 | 0.254 | 8.011 |
| | Ours | 0.673 | **0.810** | **0.587** | **0.479** | **11.410** | **0.848** | **0.381** | **3.089** |
| sub 05 | Nishimoto (Nishimoto et al. (2011)) | 0.676 | —— | —— | 0.442 | 9.498 | 0.650 | —— | —— |
| | Wen (Wen et al. (2018)) | 0.651 | —— | —— | 0.136 | 7.599 | 0.589 | —— | —— |
| | Mind-video (Chen et al. (2024)) | 0.664 | 0.757 | 0.529 | 0.140 | 8.597 | 0.792 | 0.263 | 8.124 |
| | Ours | **0.689** | **0.810** | **0.592** | **0.458** | **10.814** | **0.807** | **0.406** | **3.237** |
| sub 06 | Nishimoto (Nishimoto et al. (2011)) | 0.694 | —— | —— | 0.444 | 9.526 | 0.665 | —— | —— |
| | Wen (Wen et al. (2018)) | 0.642 | —— | —— | 0.131 | 9.675 | 0.690 | —— | —— |
| | Mind-video (Chen et al. (2024)) | 0.690 | 0.751 | 0.549 | 0.137 | 9.011 | 0.795 | 0.266 | 7.431 |
| | Ours | **0.709** | **0.783** | **0.597** | **0.489** | **11.337** | **0.834** | **0.446** | **3.399** |
| sub 07 | Nishimoto (Nishimoto et al. (2011)) | 0.674 | —— | —— | 0.446 | 9.630 | 0.672 | —— | —— |
| | Wen (Wen et al. (2018)) | 0.628 | —— | —— | 0.215 | 8.578 | 0.648 | —— | —— |
| | Mind-video (Chen et al. (2024)) | **0.687** | 0.721 | 0.574 | 0.109 | 8.409 | 0.783 | 0.209 | 7.652 |
| | Ours | 0.681 | **0.802** | **0.578** | **0.458** | **10.889** | **0.857** | **0.329** | **3.845** |
| sub 08 | Nishimoto (Nishimoto et al. (2011)) | 0.696 | —— | —— | 0.444 | 9.664 | 0.662 | —— | —— |
| | Wen (Wen et al. (2018)) | 0.596 | —— | —— | 0.205 | 9.431 | 0.610 | —— | —— |
| | Mind-video (Chen et al. (2024)) | 0.658 | 0.764 | 0.590 | 0.114 | 8.251 | 0.817 | 0.204 | 6.597 |
| | Ours | **0.709** | **0.802** | **0.592** | **0.467** | **10.893** | **0.820** | **0.376** | **3.757** |
| sub 09 | Nishimoto (Nishimoto et al. (2011)) | 0.673 | —— | —— | 0.445 | 9.573 | 0.661 | —— | —— |
| | Wen (Wen et al. (2018)) | 0.574 | —— | —— | 0.135 | 8.675 | 0.614 | —— | —— |
| | Mind-video (Chen et al. (2024)) | 0.679 | 0.780 | **0.609** | 0.117 | 8.673 | 0.784 | 0.267 | 8.102 |
| | Ours | **0.731** | **0.788** | 0.594 | **0.502** | **11.310** | **0.820** | **0.400** | **3.551** |
| sub 10 | Nishimoto (Nishimoto et al. (2011)) | **0.685** | —— | —— | 0.441 | 9.598 | 0.662 | —— | —— |
| | Wen (Wen et al. (2018)) | 0.638 | —— | —— | 0.265 | 8.565 | 0.644 | —— | —— |
| | Mind-video (Chen et al. (2024)) | 0.663 | 0.770 | 0.563 | 0.108 | 8.912 | 0.809 | 0.185 | 7.524 |
| | Ours | 0.684 | **0.777** | **0.590** | **0.465** | **11.128** | **0.858** | **0.408** | **3.533** |

learning or cross-subject learning methods that can be trained using data collected from different subjects, which we consider as a direction for future research.

## F    FURTHER RESULTS ON INTERPRETABILITY ANALYSIS

### F.1    WHY UTILIZE CONTRASTIVE LEARNING?

It is noted that there has been an increasing body of work utilizing contrastive learning for neural decoding (Chen et al. (2024); Scotti et al. (2024); Benchetrit et al. (2024)). To explore whether decoders trained with contrastive learning loss demonstrate advantages over those trained with mean squared error (MSE) loss in accuracy and generalization, we conduct the retrieval task and employ t-distributed stochastic neighbor embedding (t-SNE) for visualization.

The retrieval task involves identifying which specific visual stimulus has evoked a given fMRI response from a predefined set. This task differs from the classification task, which only recognizes the category of the visual stimulus that evoked the fMRI response. In contrast, the retrieval task demands a finer level of detail, requiring not just the correct classification but also the precise identification of the particular visual stimulus. During the training process of the semantic decoder, we align the fMRI representation to the pre-trained CLIP representational space via a tri-model contrastive learning loss. This design enables the semantic decoder to be not only applicable to video reconstruction task but also extends its utility to retrieval task.

To the best of our knowledge, no prior work has been reported on conducting retrieval task using the CC2017 dataset. Therefore, we train a simple linear regression model and a three-layer MLP as our baselines. We employ top-10 accuracy[1] and top-100 accuracy as evaluation metrics. To validate the model's generalization capability, we not only conduct the retrieval task on the CC2017 test set, comprising 1200 samples and termed the 'small test set', but also expand the stimulus set to enhance its scope. Specifically, we integrate 3040 video clips from the HCP dataset into our collection, creating an extended stimulus set totaling 4,240 samples, which we label as the 'large test set'.

Table 17: Results of retrieval task on CC2017 dataset. For the 'small test set', the chance-level accuracies for top-10 and top-100 accuracy are 0.83% and 8.3%, respectively. For the 'large test set', the chance-level accuracies for top-10 and top-100 accuracy are 0.24% and 2.4%, respectively.

| Dataset | | CC2017 | | | | | | |
|---|---|---|---|---|---|---|---|---|
| | | Subjet1 | | Subjet2 | | Subjet3 | | Average |
| Model | Test set | top-10 | top-100 | top-10 | top-100 | top-10 | top-100 | top-10 | top-100 |
| Linear + MSE | Small | 2.25 | 16.92 | 3.17 | 22.25 | 2.75 | 19.75 | 2.72 | 19.64 |
| 3-layer MLP + MSE | Small | 1.00 | 9.42 | 1.42 | 11.75 | 0.92 | 9.50 | 1.11 | 10.22 |
| 3-layer MLP + Contrast (Ours) | Small | **3.08** | **22.58** | **4.75** | **26.90** | **4.50** | **24.67** | **4.11** | **24.72** |
| Linear + MSE | Large | 1.08 | 7.83 | 1.92 | 10.75 | 1.92 | 9.92 | 1.64 | 9.50 |
| 3-layer MLP + MSE | Large | 0.42 | 3.58 | 0.75 | 6.83 | 0.25 | 1.08 | 0.47 | 3.83 |
| 3-layer MLP + Contrast (Ours) | Large | **2.17** | **12.50** | **2.25** | **17.00** | **2.75** | **16.42** | **2.39** | **15.31** |

The experimental results of the retrieval task on the CC2017 dataset across three subjects are depicted in Figure 20 and Table 17. As demonstrated in Table 17, our model outperforms the baseline models under both 'small test set' and 'large test set' conditions. Notably, when the stimulus set is expanded to nearly four times its original size, the performance of our model does not experience a sharp decline. This suggests that aligning the model's latent space to the CLIP representational space through contrastive learning loss is beneficial for enhancing the model's generalization capability.

As illustrated in Figure 21, we utilize t-SNE to visualize the predicted CLIP representations of the models. The visualization results indicate that the linear regression model exhibits the weakest generalization when performing neural decoding tasks. Increasing the number of layers in the linear model can improve its generalization ability to some extent. However, model trained with a contrastive learning loss demonstrate superior generalization performance.

---

[1]Top-k accuracy is the percentage of queries where the correct item is among the top k results returned by a retrieval model.

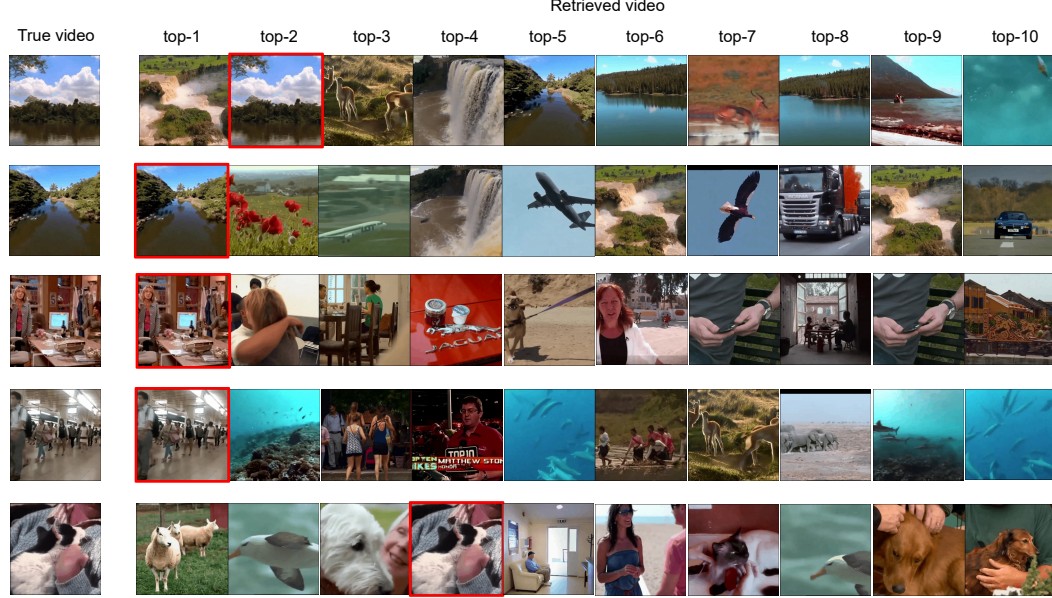

Figure 20: The retrieval performance of our model (trained with contrastive learning loss) on the CC2017 dataset. This figure showcases the top ten video stimuli retrieved based on fMRI. Owing to space limitations, a single frame is randomly selected for display from each of the video stimuli.

## F.2 WHICH BRAIN REGIONS ARE RESPONSIBLE FOR DECODING DIFFERENT FEATURES, RESPECTIVELY?

To supplement the bar chart in Figure 9, we normalized the importance values of each ROI in decoding the three features (semantic, structure, motion) and visualized them in Figure 22. In this figure, darker colors represent higher contributions of a given ROI during decoding. A horizontal comparison for each feature reveals the following:

(1) Motion Decoding: V1, V2, MT, and TPOJ regions contribute more significantly to motion decoding. Notably, MT and TPOJ are regions in the dorsal pathway responsible for processing motion information. Meanwhile, V1 and V2 transmit motion-related attributes such as speed and direction directly to MT when processing motion. This finding aligns well with prior results in neuroscience (Zeki & Shipp (1988); Nassi & Callaway (2009)).

(2) Structure Decoding: Lower-level regions like V1, V2, and V3 show greater contributions to structure decoding, while higher-level regions such as MT contribute less.

(3) Semantic Decoding: Mid-to-high-level regions, including V4, MT, and MST, play a more significant role in semantic decoding, while lower-level regions such as V1 and V2 contribute less. Interestingly, MT, typically a dorsal pathway region for motion processing, shows the highest contribution to semantic decoding. This phenomenon may be explained by the interplay between the dorsal and ventral pathways in processing dynamic visual input (Ingle et al. (1982)). Specifically, the dorsal-dorsal pathway is concerned with the control of action, whereas the ventral-dorsal pathway is involved in action understanding and recognition (Rizzolatti & Matelli (2003)).

The visualization of voxel-wise and ROI-wise importance maps on Subject 2 and 3 is depicted in the aforementioned Figures.

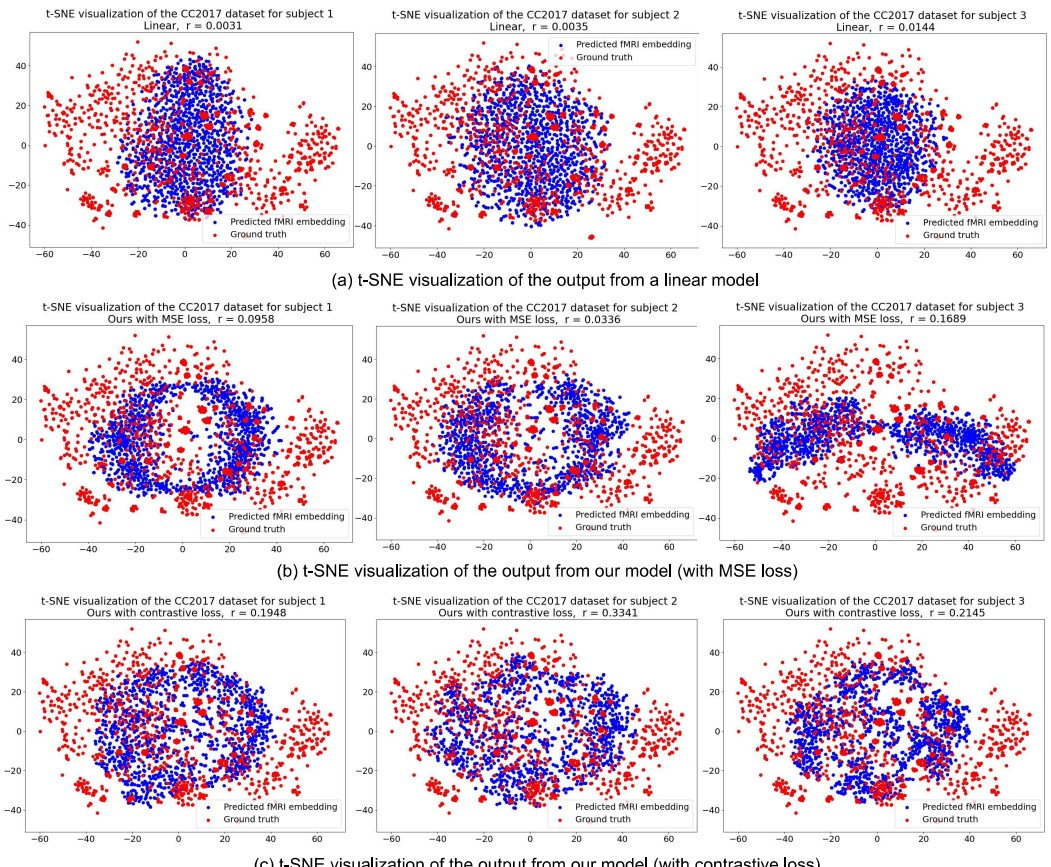

(a) t-SNE visualization of the output from a linear model

(b) t-SNE visualization of the output from our model (with MSE loss)

(c) t-SNE visualization of the output from our model (with contrastive loss)

Figure 21: t-SNE visualization presents a comparative analysis of the representations predicted by three decoders on 1200 samples from the CC2017 test set: a simple linear regression model trained with MSE loss, our semantic decoder trained with MSE loss, and our semantic decoder trained with contrastive learning loss. The red dots represent the real CLIP representations, while the blue dots denote the representations predicted by the decoders. The absolute Pearson correlation coefficient (i.e. $r$) between the real and predicted representations is displayed above each subfigure.

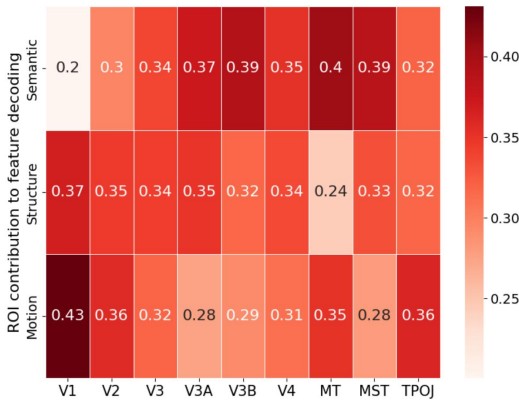

Figure 22: Supplementary ROI-wise importance map on subject 1.

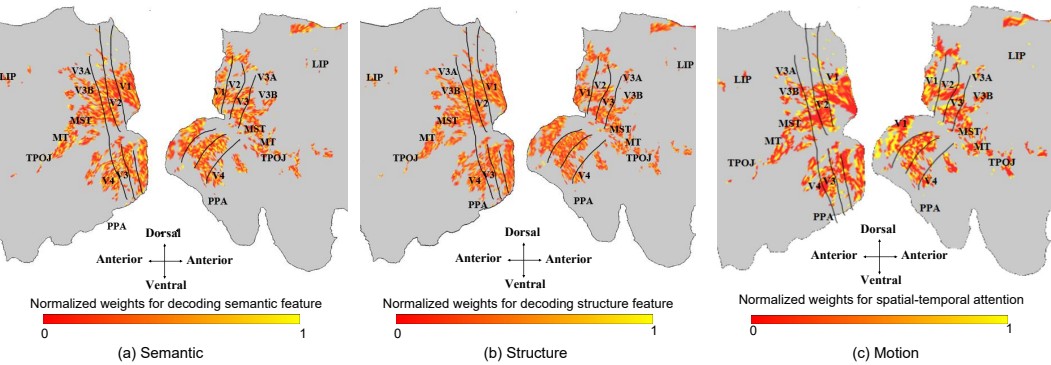

Figure 23: **Voxel-wise** importance maps projected onto the visual cortex of subject 2. The lighter the color, the greater the weight of the voxel in the interpretation of feature.

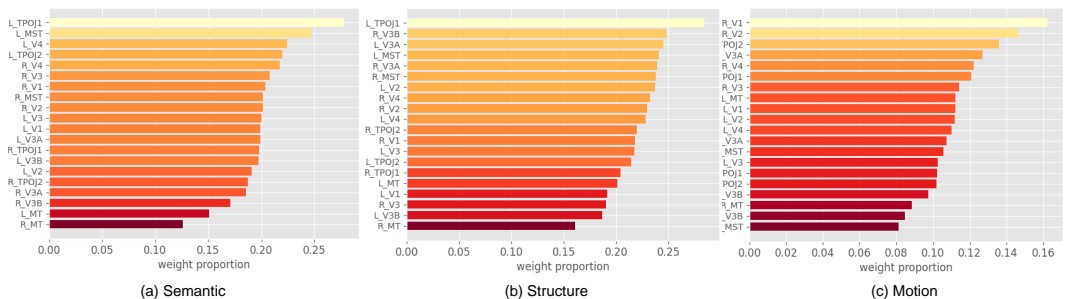

Figure 24: **ROI-wise** importance maps in the visual cortex of subject 2.

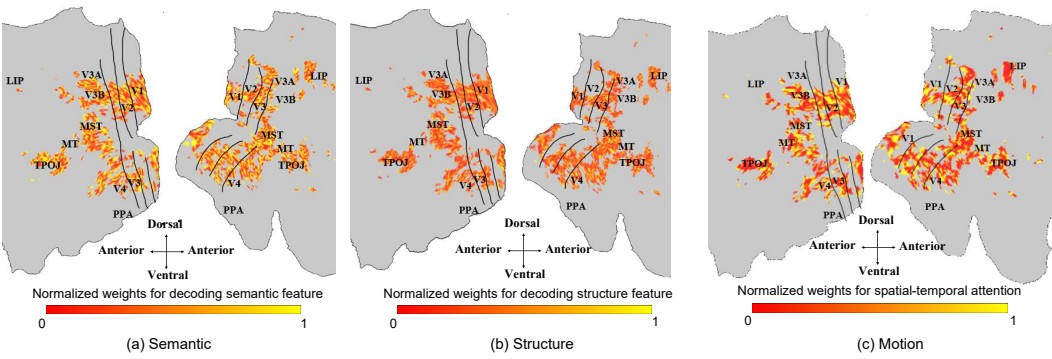

Figure 25: **Voxel-wise** importance maps projected onto the visual cortex of subject 3. The lighter the color, the greater the weight of the voxel in the interpretation of feature.

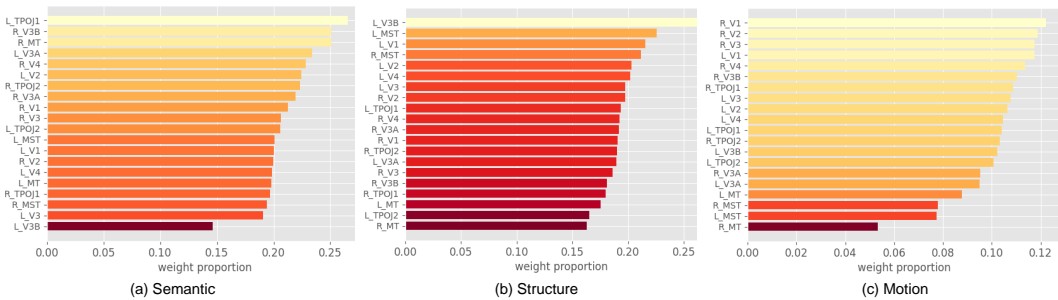

Figure 26: **ROI-wise** importance maps in the visual cortex of subject 2.

## G    SUPPLEMENTARY KNOWLEDGE ON THE MECHANISMS OF DYNAMIC VISUAL INFORMATION PROCESSING IN THE VISUAL CORTEX.

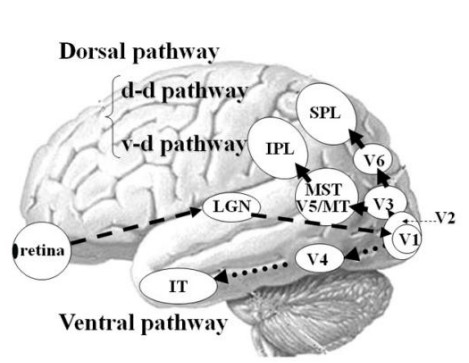

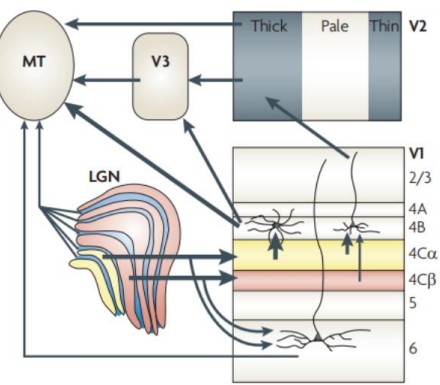

(a) Parallel visual pathways in humans. The image is cited from Yamasaki & Tobimatsu (2011).

(b) Multiple input streams to MT. The image is cited from Nassi & Callaway (2009).

Figure 27: LGN, lateral geniculate nucleus; V1, 2, 3, 4 and 6 are the primary, secondary, tertiary, quaternary and senary visual cortices, respectively; V5/MT, quinary visual cortex/middle temporal area; MST, medial superior temporal area; IPL, inferior parietal lobule, SPL, superior parietal lobule; and IT, inferior temporal cortex; d-d pathway, dorso-dorsal pathway; v-d pathway, ventro-dorsal pathway.

This study focuses on reconstructing dynamic visual information from brain responses. To validate the interpretability of our model, we employed cortical projection techniques for visualization. To facilitate readers' understanding of the mechanisms underlying how the human brain's visual cortex processes motion information, we provide additional explanations in this section.

As shown in Figure 27a, light signals are converted into electrical signals by photoreceptors. The electrical signals are transmitted through bipolar cells and modulated by horizontal cells and amacrine cells, reaching the final station of the retina—the retinal ganglion cells (RGCs). The RGCs organize and compress the stimulus information before forwarding it to the lateral geniculate nucleus (LGN). Upon receiving signals from the LGN, the visual areas of the cerebral cortex initially process the information in the V1 region. Subsequently, the signals are divided into two parallel pathways: the dorsal pathway and the ventral pathway. The dorsal pathway primarily handles **motion perception and spatial vision**, while the ventral pathway is mainly responsible for **object recognition** (Gilbert & Li (2013)). **Although the dorsal and ventral streams clearly make up two relatively separate circuits, the anatomical segregation between the two streams is by no means absolute** (Nassi & Callaway (2009); Andersen et al. (1990); Blatt et al. (1990); Maunsell & van Essen (1983)). Recently, the dorsal stream was shown to be divided into two functional streams in primates to mediate different behavioural goals: the dorsal-dorsal and ventral-dorsal streams (Rizzolatti & Matelli (2003)). The dorsal-dorsal pathway concerned with the control of action 'online' (while the action is ongoing) and the ventral-dorsal pathway concerned with space perception and 'action understanding' (the recognition of actions made by others).

**The ventral pathway exhibits a hierarchical information extraction process** (Markov et al. (2014)). From V1, V2, V4 to the inferior temporal (IT) cortex, neurons progressively encode information, transitioning from basic visual features (such as orientation and color) to intermediate shape characteristics, and finally to high-level visual semantics (such as faces, limbs, and scenes). **In contrast, the dorsal pathway does not typically follow a hierarchical information extraction process but rather adopts a parallel processing approach** (Callaway (2005)). As shown in Figure 27b, there are multiple input streams from the LGN to the MT area. The major ascending input to MT traverses the magnocellular layers of the LGN (yellow) and proceeds through layers $4C\alpha$ and 4B of the V1 (Shipp & Zeki (1989); Nassi & Callaway (2009)). Experimental evidence suggests that the direct pathway from V1 to MT primarily conveys information about motion speed and direction, while the indirect pathway is responsible for transmitting disparity information (Ponce et al. (2008)).

