# OpenReview forum: "Animate Your Thoughts: Reconstruction of Dynamic Natural Vision from Human Brain Activity"
_ICLR.cc/2025/Conference — ICLR 2025 Poster_

### Official Review · Reviewer_fFuu · 2024-10-23

**Soundness:** 3
**Presentation:** 3
**Contribution:** 3
**Rating:** 6
**Confidence:** 3

**Summary:**

This work addresses the problem of reconstructing high-quality video from fMRI data. The authors suggest that the key to achieving high-quality video reconstruction lies in decoupling and modeling semantic, structural, and motion information, also carefully handling the frequency discrepancy between fMRI data and videos. To this end, the authors develop a tri-modal contrastive learning scheme along with a next-frame prediction task. Lastly, to ensure that the generated videos are derived purely from the fMRI data, the input is fed into an untuned inflated Stable Diffusion model. Empirical evaluations show promising results and strong interpretability performance.

**Strengths:**

[Quality, Significance]:

The structure of this work is very easy to understand and follow, with sufficient context and supported citations. The model notations and figure presentations are excellent.

The authors provides comprehensive empirical metric evaluations and compared their results with many other state-of-the-art approaches, and provide comprehensive ablation study and interoperability results.

[Novelty]: One particular point that I find this work novel is that the authors do not fine-tune the stable diffusion, ensuring us that the videos that we see are not coming from the overfitting.

**Weaknesses:**

I do not find this work particularly having major weaknesses.

**Questions:**

1. How did the author chose $\lambda_1$ and $\lambda_2$ in equation 4? Can the author provide the ablation study/ discussion on how these choices affect the downstream performance?

---

> ### Author Response · Authors · 2024-11-21
> **Response to Reviewer fFuu**
>
> We thank you for the strong support and the positive comments on our work. Your inspiring questions and comments are valuable for our future work. We have carefully revised the manuscript in accordance with your suggestions, **with the changes highlighted in purple**, and have submitted the updated version. Our point-by-point responses are as follows.
>
> ## 1. How did the author chose $\lambda_1$ and $\lambda_2$ in equation 4?
> ## Response:
>
> As shown in Figure 12 of Appendix E3, setting either $\lambda_1$ or $\lambda_2$ to 0 prevents the semantic decoder from converging during training. Therefore, $\lambda_1$ and $\lambda_2$ were empirically determined to balance the magnitudes of $L_{Semantic}$ and $L_{Projection}$ as much as possible during the early stages of training.
>
> ## 2. Can the author provide the ablation study/ discussion on how these choices affect the downstream performance?
> ## Response:
>
> We conducted a sensitivity analysis on the selection of $\lambda_1$ and $\lambda_2$ using sub1 from the CC2017 dataset. As shown in Figure 12, when either $\lambda_1$ or $\lambda_2$ is set to 0, the semantic decider fails to converge, indicating that both the contrastive loss and projection loss play crucial roles in decoding semantic information. In Table 9, we set $\lambda_1 = 0.01$ and $\lambda_2 = 0.5$ (Ours) to ensure that both loss terms are balanced during optimization. We then introduced small perturbations to $\lambda_1$ and $\lambda_2$ to observe how they affect downstream performance.
>
>
> | Model                            | 2-way-I | 2-way-V | VIFI-score | SSIM  | PSNR   | Hue-pcc | CLIP-pcc↑ | EPE↓  |
> |----------------------------------|---------|---------|------------|-------|--------|---------|-----------|-------|
> | $\lambda_1 = 0.01$, $\lambda_2 = 0.25$ | 0.765 *   | 0.825  * | 0.581  *    | 0.318 | **10.199** * | **0.780**   | 0.396 *    | 6.025 *|
> | $\lambda_1 = 0.005$, $\lambda_2 = 0.5$ | 0.786 *  | 0.824  * | 0.591   *   | 0.320 | 9.109  | 0.776   | 0.407     | 5.898  *|
> | $\lambda_1 = 0.01$, $\lambda_2 = 0.5$ (Ours) | **0.812** | **0.839** | **0.604**  | **0.319** | 9.116  | 0.778   | **0.413** | **5.572** |
>
> *Note: * denotes our performance is significantly better than the compared method (paired t-test, p<0.05).*
>
>
> From the table above, it can be observed that adjusting either $\lambda_1$ or $\lambda_2$ and disrupting this balance does not affect the structural-level metrics of the reconstruction results but does influence the semantic and spatiotemporal-level metrics significantly.
>
>
> Thank you again for your valuable suggestions to improving our work. We believe that, under your review, our manuscript will be significantly improved in terms of clarity and experimental design. We look forward to your feedback and further discussions.

---

> > ### Comment · Reviewer_fFuu · 2024-11-23
> >
> > Thanks for your ablation experiments, please include them in your revision.
> >
> > Bests.

---

> > > ### Author Response · Authors · 2024-11-24
> > > **Response to Reviewer fFuu**
> > >
> > > Thank you for your response. We have incorporated the results of the ablation experiments into Appendix E. 3 (page 27) of the revised manuscript.

---

### Official Review · Reviewer_5KvP · 2024-10-24

**Soundness:** 3
**Presentation:** 3
**Contribution:** 2
**Rating:** 8
**Confidence:** 4

**Summary:**

This paper addresses the challenge of video reconstruction from fMRI data, identifying three key components for an effective decoder: semantics, visual structure, and motion patterns. Building on prior works that successfully capture semantic content using CLIP and Stable Diffusion, the authors aim to improve fidelity to the visual structure and motion dynamics present in the original videos. They propose a decoupled, multi-step reconstruction approach, utilizing distinct decoders and specialized reconstruction criteria. Their method combines tri-modal CLIP (fMRI, image, text), single-frame Stable Diffusion (T2I), and a uniquely learned internal motion prior that avoids biases from external video datasets. Applied to three publicly available movie-fMRI datasets, the approach demonstrates moderate gains in some metrics across different configurations, with notable improvements in preserving visual structure.

**Strengths:**

The authors’ effort to achieve simultaneous reconstruction of visual structure, semantics, and motion patterns represents a significant and timely contribution. Many prior works prioritize visually appealing reconstructions or focus on semantic fidelity, often overlooking finer alignment with the ground truth video structure. Advancing fidelity in this regard is important; without it, decoders risk becoming general natural video generators with limited adherence to fMRI cues.

The primary improvement over previous methods appears in the preservation of visual structure, evidenced both qualitatively and quantitatively, though this improvement is moderate. The authors’ thorough analysis across three datasets, coupled with an extensive benchmarking on various metrics assessing semantics and visual structure, adds robustness to the work. Additionally, the website offers a helpful visual supplement to this complex work, making it more accessible and digestible.

**Weaknesses:**

The results, both qualitatively and quantitatively, lack compelling evidence of substantial improvement over prior work, particularly when compared to Chen et al. (2024). In fact, it appears that with proper statistical analysis (e.g., Table 2-4), there may be no significant gains. Methodologically, the advancement over Chen et al. (2024) seems minimal, with certain versions of the authors' approach even reintroducing external videos—the very bias this paper aims to avoid. Furthermore, the claim of improved motion pattern reconstruction is insufficiently supported, as the authors compare only against shuffled frames. A meaningful comparison should have been made with prior works, similar to the comparisons on semantic and visual structure fidelity.

Additional comments:
- [36-38] Statement is unclear; consider rephrasing.
- [54-59] The text appears to confuse BOLD integration duration with fMRI sampling rate. The BOLD signal integrates neural activity over a period greater than 10 seconds (~300 video frames), which is a major limitation in recovering any motion patterns potentially encoded in neural activity. The fMRI sampling rate is limited for other technical reasons but even if was sampling at a higher rate this would likely not resolve this fundamental issue.
- [100-101] The claim about enabling video reconstruction may be overstated, as prior methods have also achieved this to an extent.
- [102-104] Similar to above, there is a mix-up regarding temporal aspects. Clarification of this contribution would help.
- [104-106] The visualizations mentioned are not novel to this work; they have been used in prior studies and are reintroduced here rather than proposed anew.
- [107-110]  It remains unclear what metric is used to evaluate successful recovery of motion patterns. Detailing this metric is necessary for interpreting the results.

**Questions:**

1) Consider adding robust statistical analyses and uncertainty estimates for the primary results. This would clarify the significance of the observed gains, if any.

2) For the claim regarding improved recovery of motion patterns, stronger evidence is needed. As presented, the motion generator appears modest/limited in comparison to the tools used for recovering structure and semantics. Since improvement in motion fidelity is a key goal, it would be impactful to demonstrate and emphasize substantial gains in this area. If stronger evidence cannot be shown, it may be best to reconsider this claim.

3) The authors note that MT, an area associated with motion processing, shows significant activation for semantics, typically associated with the ventral stream. If this outcome supports the motion recovery, it could benefit from additional context. Some clarification on MT’s role in semantics would be helpful for the reader’s interpretation of these findings.

4) For Figure 8, consider enhancing its visual accessibility to make it easier to interpret. Improvements in layout or clarity could aid the reader in understanding the figure’s contribution.

---

> ### Author Response · Authors · 2024-11-21
> **Response to Reviewer 5KvP**
>
> We sincerely appreciate your recognition of the contributions of our work. Thank you for taking the time to point out the areas for improvement in both the presentation and experimental design. We have carefully revised the manuscript in accordance with your suggestions, with **the changes highlighted in green**, and have submitted the updated version. Below are our point-by-point responses to your comments.
>
> # Experimental Issues
> ## 1. The experimental results lack compelling evidence of significant improvement over prior work (particularly when compared to Chen et al. (2024)) and would benefit from robust statistical analyses.
> ## Response:
>
> Thank you for identifying this issue. We acknowledge that without robust statistical analyses, it is challenging to determine whether the improvements achieved by our model are significant. Therefore, we have incorporated t-test results into Tables 2, 3, 4, and 5. For these tables, the experimental results were first averaged across three subjects before conducting statistical analyses.
>
> From Tables 2, 3, and 4, it can be observed that our model significantly outperforms Chen et al. (2024) on 6 out of 8 metrics in the CC2017 dataset, 3 out of 4 metrics in the HCP dataset, and all 4 metrics in the Algonauts 2021 dataset.
>
> We note, however, that our model falls short of outperforming Chen et al. (2024) on the 2-Way-Image Identification accuracy (2-Way-I) metric, which measures the semantic similarity between reconstructed results and ground truth. We provide the following explanations for this outcome:
>
> -  $ Focus$  $of$  $our$  $work  $:  Chen et al. (2024), as an influential work in reconstructing semantically meaningful videos from fMRI, overlooks structural and motion information, which are the primary focus of our study. As a result, our model was not explicitly designed for semantic decoding.
>
> - $ Pretraining$  $vs.$  $Random$  $ Initialization $ : Chen et al. (2024) leverages pretraining on a large, unpaired fMRI dataset (HCP) to learn intrinsic representations from fMRI, followed by fine-tuning on the CC2017 dataset. In contrast, our model was randomly initialized and trained directly on the CC2017 dataset. This "pretraining + fine-tuning" paradigm likely contributes to their superior performance in semantic decoding and represents a potential direction for our future work.
> - $Retrieval$ $tasks$ $with$ $varying$ $difficulty$ : The 2-Way-Image Identification accuracy is computed by retrieving the corresponding ground truth from two videos (one being the ground truth of the reconstruction and the other randomly selected), with a chance-level accuracy of 50%. This task is relatively simple. To comprehensively assess semantic decoding performance, we evaluated two more challenging retrieval tasks: (i) retrieving the ground truth from 1,200 test videos in the CC2017 dataset (Small) and (ii) retrieving the ground truth from an expanded set of 4,240 videos (Large). The chance-level accuracies for the top-10 retrieval in these tasks are 0.83% and 0.24%, respectively, making them significantly more challenging. Using the reconstruction results from Chen et al. (2024), we evaluated these tasks and found that our model significantly outperformed Chen et al. (2024), as shown in the table below.
>
> | Model         | Test set | **Subject 1** |           | **Subject 2** |           | **Subject 3** |           | **Average**  |           |
> |---------------|----------|---------------|-----------|---------------|-----------|---------------|-----------|--------------|-----------|
> |               |          | top-10        | top-100   | top-10        | top-100   | top-10        | top-100   | top-10       | top-100   |
> | Chen et al. (2024)   | Small    | **3.22**      | 19.08     | 2.75          | 16.83     | 3.58          | 22.08     | 3.18*        | 19.33*    |
> | **Ours**      | Small    | 3.08      | **22.58** | **4.75**      | **26.90** | **4.50**      | **24.67** | **4.11**     | **24.72** |
> | Chen et al. (2024)   | Large    | 1.75          | 7.17      | 0.83          | 5.17      | 1.25          | 9.00      | 1.28*        | 7.11*     |
> | **Ours**      | Large    | **2.17**      | **12.50** | **2.25**      | **17.00** | **2.75**      | **16.42** | **2.39**     | **15.31** |
>
> *Note: For the 'small test set', the chance-level accuracies for top-10 and top-100 accuracy are 0.83% and 8.3%, respectively. For the 'large test set', the chance-level accuracies for top-10 and top-100 accuracy are 0.24% and 2.4%, respectively. * denotes our performance is significantly better than the compared method (paired t-test, p<0.05).
>
> - $Limitations$ $of$ $the$ $metric$ : The 2-Way-I metric evaluates the average semantic similarity between individual video frames and the ground truth, without considering the inter-frame relationships. This paper, however, focuses on decoding the motion associations between frames. Thus, we believe that not outperforming Chen et al. (2024) on this metric is acceptable.

---

> ### Author Response · Authors · 2024-11-21
> **Response to Reviewer 5KvP (cont.)**
>
> ## 2. Reintroducing external videos introduces a bias that this paper aims to avoid.
> ## Response:
>
> To ensure a fair comparison with Chen et al. (2024) and "Kupershmidt22", we incorporated EV for further fine-tuning. However, we would like to clarify that EV was used only when comparing model performance on the CC2017 dataset. In all other parts of the paper, we report experimental results without using EV, ensuring that it does not introduce a bias that this paper aims to avoid.
>
> To address potential misunderstandings stemming from the inclusion of EV, we have removed all EV-related experiments from Table 2 and omitted any corresponding descriptions from Section 5.1 in the revised manuscript.
>
> ## 3. The claim of improved motion pattern reconstruction lacks sufficient support and should include a meaningful comparison with prior work.
> ## Response:
>
> Thank you for pointing this out. To comprehensively evaluate the reconstruction performance of our model, we propose three types of evaluation metrics (eight in total): Semantic-level, Structure-level, and Spatiotemporal-level, with the latter specifically designed to assess motion pattern reconstruction.
>
> The Spatiotemporal-level metrics include $CLIP-pcc$ and $End-Point$ $ Error$ $(EPE)$ :
>
> - $EPE$ measures the Euclidean distance between the endpoints of the predicted and ground truth trajectories for each corresponding frame. It provides a quantitative assessment of the similarity between the **motion trajectories** of the predicted and ground truth videos and is widely used in motion-sensitive tasks such as optical flow estimation [1] .
>
> - $CLIP-pcc$ calculates the CLIP image embeddings for each frame in the predicted videos and reports the average cosine similarity between all pairs of adjacent frames. This metric evaluates the **coherence of consecutive frames** in the video and is commonly applied in video generation and editing [2] .
>
> As shown in Tables 2, 15, and 16, our model significantly outperforms previous methods on both metrics in three datasets, demonstrating its improved capability for motion pattern reconstruction.
>
> | **Models**         | **Dataset**   | **CLIP-pcc↑** | EPE↓  |
> |:------------------:|:-------------:|:-------------:|:---------:|
> | Wang et al. (2022) | CC2017        | 0.399 *       | 6.344 *   |
> | Chen et al. (2024) |  CC2017             | 0.409 *       | 6.125 *   |
> | Ours               |  CC2017             | **0.425**         | **5.422**     |
> | Chen et al. (2024) | HCP           | 0.499 *       | 9.290 *    |
> | Ours               | HCP              | **0.511**         | **7.080**      |
> | Chen et al. (2024) | Algonauts2021 | 0.246 *       | 7.693 *   |
> | Ours               | Algonauts2021              | **0.401**         | **3.264**     |
>
> *Note: * denotes our performance is significantly better than the compared method (paired t-test, p<0.05).*
>
> To further clarify for readers, we have provided additional explanations of these two metrics in Section 4.2, specifically in lines 319–323 of the revised manuscript, with the updates highlighted in green.
>
>
> ## 4. Stronger evidence is needed for the claim regarding improved recovery of motion patterns.
> ## Response:
>
> Thank you for raising this issue. In addition to the shuffle test, we have incorporated an ablation study on the CMG module in Table 6, as suggested by Reviewer 1xQ5 (Soundness 1), to provide stronger evidence.
>
> Specifically, we removed the fMRI guidance in the CMG module (replacing the fMRI input in cross-attention with the token from the previous frame) while keeping all other model structures and hyperparameters unchanged. We then computed the motion-related metrics of the reconstruction results. The table below presents the results:
>
> | **Models**        | **Subject** | **CLIP-pcc↑** | **EPE↓**  |
> |:-----------------:|:-----------:|:-------------:|:---------:|
> | w/o fMRI guidance | sub 1       | 0.381 *        | 6.293   *  |
> | Full Model        | sub 1       | **0.413**        | **5.572**    |
> | w/o fMRI guidance | sub 2       | 0.343  *       | 7.571    * |
> | Full Model        | sub 2       | **0.423**         | **5.329**    |
>
> *Note: * denotes our performance is significantly better than the compared method (paired t-test, p<0.05).*
>
> As shown in the table, removing the fMRI guidance results in a significant deterioration in both CLIP-pcc and EPE, demonstrating that the motion information in our reconstruction results originates from the fMRI data rather than the video training set.
>
> References：
>
> [1] John L Barron, David J Fleet, and Steven S Beauchemin. Performance of optical flow techniques. International journal of computer vision, 12:43–77, 1994.
>
> [2] Jay Zhangjie Wu, Yixiao Ge, Xintao Wang, Stan Weixian Lei, Yuchao Gu, Yufei Shi, Wynne Hsu, Ying Shan, Xiaohu Qie, and Mike Zheng Shou. Tune-a-video: One-shot tuning of image diffusion models for text-to-video generation. In Proceedings of the IEEE/CVF International Conference on Computer Vision, pp. 7623–7633, 2023.

---

> > ### Comment · Reviewer_5KvP · 2024-11-27
> > **The claim of improved motion pattern reconstruction lacks sufficient support and should include a meaningful comparison with prior work.**
> >
> > Thanks for the clarification.
> >
> > Tables 15, 16 show no statistical assessments. Table 2 and I think also the table sent here is a cross-subject average statistic being analyzed. Arguably, the gain should hold per subject in a subject-specific analysis.

---

> > > ### Author Response · Authors · 2024-11-28
> > > **Supplementary Results and Explanations on Statistical Assessments (4/4)**
> > >
> > > ## 4. The hypothesis testing results for the three subjects on the Retrieval task.
> > > ## Response:
> > > We apologize for providing only the hypothesis testing results averaged across the three subjects in our previous rebuttal, which may have caused some misunderstanding. We greatly appreciate you pointing out this issue. We have now added the individual subject hypothesis testing results for the Retrieval task.
> > >
> > > | Model         | Test set | **Subject 1** |           | **Subject 2** |           | **Subject 3** |           | **Average**  |           |
> > > |---------------|----------|---------------|-----------|---------------|-----------|---------------|-----------|--------------|-----------|
> > > |               |          | top-10        | top-100   | top-10        | top-100   | top-10        | top-100   | top-10       | top-100   |
> > > | Wen [1]          | Small    | 2.17*          | 19.50*     | 3.33*          | 19.17*     | ——            | ——        | 2.75*        | 19.33*    |
> > > | Kupershmidt [2]   | Small    | 1.09*          | 8.57*      | 0.92*          | 8.24*      | 0.84*          | 8.24*      | 0.95*        | 8.35*     |
> > > | Mind-video [3]   | Small    | **3.22***      | 19.08*     | 2.75*          | 16.83*     | 3.58*          | 22.08*     | 3.18*        | 19.33*    |
> > > | **Ours**      | Small    | 3.08      | **22.58** | **4.75**      | **26.90** | **4.50**      | **24.67** | **4.11**     | **24.72** |
> > > | Wen   [1]        | Large    | 1.41*          | 11.58*     | 2.08*          | 9.58*      | ——            | ——        | 1.75*        | 10.58*    |
> > > | Kupershmidt [2]   | Large    | 0.17*          | 2.94*      | 0.17*          | 2.77*      | 0.25*          | 2.18*      | 0.19*        | 2.63*     |
> > > | Mind-video  [3]  | Large    | 1.75*          | 7.17*      | 0.83*          | 5.17*      | 1.25*          | 9.00*      | 1.28*        | 7.11*     |
> > > | **Ours**      | Large    | **2.17**      | **12.50** | **2.25**      | **17.00** | **2.75**      | **16.42** | **2.39**     | **15.31** |
> > >
> > > *Note: For the 'small test set', the chance-level accuracies for top-10 and top-100 accuracy are 0.83% and 8.3%, respectively. For the 'large test set', the chance-level accuracies for top-10 and top-100 accuracy are 0.24% and 2.4%, respectively. The metrics are evaluated using 100 bootstrap trials. * denotes our performance is significantly better than the compared method (Wilcoxon test for paired samples, p<0.05).*
> > >
> > > ## 5. Analysis of experimental results
> > > ## Response:
> > > After improving the hypothesis testing for each participant, we analyzed the experimental results of our model and those of Chen et al. (2024) in the tables above, leading to the following conclusions:
> > >
> > > ### (1) Pixel-level metrics:
> > >  Our model significantly outperforms Chen et al. (2024) **across all Pixel-level metrics on three datasets (16 participants in total)**, highlighting the effectiveness of incorporating structural feature decoding in video reconstruction.
> > >
> > > ### (2) ST-level metrics:
> > > For CLIP-pcc, our model is significantly weaker than Chen et al. (2024) only for sub 01 in the HCP dataset, comparable for sub 02, and **significantly better for the remaining 14 participants**. For EPE, our model **significantly outperforms Chen et al. (2024) across all three datasets**. Notably, on the Algonauts2021 dataset, our model exceeds Chen et al. (2024) by more than **two times** for all 10 subjects in terms of EPE. This result underscores the significant advantage of our model over prior SOTA model in motion pattern reconstruction.
> > >
> > > ### (3) Semantic-level metrics:
> > >  For 2-Way-I, our model is significantly weaker than Chen et al. (2024) for 2 subjects, comparable for 3 subjects, and **significantly better for the remaining 11 subjects**. For 2-Way-V, our model is significantly weaker for 4 subjects, comparable for 1 subject, and **significantly better for the other 11 subjects**. For VIFI-score, our model is comparable to Chen et al. (2024) for sub 02 in the HCP dataset and sub 08 in the Algonauts2021 dataset, while **outperforming Chen et al. (2024) for the remaining 14 subjects**.
> > >
> > > ### (4) Retrieval task:
> > >  The results from the CC2017 dataset for 3 subjects indicate that our model is significantly weaker than Chen et al. (2024) in top-10 accuracy for sub 01 when the test set is configured as "Small." However, our model outperforms Chen et al. (2024) for all other participants and settings.
> > >
> > > It is important to note that the amount of training data used by our model is significantly smaller than that of Chen et al. (2024): Chen et al. used **600,000 segments for pretraining and 18 segments for fine-tuning**, while our model only used **18 segments for training**. Despite this disparity in data volume, our model still achieves competitive results in semantic pattern reconstruction, suggesting that it holds a certain advantage over the previous SOTA model.

---

> > > > ### Comment · Reviewer_5KvP · 2024-12-02
> > > >
> > > > I believe that the authors' thorough responses and revisions introduced make this paper and its results much more compelling.
> > > > Much appreciated.
> > > > My scores are now revised accordingly.

---

> > > > > ### Author Response · Authors · 2024-12-03
> > > > > **Thanks for your recognition!**
> > > > >
> > > > > Thank you for taking the time to review our rebuttal. We are pleased that your concerns have been addressed, and we are honored to have received your support and recognition. We believe that, under your guidance, our manuscript has reached a higher level in terms of content clarity and experimental rigor.

---

> > > ### Author Response · Authors · 2024-11-28
> > > **Thanks for your suggestions!**
> > >
> > > Finally, we sincerely appreciate the time and effort you have dedicated to providing constructive feedback on our manuscript. We are truly honored by your thoughtful suggestions. If you have any further questions or additional recommendations, please do not hesitate to reach out. We look forward to your continued guidance and feedback.

---

> ### Author Response · Authors · 2024-11-21
> **Response to Reviewer 5KvP (cont.)**
>
> # Writing Issues
>
> ## 1. Line 36-38: Statement is unclear.
> ## Response:
> Thank you for pointing this out. To improve clarity, we have revised the manuscript accordingly, with the changes highlighted in green in the latest version.
>
> ## 2. Line 54-59: The text appears to confuse BOLD integration duration with fMRI sampling rate.
> ## Response:
> Thank you for pointing out the issue with our phrasing. We have made revisions in the latest version of the manuscript, with the changes highlighted in green in Lines 54-59：
>
> " Due to the inherent nature of fMRI, which relies on the slow blood oxygenation level dependent (BOLD) signal, neural activity is integrated over a period exceeding 10 seconds (~300 video frames). This integration delay poses a fundamental challenge in capturing rapid motion dynamics."
>
> ## 3. Line 100-101: The claim about enabling video reconstruction may be overstated, as prior methods have also achieved this to an extent.
> ## Response:
> We acknowledge that we are not the first to enable video reconstruction from fMRI. However, we are the first to reconstruct videos by decoupling semantic, structural, and motion information.
> We have made revisions in the latest version of the manuscript, with the changes highlighted in green in Lines 100-101.
>
> ## 4. Line 102-104: Similar to above, there is a mix-up regarding temporal aspects.
> ## Response:
> Thank you for pointing out this issue. We have made revisions in the latest version of the manuscript, with the changes highlighted in green in Lines 102-104：
>
> "This model decodes subtle yet significant motion patterns through a next-frame token prediction task despite the limitations imposed by the slow BOLD signal integration in fMRI."
>
> ## 5. Line 104-106: The visualizations mentioned are not novel to this work.
> ## Response:
> Thank you for pointing out the overclaim issue. We have revised the manuscript by changing "propose" to "use" in the relevant sections.
>
> ## 6. Line 107-110: It remains unclear what metric is used to evaluate successful recovery of motion patterns.
> ## Response:
> We have provided a detailed explanation of the two metrics used to evaluate the successful recovery of motion patterns in Section 4.2, Lines 319-323:
>
> - **End-Point Error (EPE)** measures the Euclidean distance between the endpoints of the predicted and ground truth trajectories for each corresponding frame. It provides a quantitative assessment of the similarity between the **motion trajectories** of the predicted and ground truth videos and is widely used in motion-sensitive tasks such as optical flow estimation and video editing.
>
> - **CLIP-pcc** calculates the CLIP image embeddings for each frame in the predicted videos and reports the average cosine similarity between all pairs of adjacent frames. This metric evaluates the **coherence of consecutive frames in the video** and is commonly applied in video generation and editing.
>
> ## 7. Consider improving the visual accessibility of Figure 8 to enhance its interpretability.
> ## Response:
>
> To complement the bar chart in Figure 8, we normalized the importance values of each ROI in decoding the three features (semantic, structure, motion) and visualized the results in Figure 21 in Appendix F2. Additionally, we provided an explanation for Figure 21 in Section F2, highlighted in green.

---

> ### Author Response · Authors · 2024-11-21
> **Response to Reviewer 5KvP (cont.)**
>
> ## 8. Clarification on the role of MT in semantic processing is needed to explain why MT shows significant activation for semantics.
> ## Response:
> Thank you for pointing out this important issue. After reviewing relevant literature in the field of neuroscience, we have found the following reasonable explanation:
>
> Although the dorsal and ventral streams clearly make up two relatively separate circuits, the anatomical segregation between the two streams is by no means absolute. Recently, the dorsal stream was shown to be divided into two functional streams in primates to mediate different behavioural goals: **the dorsal-dorsal and ventral-dorsal streams [1] .**   The dorsal-dorsal pathway concerned with the control of action and the ventral-dorsal pathway concerned with action understanding (the recognition and understanding of actions) [2] [3] [5] [6] . Our finding aligns with the latter.
>
> The MT area may be activated when the brain processes motion dynamics related to objects or actions in a stimulus video, aiding in the perception of motion patterns critical for interpreting the video's semantics, particularly those related to actions and relationships [2] .
> Thus, although the MT area is not directly responsible for semantic processing, it plays a crucial role in handling motion information related to the scene, contributing to the understanding of the video's semantic content [4] . **This differentiates it from how the brain processes static image information.**
>
> We have provided further clarification on this phenomenon in Lines 522-527 of the manuscript. **Additionally, we included relevant neuroscience knowledge in Appendix G to assist readers in understanding the context.**
>
> References:
>
> [1] David J Ingle, Melvyn A Goodale, Richard JW Mansfield, et al. Analysis of visual behavior. Mit Press Cambridge, MA, 1982.
>
> [2] Jonathan J Nassi and Edward M Callaway. Parallel processing strategies of the primate visual system. Nature reviews neuroscience, 10(5):360–372, 2009.
>
> [3] Giacomo Rizzolatti and Massimo Matelli. Two different streams form the dorsal visual system: anatomy and functions. Experimental brain research, 153:146–157, 2003.
>
> [4] JH Maunsell and DAVID C van Essen. The connections of the middle temporal visual area (mt) and their relationship to a cortical hierarchy in the macaque monkey. Journal of Neuroscience, 3(12): 2563–2586, 1983.
>
> [5] Gene J Blatt, Richard A Andersen, and Gene R Stoner. Visual receptive field organization and cortico-cortical connections of the lateral intraparietal area (area lip) in the macaque. Journal of Comparative Neurology, 299(4):421–445, 1990.
>
> [6] Richard A Andersen, C Asanuma, G Essick, and RM Siegel. Corticocortical connections of anatomically and physiologically defined subdivisions within the inferior parietal lobule. Journal of Comparative Neurology, 296(1):65–113, 1990.
>
>
> Thank you again for your valuable suggestions to improving our work. We believe that, under your review, our manuscript will be significantly improved in terms of clarity and experimental design. We look forward to your feedback and further discussions.

---

> ### Comment · Reviewer_5KvP · 2024-11-26
> **Statistical assessments**
>
> Thanks for incorporating the statistical assessments.
> I feel only partially satisfied by this analysis and related changes, as follows:
> 1) Reading data directly from the table, I find it very difficult to map it to your conclusions (e.g., outperforming in 6/8 or 3/4 metrics) given the presentation style, which highlights significance even when it supports your method being actually significantly inferior.
> 2) You used t-test though I am not sure your data is Gaussian. Perhaps consider Mann-Whitney or Wilcoxon.
> 3) I am missing simple details on how you created the distributions used to compute significance (Algorithm 2 looks too cryptic to me). Related, it says all metrics are averaged across subjects in Table 2 - why is this the case? Is significance assessed on such pooled data as well? Method assessment should be made per-subject as these are subject-specific analyses.
> 4) Focus of your work not being on semantics. I believe this works aims simultaneously reconstruct of visual structure, semantics, and motion patterns. Not trade one aspect with another.
> 5) pretraining + fine-tuning likely contributing to Chen et al. (2024) superiority in some cases. That's a conjecture that has to be tested and properly compared apples-to-apples. This cannot, by itself, exempt this newly proposed method from proving its superiority.
> 6) New challenging retrieval tasks. Are there missing asterisks on this table, and if not, would it be fair to say there are no significant gains in favor of the proposed method at the _per-subject_ level? The average analysis appears irrelevant given that the focus of the entire paper and analyses is subject-specific.
> 7) Not outperforming Chen et al. (2024) on the 2-Way-I metric is acceptable. The metrics should either be taken as valid to demonstrate or refute gains or not be used at all. Using them to evaluate your method, and then deeming them invalid post-hoc doesn't make sense to me.

---

> ### Author Response · Authors · 2024-11-28
> **Supplementary Results and Explanations on Statistical Assessments (1/4)**
>
> Thank you very much for taking the time to review our rebuttal and for your careful critique of the shortcomings in our hypothesis testing setup. In Tables 2, 3, and 4 of the main text, we present results averaged across multiple subjects, and the hypothesis tests were conducted after averaging across subjects as well.
>
> Following your suggestion, we have added the Wilcoxon signed-rank test for the individual subject paired t-tests in Tables 5, 15, 16, and 17. (Note that Wang et al. (2022) only publicly reported the mean SSIM and PSNR values for the HCP dataset and did not release the full set of reconstruction results, thus preventing us from conducting hypothesis testing for their results.)
>
> Below, we provide further clarifications in response to your comments and suggestions.
>
> ## 1. A brief explanation of Algorithm 2.
> ## Response:
> Taking the 2-way top-1 accuracy used in this study as an example, the calculation of this metric is as follows: For each reconstruction result ($recons$), a non-ground truth video ($gt*$) is randomly selected from the test set to form a triplet {$recons$, $gt$, $gt*$}. A classification model is then used to compute the logits for the three components and determine whether the $recons$ can be classified in the same category as the $gt$.
>
> For 2-Way-I, the classification model used is ViT-base-patch16-224, with results computed and averaged across all frames of the video. For 2-Way-V, the model used is VideoMAE. Since the selection of non-ground truth videos is random, this process is repeated for 100 trials. The mean of the 100 trials is reported in the tables, and a paired t-test is performed using the results from these 100 trials for hypothesis testing.
>
>
>
>
>
> ## 2. The construction of data distributions for hypothesis testing.
> ## Response:
> For evaluation metrics with random sampling, such as 2-Way-I and 2-Way-V, we created the distributions by repeating the experiment 100 times.
>
> For other metrics, we used the bootstrap method to create the distributions. Taking the SSIM for Subject 1 from the CC2017 dataset as an example: the test set contains 1200 samples. When reporting the results in the tables, we directly calculate the SSIM between the reconstruction and ground truth  and report the average over the 1200 samples. For hypothesis testing, we performed bootstrap sampling with replacement from the 1200 samples, recording the mean of the metric each time, and repeating this process 100 times to obtain 100 means. A paired t-test was then performed using the results from these 100 trials.

---

> ### Author Response · Authors · 2024-11-28
> **Supplementary Results and Explanations on Statistical Assessments (2/4)**
>
> ## 3. Comparison with Chen et al. (2024) 's results on the CC2017, HCP, and Algonauts 2021 datasets and the hypothesis testing results for each subject.
> ## Response:
> After constructing the data distributions as described above, and to account for the potential non-Gaussian nature of the distributions, we followed your suggestion and used the Wilcoxon signed-rank test for paired comparisons for each subject. The comparison with Chen et al. (2024) on the three datasets is shown in the table below. **The full experimental results have been added to Tables 15, 16, and 17, and we have submitted the updated PDF.**
>
> | Sub ID   | Models      | Semantic-level ↑  |    |    | Pixel-level ↑  |    |    | ST-level        |         |         |
> |----------|-------------|--------------------|--------------------|--------------------|----------------|----------------|----------------|------------------|------------------|------------------|
> |          |             | 2-way-I            | 2-way-V            | VIFI-score         | SSIM           | PSNR           | Hue-pcc        | CLIP-pcc ↑      | EPE ↓            |
> | sub 01   | Mind-video  | 0.792***              | **0.853*****              | 0.587***              | 0.171***          | 8.662***          | 0.760***          | 0.408***            | 6.119***            |
> |          | Ours        | **0.812**          | 0.841              | **0.602**          | **0.321**      | **9.124**          | **0.774**      | **0.425**        | **5.580**        |
> | sub 02   | Mind-video  | 0.789***              | **0.842*****              | 0.595***              | 0.172***          | 8.929***          | 0.773***          | 0.409***            | 6.062***            |
> |          | Ours        | **0.811**          | 0.827              | **0.615**          | **0.292**      | **9.250**          | **0.791**      | **0.429**        | **5.329**        |
> | sub 03   | Mind-video  | **0.811****          | **0.848*****          | 0.597***              | 0.187***          | 9.013***          | 0.771***          | 0.410**            | 6.193***            |
> |          | Ours        | 0.792              | 0.823              | **0.607**          | **0.349**      | **9.287**          | **0.794**      | **0.421**        | **5.356**        |
>
> *Quantitative comparison of reconstruction results across three subjects from the **CC2017 dataset**. For the 2-way-I and 2-way-V metrics, 100 repetitions were conducted, while other metrics were evaluated using 100 bootstrap trials. All metrics are averaged over the entire test set. The superior results are highlighted in bold. Asterisks indicate statistical significance (Wilcoxon test for paired samples) compared to our model. p<0.0001(\*\*\*), p<0.01(\*\*), p<0.05(\*).*
>
> | Sub ID   | Models      | Semantic-level ↑  |    |    | Pixel-level ↑  |    |    | ST-level        |         |         |
> |----------|-------------|--------------------|--------------------|--------------------|----------------|----------------|----------------|------------------|------------------|------------------|
> |          |             | 2-way-I            | 2-way-V            | VIFI-score         | SSIM           | PSNR           | Hue-pcc        | CLIP-pcc ↑      | EPE ↓            |
> | sub 01 | Mind-video | 0.798**   | 0.752***   | 0.605***       | 0.123***  | 9.302***  | 0.774***    | **0.486***    | 12.746***  |
> |        | Ours       | **0.819** | **0.783** | **0.613** | **0.325** | **10.757** | **0.820** | 0.476    | **7.825** |
> | sub 02 | Mind-video | **0.761** | **0.777*** ** | **0.611**  | 0.115***  | 9.414***  | 0.804***    | 0.483    | 7.358***  |
> |        | Ours       | 0.756   | 0.759   | 0.609      | **0.371** | **11.894** | **0.834** | **0.485** | **6.624** |
> | sub 03 | Mind-video | 0.779   | 0.778***    | 0.612***       | 0.118***  | 9.109***  | 0.803***    | 0.529***     | 7.767***  |
> |        | Ours       | **0.781** | **0.793** | **0.634** | **0.336** | **11.018** | **0.834** | **0.573** | **6.792** |
>
> *Quantitative comparison of reconstruction results across three subjects from the **HCP dataset**. For the 2-way-I and 2-way-V metrics, 100 repetitions were conducted, while other metrics were evaluated using 100 bootstrap trials. All metrics are averaged over the entire test set. The superior results are highlighted in bold. Asterisks indicate statistical significance (Wilcoxon test for paired samples) compared to our model. p<0.0001(\*\*\*), p<0.01(\*\*), p<0.05(\*).*

---

> ### Author Response · Authors · 2024-11-28
> **Supplementary Results and Explanations on Statistical Assessments (3/4)**
>
> | Sub ID   | Models      | Semantic-level ↑  |    |    | Pixel-level ↑  |    |    | ST-level        |         |         |
> |----------|-------------|--------------------|--------------------|--------------------|----------------|----------------|----------------|------------------|------------------|------------------|
> |          |             | 2-way-I            | 2-way-V            | VIFI-score         | SSIM           | PSNR           | Hue-pcc        | CLIP-pcc ↑      | EPE ↓            |
> | sub 01 | Mind-video | 0.702***   | 0.761***   | 0.568***       | 0.135***  | 8.642***  | 0.794***    | 0.277***    | 8.368***  |
> |        | Ours       | **0.722** | **0.790** | **0.599** | **0.401** | **10.088** | **0.824** | **0.439**    | **4.420** |
> | sub 02 | Mind-video | 0.698*** | **0.769** | 0.573***  | 0.132***  | 9.004***  | 0.773***    | 0.265***    | 7.458***  |
> |        | Ours       | **0.734**   | 0.765   | **0.596**      | **0.465** | **10.932** | **0.796** | **0.425** | **3.806** |
> | sub 03 | Mind-video | **0.701*****   | 0.729***    | 0.564***       | 0.117***  | 8.796***  | 0.806***    | 0.271***     | 7.659***  |
> |        | Ours       | 0.679 | **0.794** | **0.591** | **0.466** | **11.089** | **0.863** | **0.397** | **3.406** |
> | sub 04 | Mind-video | 0.665***      | 0.785***      | 0.556***      | 0.126***      | 8.439***      | 0.811***      | 0.254***      | 8.011***     |
> |        | Ours       | **0.673**      | **0.810**      | **0.587**      | **0.479**      | **11.410**     | **0.848**      | **0.381**      | **3.089**      |
> | sub 05 | Mind-video | 0.664***      | 0.757***      | 0.529***      | 0.140***      | 8.597***      | 0.792**      | 0.263***      | 8.124***      |
> |        | Ours       | **0.689**      | **0.810**      | **0.592**      | **0.458**      | **10.814**     | **0.807**      | **0.406**      | **3.237**      |
> | sub 06 | Mind-video | 0.690*      | 0.751***      | 0.549***      | 0.137***      | 9.011***      | 0.795***      | 0.266***      | 7.431***      |
> |        | Ours       | **0.709**      | **0.783**      | **0.597**      | **0.489**      | **11.337**     | **0.834**      | **0.446**      | **3.399**      |
> | sub 07 | Mind-video | **0.687**      | 0.721***      | 0.574*      | 0.109***      | 8.409***      | 0.783***      | 0.209***      | 7.652***      |
> |        | Ours       | 0.681      | **0.802**      | **0.578**      | **0.458**      | **10.889**     | **0.857**      | **0.329**      | **3.845**      |
> | sub 08 | Mind-video | 0.658***      | 0.764***      | 0.590      | 0.114***      | 8.251***      | 0.817      | 0.204***      | 6.597***      |
> |        | Ours       | **0.709**      | **0.802**      | **0.592**      | **0.467**      | **10.893**     | **0.820**      | **0.376**      | **3.757**      |
> | sub 09 | Mind-video | 0.679***      | 0.780*      | **0.609****      | 0.117***      | 8.673***      | 0.784***      | 0.267***      | 8.102***      |
> |        | Ours       | **0.731**      | **0.788**      | 0.594      | **0.502**      | **11.310**     | **0.820**      | **0.400**      | **3.551**      |
> | sub 10 | Mind-video | 0.663***      | 0.770*      | 0.563***      | 0.108***      | 8.912***      | 0.809***      | 0.185***      | 7.524***     |
> |        | Ours       | **0.684**      | **0.777**      | **0.590**      | **0.465**      | **11.128**     | **0.858**      | **0.408**      | **3.533**      |
>
> *Quantitative comparison of reconstruction results across ten subjects from the **Algonauts2021 dataset**. For the 2-way-I and 2-way-V metrics, 100 repetitions were conducted, while other metrics were evaluated using 100 bootstrap trials. All metrics are averaged over the entire test set. The superior results are highlighted in bold. Asterisks indicate statistical significance (Wilcoxon test for paired samples) compared to our model. p<0.0001(\*\*\*), p<0.01(\*\*), p<0.05(\*).*

---

### Official Review · Reviewer_GXPV · 2024-11-03

**Soundness:** 3
**Presentation:** 3
**Contribution:** 3
**Rating:** 6
**Confidence:** 4

**Summary:**

Paper presents an approach the reconstruct video clips for fMRI brain recordings.
The reconstruction is broken to 3 streams: structure, semantic and motion.
First the fMRI signal is transformed to align with image embeddings, the fMRI embedding is used with pretrained image diffusion models to generate reconstructions.
The results are compared against multiple previous works on a variety of metrics.

**Strengths:**

Competitive results on multi evaluation metrics, the provided reconstruction look good visually.

**Weaknesses:**

- Full reconstruction of the video clips not provided, this would make the work more transparent, and would allow future works to easily compare on new metrics.
- No comparison to other methods is provided for retrieval metric (Which I think is one of the most objective/relevant metrics)
- Authors don't show results for sequences with actual motion, give that one of the focuses of the work is motion, it would make sense to show the ability to reconstruct motion. (for example the clip with the soldier)

**Questions:**

- Figure 7 color scheme is not consistent across plots(relevant to all similar plots)
- I think it makes sense to put retrieval results in a more centric place in the paper.
- The work " Kupershmidt22" provides a retrieval metric for itself and "Wen18", you should consider adding this results.
- It would be helpful to add standard error/ statistical tests to the comparisons between the models.

---

> ### Author Response · Authors · 2024-11-20
> **Response to Reviewer GXPV**
>
> Thank you for recognizing and supporting our work, as well as for taking the time to provide valuable suggestions for improvement. We have thoroughly revised the manuscript based on your feedback, **with changes highlighted in red**, and have submitted the updated version. Below are our detailed responses to your comments.
>
> ## 1. Provide the full reconstruction of the video clips to enhance the transparency of the work.
>
> ## Response:
>
> Thank you for your suggestion. We agree with your perspective that open-sourcing all experimental results would facilitate evaluation and comparison using new metrics for other researchers. We also appreciate the generosity of the works compared in Table 2, which kindly provided their reconstruction results for evaluation. However, due to OpenReview's file size limitation for supplementary materials (no more than 100MB), it is challenging to upload all our reconstruction results, which total 32GB across the three datasets, at this stage. Nevertheless, we assure you that once the paper is accepted, we will make all the preprocessed datasets, code, and reconstruction results publicly available.
>
>
>
> ## 2. Authors don't show results for sequences with actual motion.
>
> ## Response:
> We have provided some reconstructed videos on the anonymous project homepage (https://mind-animator-design.github.io/). On the homepage's first page, several examples demonstrate a strong consistency between the reconstructed results and the ground truth in terms of motion information. For instance, in the sixth video of the second row, we successfully reconstructed a crowd walking forward, and in the fourth video of the fourth row, two people looking up and laughing. Additionally, in the "More samples" section, you can observe reconstructions such as an airplane flying from right to left, a fish swimming from right to left, and a car driving along a road. These results exhibit actual motion and closely align with the ground truth.
>
>
> ## 3. Figure 7 color scheme is not consistent across plots.
> ## Response:
> Thank you for your careful attention to this issue. We have corrected the color schemes in Figure 7(c), 22(c), and 24(c) in the latest submitted manuscript to ensure consistency across all similar figures.
>
> ## 4. Add standard error/ statistical tests to the comparisons between the models.
> ## Response:
> Thank you for your suggestion. In the latest submitted manuscript, we have added statistical tests to the experimental results in Tables 2, 3, 4, and 5. The results show that our model significantly outperforms the previous comparison methods on most metrics, further highlighting its superior performance.

---

> ### Author Response · Authors · 2024-11-20
> **Response to Reviewer GXPV (cont.)**
>
> ## 5. Include more baseline comparisons and position the retrieval results more prominently within the paper.
>
> ## Response:
>
> Thank you for your valuable feedback. We have added  "Wen18" [1],  "Kupershmidt22" [2]   and "Mind-video" [3] as comparison methods in Table 5 and moved the table to Section 5.1 of the main text. Additionally, we have highlighted the description and interpretation of the table in red for clarity.
>
> | Model         | Test set | **Subject 1** |           | **Subject 2** |           | **Subject 3** |           | **Average**  |           |
> |---------------|----------|---------------|-----------|---------------|-----------|---------------|-----------|--------------|-----------|
> |               |          | top-10        | top-100   | top-10        | top-100   | top-10        | top-100   | top-10       | top-100   |
> | Wen [1]          | Small    | 2.17*          | 19.50*     | 3.33*          | 19.17*     | ——            | ——        | 2.75*        | 19.33*    |
> | Kupershmidt [2]   | Small    | 1.09*          | 8.57*      | 0.92*          | 8.24*      | 0.84*          | 8.24*      | 0.95*        | 8.35*     |
> | Mind-video [3]   | Small    | **3.22***      | 19.08*     | 2.75*          | 16.83*     | 3.58*          | 22.08*     | 3.18*        | 19.33*    |
> | **Ours**      | Small    | 3.08      | **22.58** | **4.75**      | **26.90** | **4.50**      | **24.67** | **4.11**     | **24.72** |
> | Wen   [1]        | Large    | 1.41*          | 11.58*     | 2.08*          | 9.58*      | ——            | ——        | 1.75*        | 10.58*    |
> | Kupershmidt [2]   | Large    | 0.17*          | 2.94*      | 0.17*          | 2.77*      | 0.25*          | 2.18*      | 0.19*        | 2.63*     |
> | Mind-video  [3]  | Large    | 1.75*          | 7.17*      | 0.83*          | 5.17*      | 1.25*          | 9.00*      | 1.28*        | 7.11*     |
> | **Ours**      | Large    | **2.17**      | **12.50** | **2.25**      | **17.00** | **2.75**      | **16.42** | **2.39**     | **15.31** |
>
> *Note: For the 'small test set', the chance-level accuracies for top-10 and top-100 accuracy are 0.83% and 8.3%, respectively. For the 'large test set', the chance-level accuracies for top-10 and top-100 accuracy are 0.24% and 2.4%, respectively. The metrics are evaluated using 100 bootstrap trials. * denotes our performance is significantly better than the compared method (Wilcoxon test for paired samples, p<0.05).*
>
>
> It is worth noting that the evaluation metrics used in "Wen18" and "Kupershmidt22" are simpler compared to ours. Specifically, "Wen18" reported classification results in their paper, where video stimuli from the CC2017 dataset were divided into 15 categories, achieving a top-10 accuracy with a chance level of 66.7%. "Kupershmidt22" on the other hand, employed a 100-way Identification Test, which measures whether the corresponding video can be retrieved from a pool of 100 videos, with a top-10 accuracy chance level of 10%. In contrast, our study requires retrieving the corresponding video from 1,200 video clips (small set) and an expanded set of 4,240 video clips (Large set), with top-10 accuracy chance levels of 0.83% and 0.24%, respectively. Thus, our evaluation is more challenging and better reflects the reconstruction performance of the model.
>
> We recalculated the evaluation metrics for "Kupershmidt22",  "Wen18" and "Mind-video" using their reconstruction results on the CC2017 dataset, as recorded in Table 5. Statistical significance tests  across three subjects demonstrate that our model significantly outperforms the comparison methods.
>
> References：
>
> [1] Haiguang Wen, Junxing Shi, Yizhen Zhang, Kun-Han Lu, Jiayue Cao, and Zhongming Liu. Neural encoding and decoding with deep learning for dynamic natural vision. Cerebral cortex, 28(12): 4136–4160, 2018.
>
> [2] Ganit Kupershmidt, Roman Beliy, Guy Gaziv, and Michal Irani. A penny for your (visual)
> thoughts: Self-supervised reconstruction of natural movies from brain activity. arXiv preprint
> arXiv:2206.03544, 2022.
>
> [3] Zijiao Chen, Jiaxin Qing, and Juan Helen Zhou. Cinematic mindscapes: High-quality video reconstruction from brain activity. Advances in Neural Information Processing Systems, 36, 2024.
>
> Thank you again for your valuable suggestions to improving our work.  We believe that, under your review, our manuscript will be significantly improved in terms of clarity and experimental design. We look forward to your feedback and further discussions.

---

> ### Author Response · Authors · 2024-11-28
> **Supplementary Results on Statistical Assessments**
>
> The results presented in Tables 2, 3, and 4 were obtained by first averaging across subjects within each dataset, followed by significance testing on the averaged results. Following the suggestion of Reviewer 5KvP, we have additionally conducted per-subject significance testing, as shown in Tables 15, 16, and 17. These updates have been incorporated into the manuscript and uploaded.

---

### Official Review · Reviewer_1xQ5 · 2024-11-03

**Soundness:** 3
**Presentation:** 3
**Contribution:** 4
**Rating:** 8
**Confidence:** 4

**Summary:**

In this work, the authors present a novel method for video reconstruction from fMRI recordings. There are two core components to this novelty, (1) the method is interpretable and learns separate semantic, structural, and motion features from the fMRI - later used to generate video frames, and (2) the motion component of the generated video is solely based on the motion predicted from the fMRI because the video generator is an inflated image diffusion model. The videos are generated in two stages, fMRI-to-feature which has trainable components, and feature-to-video which is completely frozen. At training time of stage 1, a Semantic decoder using frozen CLIP encoders, a Structure decoder using a frozen VQ-VAE, and a custom transformer-based Consistency Motion Generator with a masked causal frame prediction task are trained to learn the semantic, structural, and motion features respectively. The performance of the reconstruction model is evaluated on 3 public fMRI datasets with 8 different (semantic, pixel, and spatiotemporal) evaluation metrics, compared against previous work, analyzed with ablation studies, and assessed under two different interpretability analyses. The model is found to exceed state-of-the-art both in quantitative and qualitative metrics, yield sensible component contributions in the ablation, and offer interpretability insights that include neurobiological plausibility.

**Strengths:**

This paper has good quality, with sound methodology and extensive experiments. It is an original, systematic, and rational approach that addresses important issues in previous work; namely the entanglement of different types of features that are decoded from fMRI (semantic, structural, motion), and the entanglement of motion information learned by external training of the video generator with the motion information derived from fMRI. The improvements brought forward by this approach are clearly indicated in the quantitative and qualitative results and consist significant contributions in the research aiming to recover, as well as understand, dynamic visual information from brain recordings.

**Weaknesses:**

**Presentation** could be more clear at many points throughout the paper. There are technical parts that require more explanation or are currently creating some confusion, which are outlined as follows in order from more major to more minor issues.
* In section 3.2, it is not mentioned at all how the ground truth text condition $\bf{c}_i$ is obtained - are these from the training dataset of the (frozen) inflated image generator or are these the same as the video captions $\bf{t}$? It would help to have an intuitive explanation of this in figure 3 as well ($\bf{c}$ is not shown at all in the training pipeline).
* In section 5.1, it is not clear why the videos are external in the “External Videos (EV)” case; This is described as “we further fine-tuned the image diffusion model using videos from the CC2017 dataset (with the training set)” which is still videos from within the video-fMRI dataset, already seen during training in earlier stages of the pipeline.
* In section 3.2, authors are not explicit enough in the description of the trainable modules Semantic & Structure decoder; From the notation it can be assumed that the Semantic decoder is just one trainable vector $\bf{f}$ initialized by the fMRI vector, and the Structure decoder is an MLP $D_{Structure}$ (of unknown number of layers) but it needs to be outlined explicitly.
* In the same section, it is also not explicit how the frames are aggregated to create $\bf{v}$ - is it the average of the CLIP visual embedding across all frames or another aggregation?
* In section 4.2, the description of End-Point-Error(EPE) is missing - the reader has no clue what it is, if it is something existing in the literature (missing citation) or introduced by the authors (missing formula). There are also missing citations for some of the other metrics (Hue-pcc, CLIP-pcc).
* In section 3.3, a mention to the VQ-VAE decoder depicted in the figure is missing - the reader is left to wonder about this. Additionally, this section would benefit from more information on the inflation process (e.g. at least an in-line equation).
* In section 5.2, there is no mention of which dataset the ablation study is on.
* In section 3.2, figure 4 is missing $\bf{E_{pos}}$ and would benefit from a more informative caption. Also, the abbreviation LDM used in the text is probably not familiar to all readers (either remove or expand it).
* In the rightmost box of the training pipeline in figure 3 (CMG), the bracket of $\bf{L_{consistency}}$ is placed in a misleading way - a suggestion would be to move the mask matrix to the bottom and have the input and output frames on the right, with $\bf{L_{consistency}}$ connecting the original (masked) future frames with the predicted future frames.
* In figure 2c the arrow for the noise would be more accurate if it pointed to the Motion instead of the Structure box.
* Prior work is sometimes too generically described e.g. “Subsequently, Han et al. (2019), Wen et al. (2018) and Wang et al. (2022) map brain responses to the feature spaces of deep neural network (DNN) to reconstruct video stimuli.” - what DNN?
* In section 6.2, where authors describe “visual cortices”, probably a more adept term is to say “visual cortical areas” or “areas of the visual cortex”.

**Soundness** related issues exist in parts of the methodology, or sometimes its interpretation.
* The reader is not yet completely convinced that motion information is from the fMRI and not from the training videos. It would help to see an ablation with the CMG trained without the fMRI entirely (output from temporal module is then the Q, K, and V of spatial module), and see if the next frames can be predicted at inference from the structural latent fMRI embedding of the first frame. If the CMG module is still good, then the motion information is not from the fMRI but from the videos in the training set.
* The reader finds several issues with the analysis in  6.1. First the y-axes should end at 1.0 as the maximum value of $\sum _i \delta _i$ is 100. Additionally the y-axes are different across the 3 panels which is also misleading. Second, it seems that although p-values are lower with the CMG, they are still very high and much higher than 0.05, meaning that the order of the generated frames does not matter significantly for these metrics. This is not commented on at all by the authors. Here, the reader’s suggestion for a better baseline to compare against (instead of the standard threshold of 0.05) is the p-value of the shuffle test with the ground-truth video, as it is not certain that even this would be below 0.05. Third, it is not clear why the authors are examining the structural metrics for the shuffle test instead of solely the spatio-temporal metrics, and why for the latter only CLIP-pcc is shown and not EPE. In the view of the reader, only CLIP-pcc and EPE are relevant and should be shown. Finally, the results are vastly different across the 3 subjects which is also not (and should be) commented on in the text.
* In section 5.2, table 5, it is observed that the structure metric Hue-pcc is increasing significantly when the structure module is removed. This seems like an important inconsistency in the results, yet the authors do not comment on it. It is expected to provide some sort of explanation, perhaps based on how this specific metric works and on how the w/o Structure videos look, since the other pixel-level metrics seem to decrease.
* In the analysis in 6.2, it is noticeable that the weight proportion of V1 in the motion information is almost double that of the next highest-weighted areas (e.g. TPOJ, MT, V2, V3), which seems very significant and is not (and should be) commented on - this effect is hidden when the whole of LVC weights are added up.

**Questions:**

The reviewer would like the authors to address the points outlined in the “Weaknesses” section, in each case either by making the suggested change or another change that to the authors’ opinion fixes the issue better, or lastly by giving a sufficient (and convincing) explanation of why no change is needed. This way the reviewer’s opinion of the paper would be improved, to a rating of 6 or above.

**Update after rebuttal:** The score has been updated from an initial 5 to an 8, since all of this reviewer's points were addressed in a very careful, timely, and complete manner. More specifically, the crucial points outlined above in "Soundness" were fixed and the additional ablation experiments strengthen the paper and alleviate this reviewers concerns, which on its own brings the paper rating above acceptance threshold. On top of this, the paper's presentation was substantially improved, making it a consistent and clear read, and overall a good paper (8).

**Details Of Ethics Concerns:**

The paper would benefit from an “Ethics Statement” (which does not count toward page limit and is placed at the end) addressing the issue of potentially harmful use cases of “mind reading” (combined with more portable neuroimaging methods).

---

> ### Author Response · Authors · 2024-11-19
> **Response to Reviewer 1xQ5**
>
> We sincerely appreciate your recognition of the contribution and novelty of our work. Thank you for taking the time to point out the shortcomings in the presentation and experimental design.  We have thoroughly revised the manuscript based on the suggestions provided by you, **with the changes highlighted in blue**, and have submitted the updated version. Our point-by-point responses to your comments are as follows.
>
> # Presentation:
> ## 1. What is the text condition $c_i$ in Section3.2 ?
> ## Response:
> We appreciate your pointing out this issue. The Stable Diffusion model used in this study is a text-to-image model, which operates by first encoding the input text into a text condition $c_i$  using its built-in text encoder. The corresponding image is then generated based on this text condition. In the semantic decoding phase, our goal is to fit the fMRI data to the text condition $c_i$  . However, $c_i$  has a very high dimensionality (1x20x768), and directly fitting the two would lead to severe overfitting. Therefore, we chose the CLIP representation space, which has lower dimensionality (1X512) and strong generalization capability, as an intermediary. First, we align the fMRI representation f with the CLIP text representation t and visual representation v using a multi-modal contrastive loss ($L_{Semantic}$). Then, we map the fMRI representation f to the text condition $c_i$  using a projection loss ($L_{Projection}$). Therefore, the ground truth text condition is obtained by inputting the text description into the text encoder of Stable Diffusion.
>
> We have made the corresponding revisions in Section 3.2 and Figure 3.
>
>
> ## 2. llustration issues (Figure 2(c), Figure 3, Figure 4).
> ## Response:
> We sincerely appreciate your careful attention to these issues. In order to enhance the clarity of our work for readers, we have made the following revisions: In Figure 2(c), we have adjusted the arrows to point to "Motion"; in Figure 3, we have modified the layout of the rightmost panel following your suggestion; and in Figure 4, we have added the positional encoding vectors $E_{pos}$ .
>
>
> ## 3. In Section 5.1, it is not clear why the videos are external in the “External Videos (EV)” case.
> ## Response:
> We apologize for the confusion caused by our phrasing. Our model consists of two stages: the fMRI-to-feature stage and the feature-to-video stage. Although we used a video-fMRI dataset in the fMRI-to-feature stage, to ensure that the motion information in the reconstructed video comes solely from the fMRI data, we employed Inflated Stable Diffusion rather than a text-to-video model in the feature-to-video stage. Therefore, the term "external videos" refers to Stable Diffusion in this context. Specifically, the generative model we used was trained only on image datasets and has never been exposed to "external videos." This issue was similarly pointed out by Reviewer 5KvP. To avoid any potential misunderstanding among readers, we have decided to remove the experiments related to "Ours with EV" from the main text.
>
> ## 4. In Section 3.2, the description of the structure regarding the Semantic & Structure decoder is insufficiently detailed.
> ## Response:
> We apologize for this oversight in the manuscript. The Semantic decoder is a 3-layer MLP, while the Structure decoder is a 2-layer MLP. We have now included a detailed explanation of this in Section 3.2 of the latest version of the manuscript.
>
> ## 5. In Section 3.2, how the frames are aggregated to create $v$ ?
> ## Response:
> For each video, we input each frame into the CLIP visual encoder and then compute the average across all frames to obtain $v$. To clarify, we have provided additional details in Section C1 of the Appendix.
>
> ## 6. In Section 4.2, the description and citations of End-Point-Error (EPE), Hue-pcc, and CLIP-pcc are missing.
> ## Response:
> End-Point Error (EPE) is calculated as the Euclidean distance between the endpoints of the predicted and ground truth trajectories for each corresponding frame, providing a measure of the similarity between the motion trajectories of the predicted and ground truth videos. The Hue-PCC (Hue-based Pearson Correlation Coefficient) is calculated by first converting the frames to the HSV color space, then computing the Pearson correlation coefficient (PCC) between the hue values of the two frames. This metric measures the linear relationship between the hue distributions of the frames, capturing their color similarity.
>
> We have added a supplementary description of EPE and citations for these three evaluation metrics in the relevant section of Section 4.2.

---

> ### Author Response · Authors · 2024-11-19
> **Response to Reviewer 1xQ5 (cont.)**
>
> ## 7. In Section 3.3, the description of the VQ-VAE decoder and the inflation process is insufficiently detailed.
> ## Response:
> We apologize for the oversight that may have caused confusion for the readers. Thank you for pointing out this issue.
>
> **The inflation process refers to utilizing a Stable Diffusion model pre-trained on 2D data (images) to directly process 3D data (videos).** The specific process involves:
>
> After the motion features $\Phi(\mathbf{v}_{i}) \in \mathbb{R}^{B \times f \times 3 \times \frac{H}{8} \times \frac{W}{8}}$  are decoded, they are reshaped ($(B, f, 3, \frac{H}{8}, \frac{W}{8}) \rightarrow (B \cdot f, 3, \frac{H}{8}, \frac{W}{8})$) and input into the
>  Stable Diffusion U-Net for reverse denoising.
>
> **The result is then mapped back to pixel space through the VQ-VAE decoder** and reshaped ($(B \cdot f, 3, H, W) \rightarrow (B , f, 3, H, W)$) to yield the final video  $\mathbf{v}_{i} \in \mathbb{R}^{1 \times f  \times 3 \times H \times W}$. In this context, $B$ denotes the batch dimension, with $B = 1$ during inference.
>
> To facilitate the readers' understanding, we have also provided a more detailed description of these two techniques in Section 3.3.
>
> ## 8.In Section 5.2,  there is no mention of which dataset the ablation study is on.
> ## Response:
> The ablation study in Table 6 was conducted on sub1 of the CC2017 dataset, while the experimental results for sub2 and sub3 are presented in Table 12 and Table 13 of the Appendix. Due to time and space constraints, the parameter sensitivity analysis was only performed on sub1 of the CC2017 dataset, as shown in Table 10 and Table 11.
>
> We have added the relevant explanations in the captions of Table 6, 10, 11, 12, and 13.
>
> ## 9. Prior work is sometimes too generically described
> ## Response:
> Han et al. mapped fMRI data to a VAE [1] pretrained on the ImageNet ILSVRC2012 dataset [2] to reconstruct a single frame, while Wen et al. mapped fMRI data to the feature space of AlexNet  [3] and used a deconvolutional neural network [4] to reconstruct a single frame. Wang et al. developed an f-CVGAN that learns temporal and spatial information in fMRI through separate discriminators [5].
>
> Due to space limitations, these details are not included in the main text but are fully provided in Appendix A.
>
> ## 10. In Section 6.2, there is inconsistent wording.
> ## Response:
> Thank you for pointing out this issue. Following your suggestion, we have replaced 'visual cortices' with 'visual cortical areas' in the relevant sentences of Section 6.2.
>
>
> References：
>
> [1] Diederik P Kingma and Max Welling. Auto-Encoding Variational Bayes. International Conference on Learning Representations, 2014.
>
> [2] Olga Russakovsky, Jia Deng, Hao Su, Jonathan Krause, Sanjeev Satheesh, Sean Ma, Zhiheng Huang, Andrej Karpathy, Aditya Khosla, Michael Bernstein, et al. Imagenet large scale visual recognition challenge. International journal of computer vision, 115:211–252, 2015.
>
> [3] Alex Krizhevsky, Ilya Sutskever, and Geoffrey E Hinton. Imagenet classification with deep convolutional neural networks. Advances in neural information processing systems, 25, 2012.
>
> [4] Matthew D Zeiler, Dilip Krishnan, Graham W Taylor, and Rob Fergus. Deconvolutional networks. In 2010 IEEE Computer Society Conference on computer vision and pattern recognition, pp. 2528–2535. IEEE, 2010.
>
> [5]  Ian Goodfellow, Jean Pouget-Abadie, Mehdi Mirza, Bing Xu, David Warde-Farley, Sherjil Ozair, Aaron Courville, and Yoshua Bengio. Generative Adversarial Networks. Communications of the ACM, 63(11):139–144, 2020.

---

> ### Author Response · Authors · 2024-11-19
> **Response to Reviewer 1xQ5 (cont.)**
>
> # Soundness:
> ## 1. Further ablation studies on the CMG module are needed to confirm that the motion information originates from the fMRI rather than the training videos.
> ## Response:
> Thank you for your constructive suggestion. Our team has also been considering whether more direct evidence could demonstrate that the motion information originates from the fMRI rather than the training videos. Following your advice, we removed the fMRI guidance during the training of the CMG module by replacing the cross-attention in the Spatial module with self-attention, while keeping the rest of the architecture and hyperparameters unchanged. Due to time and computational resource constraints, we conducted experiments only on sub1 and sub2 of the CC2017 dataset. **The results for sub1 with t-test have been incorporated into Table 6 of the main text, while the training and validation loss curves, as well as the results for sub2, are provided in Appendix E.5.**
>
> Based on the experimental results, we draw the following observations and conclusions:
>
> -  From the training and validation loss curves, it is evident that removing fMRI guidance reduces the generalization ability of the CMG module.
>
> - As shown in Table 6 and Table 12, removing fMRI guidance significantly affects the decoding of motion information, as evidenced by a substantial decrease in CLIP-pcc and a significant increase in EPE. This confirms that the motion information originates from the fMRI rather than the training videos.
>
> - Removing fMRI guidance has minimal impact on some semantic- and structural-level metrics, and even leads to a significant increase in PSNR, a measure of video signal-to-noise ratio. We hypothesize that this is because the inherently low signal-to-noise ratio of fMRI, while introducing motion information to video reconstruction, also introduces adverse effects such as reducing the generation quality of Stable Diffusion. This presents a challenge worth addressing in future research.
>
> ## 2.Issues and suggestions with the analysis in 6.1.
> ### (1) The labeling error on the y-axis
> ### Response:
>
> We apologize for the labeling error on the y-axis, which may have caused confusion in your reading. First, we would like to clarify that the values in the bar chart did not exceed 1.0. The y-axis scale exceeding 1.0 occurred because, when using matplotlib to plot the graph, we set the legend to appear in the upper left corner, which caused the y-axis to be automatically extended. We have corrected this issue in the latest version of the figure.
>
> ### (2) Why the authors are examining the structural metrics for the shuffle test instead of solely the spatio-temporal metrics?
> ### Response:
> We believe that shuffling the video frames does not affect the semantic-level metrics. Therefore, our original plan was to measure the shuffle test results for all structure-level and ST-level metrics. However, during the shuffle test experiments, our Table 2 only included 7 evaluation metrics: 3 semantic-level metrics, 3 structure-level metrics, and CLIP-pcc as the only ST-level metric. Towards the submission deadline, following our advisor's suggestion, we added the EPE evaluation results to Tables 2 and 6, but we forgot to include the EPE results in the shuffle test. Based on your suggestion, we have now removed the structure-level metrics and retained only the ST-level metrics (CLIP-pcc and EPE) in the latest version of the paper.
>
> ### (3) Using a better baseline to compare against
> ### Response:
> Thank you for your valuable suggestion. However, we believe that using the noise ceiling as a baseline is more reliable than directly using the ground truth. Specifically, we input the semantic feature $c$ and motion feature $z_{1:8}$ from the test set directly into Inflated Stable Diffusion and use the generated results as the noise ceiling for video reconstruction. We conducted a shuffle test on the noise ceiling for both CLIP-pcc and EPE, and the estimated p-values were 0.09 ± 0.01 and 0.005 ± 0.004, respectively, showing that:
>
> - Even for the noise ceiling reconstruction results, the p-value from the shuffle test on CLIP-pcc is significantly greater than 0.05. We believe this is related to the calculation method of CLIP-pcc, which measures semantic similarity between adjacent frames, focusing more on frame-to-frame consistency rather than the order of all frames. Therefore, this metric is not sensitive to shuffling video frames. Nonetheless, even in this case, the p-value with CMG is still significantly smaller than without CMG, indicating that our CMG can capture correlations between frames.
>
> - The p-value for the noise ceiling reconstruction results in EPE is significantly smaller than 0.05. This is because EPE calculates the distance between the reconstructed result and the ground truth optical flow trajectories, considering the order of all frames. Therefore, EPE serves as a better metric for evaluating motion decoding capability.

---

> ### Author Response · Authors · 2024-11-19
> **Response to Reviewer 1xQ5 (cont.)**
>
> ### (4) The p-values are very high and much higher than 0.05
> ### Response:
>
> Following your suggestion, we have removed the structure-level evaluation metrics and retained only CLIP-pcc and EPE. The p-value estimated from the shuffle test for EPE is significantly smaller than 0.05, indicating that we have indeed decoded some motion information from fMRI.
>
> ### (5) Why the results are vastly different across the 3 subjects ?
> ### Response:
>
> In the revised figure, the estimated p-values of shuffle test for the EPE metric across all three subjects are significantly smaller than 0.05, showing consistent results. However, there are vast differences in the p-values for CLIP-pcc and the previously tested structure-level metrics between sub1, sub2, and sub3. We believe the following factors may explain these variations:
>
> - Due to individual differences in brain structure and function, even when subjects watch the same stimulus video, their brain responses can differ significantly [1].
>
> -  According to Kupershmidt et al.'s paper, 'A Penny for Your (Visual) Thoughts: Self-Supervised Reconstruction of Natural Movies from Brain Activity' [2] , in the final paragraph of the Appendix, they calculated the signal-to-noise ratio (SNR) of the fMRI data from the three subjects in the CC2017 dataset. The results were $SNR_{sub1} = 1.16$,  $SNR_{sub2}  = 0.96$, and $SNR_{sub3}  = 0.63$. Therefore, the substantial differences observed across subjects in various metrics may also be attributed to noise in the fMRI data.
>
> ## 3. In Section 5.2, why the structure metric Hue-pcc is increasing significantly when the structure module is removed ?
> ## Response:
> Thank you for pointing out this issue.
> Firstly, as noted in Table 6, removing the structure decoder leads to a significant degradation in 7 out of 8 metrics. Overall, we believe this result is acceptable when considered comprehensively.
>
> To address your concern regarding the notable increase in Hue-PCC, we first explain the extraction process of structure features. In this study, structure features are obtained by extracting the first-frame features of videos using a VQ-VAE model pre-trained on a natural image dataset. Since VQ-VAE is not explicitly trained to disentangle and preserve color information in its latent space, reconstructing hue information from fMRI data is inherently challenging. To overcome this, prior research [3] has proposed explicitly extracting image color information using spatial color palettes, achieving promising results. However, this study focuses on a different goal: decoding motion information from fMRI. Thus, we did not specifically emphasize hue recovery, which will be considered as a potential direction for future improvement.
>
> We have added a discussion on the anomalous changes in this metric at the corresponding position in Section 5.2 of the main text.
>
>
> References:
>
> [1]  Haxby J V, Guntupalli J S, Connolly A C, et al. A common, high-dimensional model of the representational space in human ventral temporal cortex[J]. Neuron, 2011, 72(2): 404-416.
>
> [2] Ganit Kupershmidt, Roman Beliy, Guy Gaziv, and Michal Irani. A penny for your (visual) thoughts: Self-supervised reconstruction of natural movies from brain activity. arXiv preprint arXiv:2206.03544, 2022.
>
> [3] Xia W, de Charette R, Oztireli C, et al. Dream: Visual decoding from reversing human visual system[C]//Proceedings of the IEEE/CVF Winter Conference on Applications of Computer Vision. 2024: 8226-8235.

---

> ### Author Response · Authors · 2024-11-19
> **Response to Reviewer 1xQ5 (cont.)**
>
> ## 4. In the analysis in Section 6.2, why the weight proportion of V1 in the motion information is almost double that of the next highest-weighted areas?
> ## Response:
>
> We sincerely thank you for pointing out this important observation. After reviewing the relevant literature, we identified the reason why V1 plays a dominant role in decoding motion features:
>
> - Parallel processing is a key characteristic of the visual system [1] , and in the dorsal pathway, motion information is not processed strictly hierarchically [2] . **Experimental evidence suggests that the direct pathway from V1 to MT primarily conveys information about motion speed and direction.**  Additionally, several indirect pathways originating from V1 (e.g., through V2 and V3) also transmit related information to MT [3] [4] . **(Figure 26(b) in Appendix G provides a more intuitive illustration of this process.)**
>
> - The earliest paper in the field of video reconstruction utilized neural encoding to map cortical projections [5] , as illustrated in Figure 2(c) of their work. Their results also demonstrated that, compared to other brain regions, V1 exhibited the strongest activation in response to motion information.  Following your suggestion, we have discussed this phenomenon in Section 6.2 of the main text, which has significantly improved our manuscript by enhancing its comprehensiveness and strengthening its contribution to the field of video reconstruction.
>
>
> Therefore, the observation shown in Figure 8, where V1 contributes the most to motion information decoding, is reasonable.
>
> Additionally, to facilitate readers' understanding of the neuroscience background, we have included two illustrative figures in Appendix G and provided a concise explanation of how the human visual cortex processes visual information.
>
> References:
>
> [1] Jonathan J Nassi and Edward M Callaway. Parallel processing strategies of the primate visual system. Nature reviews neuroscience, 10(5):360–372, 2009.
>
> [2] Edward M Callaway. Structure and function of parallel pathways in the primate early visual system. The Journal of physiology, 566(1):13–19, 2005.
>
> [3] Semir Zeki and Stewart Shipp. The functional logic of cortical connections. Nature, 335(6188): 311–317, 1988.
>
> [4] Carlos R Ponce, Stephen G Lomber, and Richard T Born. Integrating motion and depth via parallel
> pathways. Nature neuroscience, 11(2):216–223, 2008.
>
> [5] Shinji Nishimoto, An T Vu, Thomas Naselaris, Yuval Benjamini, Bin Yu, and Jack L Gallant. Reconstructing visual experiences from brain activity evoked by natural movies. Current biology, 21(19):1641–1646, 2011.
>
>
>
>
> Finally, we would like to express our sincere gratitude for raising important questions and providing valuable suggestions for improving our manuscript. We also appreciate your time and patience in thoroughly reviewing our responses. We believe that, under your review, our manuscript will be significantly improved in terms of clarity and experimental design. We look forward to your feedback and further discussions.

---

> > ### Comment · Reviewer_1xQ5 · 2024-11-24
> >
> > Thank you for carefully considering and implementing all suggestions.
> >
> > Regarding the presentation issues, the paper is now much more readable and should not confuse a future reader (at least on the points raised in this review).
> > The only objection would be on the complete removal of the EV results, as they are still valuable to compare with competitors and also offer transparency. Nevertheless, this reviewer also agrees with reviewer 5KvP that it is not fair to include them as part of "Ours", as an integral part of "Ours" is the lack of video finetuning. This could be debated some more on the basis of "are the videos really external, as they are used to train other parts of the pipeline as well", but the claim of a completely video-agnostic generator is still a big part of the paper. Recommendations would be to either (1) keep them in the main table, but put them above the line and don't call them "Ours", or (2) put them in the Appendix (and reference it in main text). In either case, EV should be named something else (e.g. SD-video-finetuning), as the name EV points to videos external to CC2017.
> >
> > As for the soundness related points, the ablation removing the fMRI guidance alleviates this reviewer's main concerns and strengthens the papers' argument considerably, especially when considering the motion metrics. Comparing the removal of the whole CMG module (w/o Motion) with the removal of fMRI guidance from the CMG (w/o fMRI guidance), it is observed that the latter makes up most of the impact of the former (i.e. the whole CMG module improves performance by 0.037 in CLIP-pcc, out of which 0.032 comes from the fMRI guidance, and by 0.802 in EPE, out of which 0.721 comes from the fMRI guidance). An additional suggestion would be to explicitly mention this comparison.
> > The rest of the soundness points were also improved to satisfaction.

---

> > > ### Author Response · Authors · 2024-11-25
> > > **Thanks for your suggestions!**
> > >
> > > We sincerely appreciate the time you spent reviewing our responses and the revised manuscript, as well as your valuable feedback.
> > >
> > > In response to your comment regarding "the complete removal of the EV results," we have made the following improvements: Due to space limitations, we have moved the results of fine-tuning Stable Diffusion on the video dataset used in our model to Appendix E 6.1, Table 14, and referenced it in the main text's Table 2. Additionally, regarding the naming issue, we have made the necessary corrections in both Table 2 and Table 14. In Table 2, we use the symbol $\dagger$ for Mind-video to denote "using Stable Diffusion fine-tuned on video data." In Table 14, we have removed "Ours" as per your suggestion and replaced it with "SD-video-finetuning".
> > >
> > > In regard to the comparison between "w/o Motion" and "w/o fMRI guidance," we have added the following explanation in lines 431–436 of Section 5.2:
> > >
> > > "Meanwhile, comparing the removal of the whole CMG module (w/o Motion) with the removal of fMRI guidance from the CMG (w/o fMRI guidance), it is observed that the latter contributes to the majority of the impact of the former on ST-level metrics. Specifically, in the CLIP-pcc metric, 86% of the decrease observed in the w/o Motion scenario can be attributed to the absence of fMRI guidance, while in the EPE metric, 90% of the decrease is due to the removal of fMRI guidance. This further emphasizes the critical role of fMRI guidance in decoding accurate motion information from brain signals."
> > >
> > > All the points mentioned above have been addressed in the manuscript, and the updated PDF has been provided for your review. We greatly appreciate your valuable feedback and look forward to your response and further discussion.

---

> > > > ### Comment · Reviewer_1xQ5 · 2024-11-25
> > > > **Score update**
> > > >
> > > > Thank you for these additional revisions, much appreciated!
> > > >
> > > > The score has been updated to 8, as the concrete improvements with respect to Soundness and Presentation reinforce this paper's contributions, and make it a good paper to this reviewer's opinion (see also updated comment in Questions part of the original review).

---

> > > > > ### Author Response · Authors · 2024-11-26
> > > > > **Thanks for your recognition!**
> > > > >
> > > > > We would like to express our sincere gratitude to the reviewer for spending time and effort in providing constructive suggestions for our manuscript and for recognizing our work. We believe that under your guidance, our manuscript has achieved a higher level in terms of content readability and experimental integrity.

---

### Author Response · Authors · 2024-11-23
**Official Comment by Authors**

Dear reviewers,

Thank you for your valuable time, insightful comments, and useful suggestions. We have made thorough revisions in the latest PDF submission based on your feedback. To address each reviewer’s comments, **we have highlighted the changes in different colors: Reviewer 1xQ5 in blue, Reviewer GXPV in red, Reviewer 5KvP in green, and Reviewer fFuu in purple.**

Our point-by-point response to the reviewers’ comments has been added to the individual chat box for each reviewer. We are confident that, thanks to the insightful suggestions and constructive feedback from the reviewers, our manuscript will experience substantial improvements in both its clarity and the thoroughness of the experiments.

---

> ### Comment · Reviewer_fFuu · 2024-11-24
>
> Dear Authors,
>
> Thanks for your detailed response and additional experiments to improve your work. It is greatly appreciated.
>
> Bests,
>
> Reviewer fFuu

---

> > ### Author Response · Authors · 2024-11-25
> > **Response to Reviewer fFuu**
> >
> > Dear Reviewer fFuu,
> >
> > Thank you very much for your kind words and feedback. We greatly appreciate your time and effort in reviewing our work and are glad that the additional experiments and revisions have improved the manuscript.
> >
> > Best regards,
> >
> > Authors

---

### Meta-Review · Area_Chair_VmHG · 2024-12-20

**Metareview:**

The paper presents a framework for reconstructing videos from fMRI signals, where the proposed method first decomposes the signals to characterize the semantics, structure, and dynamics of content in the videos, the final reconstruction is produced by passing the decoded video signals through an inflated Stable Diffusion model. Experiments are presented on three datasets and show promising results.

The paper received overall positive scores with two accepts and two borderline accepts. The reviewers liked the overall approach, especially the disentanglement of the fMRI features and the competitive reconstruction improvements on public benchmarks.

**Additional Comments On Reviewer Discussion:**

The paper received a long discussion between the authors and the reviewers. There were several key concerns raised by the reviewers on several aspects of the paper, namely:
* Clarity and soundness in the technical details (Reviewers 1xQ5, 5KvP) and
* Qualitative/substantiative improvements or comparisons (Reviewers GXPV, 5KvP)
* Missing ablation studies (Reviewer fFuu)

Authors revised the paper to fix the issues pointed out by the reviewers, presented qualitative results through an anonymous website that showed reasonable reconstructions, and provided additional numerical results supporting the need for various components in the model, as well as new results comparing the method to prior methods (such as Mind-Video). Overall, the reviewers were satisfied through the discussion.

AC agrees with the reviewers sentiment that the paper makes an interesting attempt at reconstructing video from fMRI signals. The idea of decomposing the signals to the three constituents and extracting motion information to produce videos using a diffusion model is interesting. However, upon independent reading of the revised draft, AC finds several technical issues remaining in the paper. For example, Eq (1) the arguments are not specified in the LHS and the two components in the RHS appear the same, the notation **f** is overloaded and inconsistent across the paper, the text features **t** are not precisely defined, there are issues with the \hat notation through out, and most importantly how the diffusion model is used on the video features is not clearly stated in a mathematically precise manner. Authors are encouraged to fix these issues in the camera-ready paper. As such, the paper is accepted.

---

### Decision · Program_Chairs · 2025-01-22

Accept (Poster)